# Protective effects of macrophage-specific integrin α5 in myocardial infarction are associated with accentuated angiogenesis

Ruoshui Li [1,2], Bijun Chen[1,2], Akihiko Kubota[1,2], Anis Hanna [1,2], Claudio Humeres[1,2], Silvia C. Hernandez [1,2], Yang Liu[3], Richard Ma[3], Izabela Tuleta[1,2], Shuaibo Huang [1,2], Harikrishnan Venugopal[1,2], Fenglan Zhu[1,2], Kai Su[1,2], Jun Li[1,2], Jinghang Zhang[2], Deyou Zheng [3,4,5] & Nikolaos G. Frangogiannis[1,2] ✉

Macrophages sense changes in the extracellular matrix environment through the integrins and play a central role in regulation of the reparative response after myocardial infarction. Here we show that macrophage integrin α5 protects the infarcted heart from adverse remodeling and that the protective actions are associated with acquisition of an angiogenic macrophage phenotype. We demonstrate that myeloid cell- and macrophage-specific integrin α5 knockout mice have accentuated adverse post-infarction remodeling, accompanied by reduced angiogenesis in the infarct and border zone. Single cell RNA-sequencing identifies an angiogenic infarct macrophage population with high *Itga5* expression. The angiogenic effects of integrin α5 in macrophages involve upregulation of Vascular Endothelial Growth Factor A. RNA-sequencing of the macrophage transcriptome in vivo and in vitro followed by bioinformatic analysis identifies several intracellular kinases as potential downstream targets of integrin α5. Neutralization assays demonstrate that the angiogenic actions of integrin α5-stimulated macrophages involve activation of Focal Adhesion Kinase and Phosphoinositide 3 Kinase cascades.

The adult mammalian heart lacks endogenous regenerative capacity and heals through the formation of a collagen-based scar. After myocardial infarction, the sudden death of hundreds of millions of cardiomyocytes generates damage-associated molecular patterns that contribute to the activation of immune and reparative cells[1–3]. Early activation of inflammatory cells plays a central role in the clearance of the infarct from dead cells and matrix debris. Subsequent suppression of inflammation is followed by stimulation of myofibroblasts, the main matrix-secreting cells in the infarct, which are critically involved in the formation of a scar, protecting from catastrophic rupture[4]. Repair of the infarcted heart is dependent on timely recruitment, activation, and de-activation of immune cells, fibroblasts, and vascular cells[5]. Defective or dysregulated cellular responses can cause adverse remodeling, fibrosis, and progression to heart failure.

A large body of evidence has demonstrated that macrophages are central cellular effectors of repair[6,7] and adverse remodeling[8,9] after myocardial infarction regulating a wide range of cellular processes[10]. In addition to their actions in phagocytosing necrotic and apoptotic cells and matrix fragments[11], infarct macrophages are also implicated in initiation[12], progression, suppression[13], and resolution[14,15] of the inflammatory cascade, regulate fibroblast activation[16] and extracellular matrix (ECM) remodeling[17–19] and stimulate angiogenesis[20].

[1]The Wilf Family Cardiovascular Research Institute, Department of Medicine (Cardiology), Albert Einstein College of Medicine, Bronx, NY, USA. [2]Department of Microbiology and Immunology, Albert Einstein College of Medicine, Bronx, NY, USA. [3]Department of Genetics, Albert Einstein College of Medicine, Bronx, NY, USA. [4]Department of Neurology, Albert Einstein College of Medicine, Bronx, NY, USA. [5]Department of Neuroscience, Albert Einstein College of Medicine, Bronx, NY, USA. ✉e-mail: nikolaos.frangogiannis@einsteinmed.edu

The functional diversity of the macrophages in healing infarcts reflects their heterogeneity, and also their capacity to undergo dramatic phenotypic changes in response to a dynamic microenvironment. Alterations in the transcriptomic, proteomic, and functional profile of infarct macrophages are triggered not only by cytokines and growth factors, but also by changes in the ECM network[21].

Cells sense mechanical cues from the ECM through the integrins, a family of heterodimeric surface receptors that bind to components of the ECM, and activate downstream signaling cascades[22,23]. Macrophages express integrin chains on their surface and respond to biophysical stimuli through integrin-mediated pathways[24]. Changes in the biochemical and mechanical properties of the ECM induce marked alterations on the integrin expression profile in macrophages[25]. Moreover, in injured and inflamed tissues, specialized matrix proteins enrich the provisional matrix network, and bind to integrins on the macrophage surface, activating downstream signaling cascades that modulate the functional properties of the cells. The α5 integrin chain (ITGA5) forms a dimer with β1 integrin (ITGB1), and transduces signals upon binding with its main ligand, fibronectin[26], or (under certain conditions) by interacting with matricellular proteins, such as tenascin-C[27] and osteopontin[28]. In vitro studies have demonstrated that ITGA5 may promote a pro-inflammatory and matrix-degrading[29] phenotype in macrophages, through activation of NF-κB signaling[30], and stimulation of the NLRP3 inflammasome[31]. However, the in vivo role of macrophage ITGA5 signaling in regulating myocardial inflammation, repair, and fibrosis has not been investigated.

We hypothesized that ITGA5 signaling plays an important role in the regulation of phenotype and function in infarct macrophages. Our RNA-sequencing data identified *Itga5* as one of the highest expressed and most upregulated members of the integrin family in macrophages infiltrating the healing infarct. Flow cytometry and immunofluorescence showed upregulation of ITGA5 in macrophages 7 days after infarction. Using 2 different Cre drivers that target myeloid cells (LyzM-Cre) and macrophages (inducible CX3CR1-Cre driver), we demonstrated that macrophage-specific ITGA5 signaling protects the infarcted heart from adverse remodeling. These protective effects are associated with the acquisition of an angiogenic macrophage phenotype. ScRNA-seq experiments identified a macrophage cluster with an angiogenic transcriptional profile that expands after infarction and expresses high levels of *Itga5*. Bioinformatic analysis of RNA-sequencing experiments in vivo and in vitro identified candidate integrin-dependent pathways involved in angiogenic macrophage conversion. Neutralization assays demonstrated that the angiogenic effects of ITGA5 are mediated through Focal Adhesion Kinase (FAK) and Phosphoinositide-3-kinase (PI-3K) signaling.

## Results

### Integrin induction in infarct macrophages

In order to study the dynamic changes in the integrin expression profile in infarct macrophages, we compared integrin gene expression between macrophages harvested from normal hearts and infarct macrophages isolated 3 and 7 days after myocardial infarction (Supplementary Fig. 1A). Infarct macrophages exhibited marked induction of *Itga5*, *Itgav* and *Itga6* that was first noted 3 days after infarction and further increased at the 7-day timepoint (Supplementary Fig. 1B–D). *Itgam* expression increased in 3-day infarct macrophages, then returned to baseline at the 7-day timepoint (Supplementary Fig. 1E). *Itgax* levels were markedly increased at both 3 and 7-day timepoints (Supplementary Fig. 1F). In contrast, *Itgb1* was highly expressed in control macrophages and did not significantly change after infarction (Supplementary Fig. 1G). *Itgb2* expression showed an early peak, 3 days after myocardial infarction (Supplementary Fig. 1H), whereas *Itgb5* mRNA levels progressively increased after 3 and 7 days (Supplementary Fig. 1J). *Itgb3* expression remained low in control and infarct macrophages (Supplementary Fig. 1I).

### ITGA5 expression in infarct macrophages peaks 7 days after myocardial infarction

Next, we examined whether the marked induction of *Itga5* mRNA in infarct macrophages is associated with increased ITGA5 protein levels. In order to assess ITGA5 expression in macrophages, we performed flow cytometry and immunofluorescence. The gating strategy for the ITGA5 flow cytometry is shown in Supplementary Fig. 2. Mean fluorescent intensity (MFI) for ITGA5 increased by ~100% in CD45+/CD11b+/Ly6G-/CD64+/MerTK+ macrophages infiltrating the infarct 7 days after coronary occlusion, in comparison to sham heart macrophages (Fig. 1A–C). Moreover, dual immunofluorescence was performed in infarcted CSF1R^EGFP macrophage reporter mice (Fig. 1D–N, Supplementary Figs. 3, 4). No significant ITGA5 immunoreactivity was noted in control mouse hearts. ITGA5+ macrophages were first noted in the infarcted myocardium 3 days after myocardial infarction, and their number markedly increased at the 7-day timepoint (Supplementary Fig. 3). Quantitative analysis showed significant infiltration of the infarcted myocardium with CSF1R+ macrophages 24–72h after infarction (inflammatory phase of infarct healing), followed by a marked increase in macrophage density after 7 days (proliferative phase) and a reduction in macrophage content after 28 days (maturation phase) (Fig. 1J). Confocal microscopy localized ITGA5 immunoreactivity on the cell surface and in the cytoplasm of infarct macrophages (Fig. 1L–N, Supplementary Fig. 4), likely reflecting de novo synthesis of ITGA5 in infarct macrophages and subsequent shuttling to the cell membrane.

### Myeloid cell-specific ITGA5 KO (Myα5KO) mice have no baseline defects

Mice with myeloid cell-specific ITGA5 loss were generated using the LyzM-Cre driver (Supplementary Fig. 5A). Loss of ITGA5 was demonstrated in bone marrow macrophages isolated from Myα5KO mice using western blotting (Supplementary Fig. 5B, C). Moreover, qPCR showed that CD11b+Ly6G- macrophages isolated from Myα5KO infarcts after 7 days of coronary occlusion had marked reduction in *Itga5* mRNA expression (Supplementary Fig. 5D). Young Myα5KO mice were healthy and had normal body weight (Supplementary Fig. 6A–C). The echocardiographic analysis demonstrated that myeloid cell-specific ITGA5 loss had no significant effects on cardiac systolic function, chamber dimensions and left ventricular mass (Supplementary Fig. 6D–O).

### Myeloid cell-specific ITGA5 loss accentuates adverse remodeling after myocardial infarction

Non-reperfused myocardial infarction in mice is associated with significant mortality during the first week after coronary occlusion, which is higher in male mice, and is caused by left ventricular rupture in ~50% of animals[32]. Myeloid cell-specific ITGA5 loss did not have a significant effect on post-infarction mortality in male or female mice (n = 33–35/group, Fig. 2A–C). Echocardiographic analysis showed that myeloid cell-specific ITGA5 deletion accentuated adverse remodeling after infarction, increasing LVEDV (Fig. 2D) and LVESV (Fig. 2E), 28 days after coronary occlusion. The effects of myeloid cell-specific ITGA5 loss on ejection fraction did not reach statistical significance (Fig. 2F). Thus, myeloid cell-specific ITGA5 signaling protects the infarcted heart from adverse dilative remodeling.

### Myeloid cell-specific α5 integrin loss is associated with late scar expansion

Next, we examined whether the accentuated post-infarction remodeling in Myα5KO mice is due to effects on the size of the infarct, or on scar remodeling. In order to accurately measure scar size we sectioned the entire heart from base to apex at 300 μm partitions, thus reconstructing the geometry of the ventricle, using previously published methodology (Fig. 3A)[33,34]. Although at the 7-day timepoint, scar size was comparable between Myα5KO mice and corresponding ITGA5 fl/fl

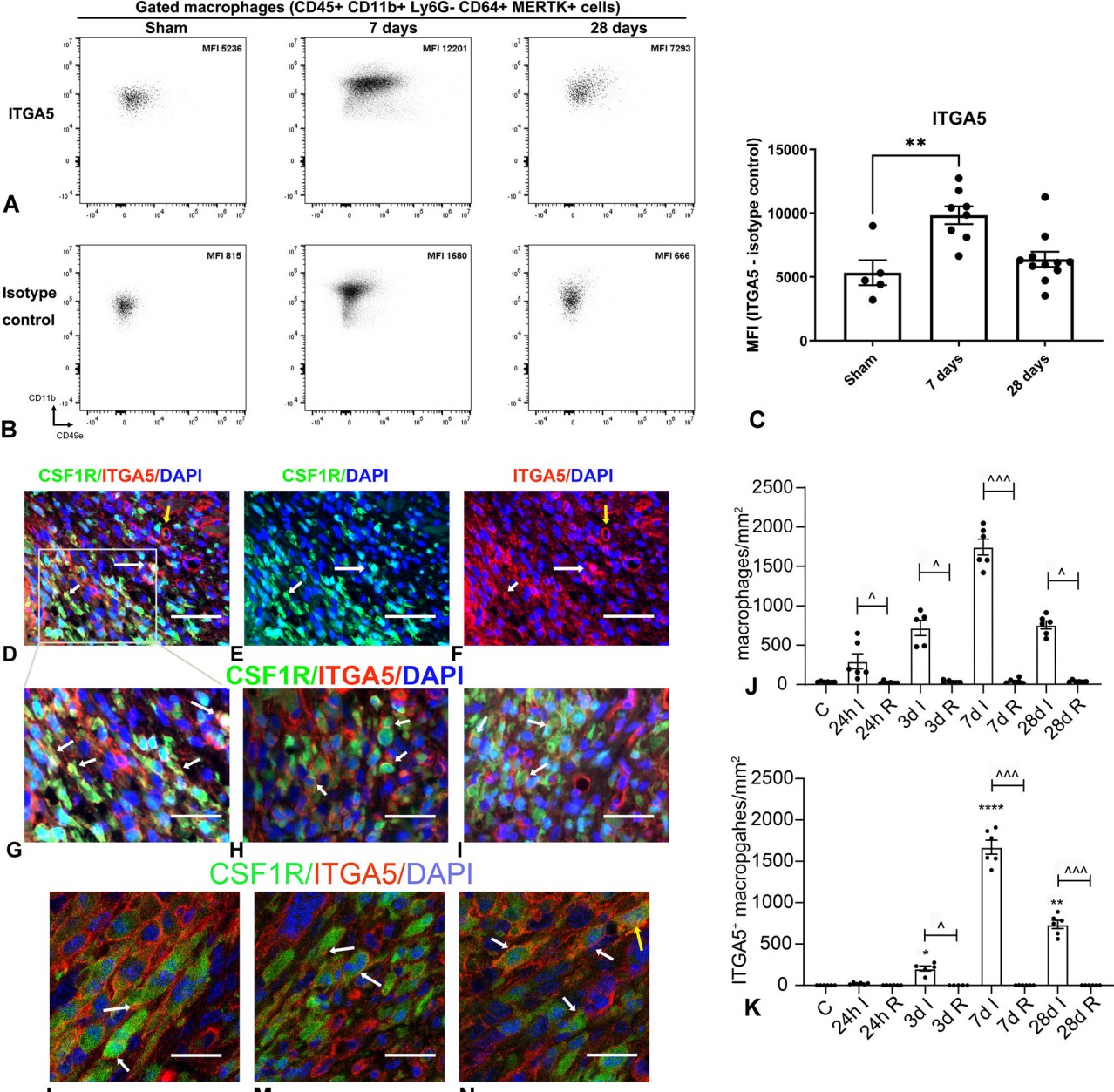

**Fig. 1 | Time course of ITGA5 expression in infarct macrophages. A–C** Flow cytometry was used to assess ITGA5 expression in infarct macrophages. The gating strategy is shown in Supplementary Fig. 2. Representative images show the mean fluorescent intensity (MFI) for ITGA5 antibody (**A**) and isotype control (**B**) in CD45 + /CD11b + /Ly6G-/CD64 + /MerTK+ macrophages in sham animals and in infarcts. Quantitative analysis showed a 2-fold increase in ITGA5 mean fluorescent intensity (MFI) in macrophages 7 days after infarction (**\*\*p < 0.01; Sham: *n* = 5 biologically independent experiments, 7-day: *n* = 8 biologically independent experiments, 28-day: *n* = 11 biologically independent experiments). Statistical analysis was performed using one-way ANOVA followed by Bonferroni post-hoc test. To examine the time course of ITGA5 expression in infarct macrophages, we have also performed dual immunofluorescence in myocardial sections from control and infarcted CSF1R^EGFP macrophage reporter mice, combining ITGA5 staining (red) and CSF1R (GFP) labeling. Abundant ITGA5+ macrophages were noted in 7-day infarcts (**D–I**, arrows). CSF1R-negative cells with vascular, or fibroblast morphology (**D–F**, yellow arrow) also expressed ITGA5. **J** Quantitative analysis showed that the density of CSF1R+ macrophages was significantly higher in infarcted segments, when compared with remote non-infarcted myocardium from the same timepoint (^*p < 0.05, ^^^p < 0.001, *n* = 6 biologically independent experiments in control (C),

24-hour, 7-day and 28-day groups, *n* = 5 biologically independent experiments in the 3-day group). Statistical analysis was performed using non-parametric ANOVA (Kruskal-Wallis) followed by Dunn's post-hoc test. No significant increase in the density of myeloid cells was noted in non-infarcted remodeling segments. **K** An increase in the density of ITGA5+ macrophages was first noted 3 days after coronary occlusion, and peaked after 7 days of coronary occlusion, but was significantly reduced at the 28-day timepoint (*p < 0.05, \*\*p < 0.01 \*\*\*\*p < 0.0001 vs. C ^p < 0.05, ^^^p < 0.001, vs. corresponding remote non-infarcted myocardium, *n* = 6 biologically independent experiments in control (C), 24-hour, 7-day and 28-day groups, *n* = 5 biologically independent experiments in 3-day group). Statistical analysis was performed using non-parametric ANOVA (Kruskal–Wallis) followed by Dunn's post-hoc test. **L–N** Confocal microscopy showed that ITGA5 was localized not only on the macrophage surface (white arrows), but also in the cytoplasm (yellow arrow), likely reflecting de novo synthesis of ITGA5, followed by subsequent shuttling to the cell membrane. Representative images were selected from 50 different scanned fields from 6 biologically independent experiments. Scalebar = 50 μm (for panels **D–F**), =30 μm (panels **G–I**), = 10 μm (panels **L–N**). All data are shown as mean values +/- SEM. Source data are provided as a Source Data file.

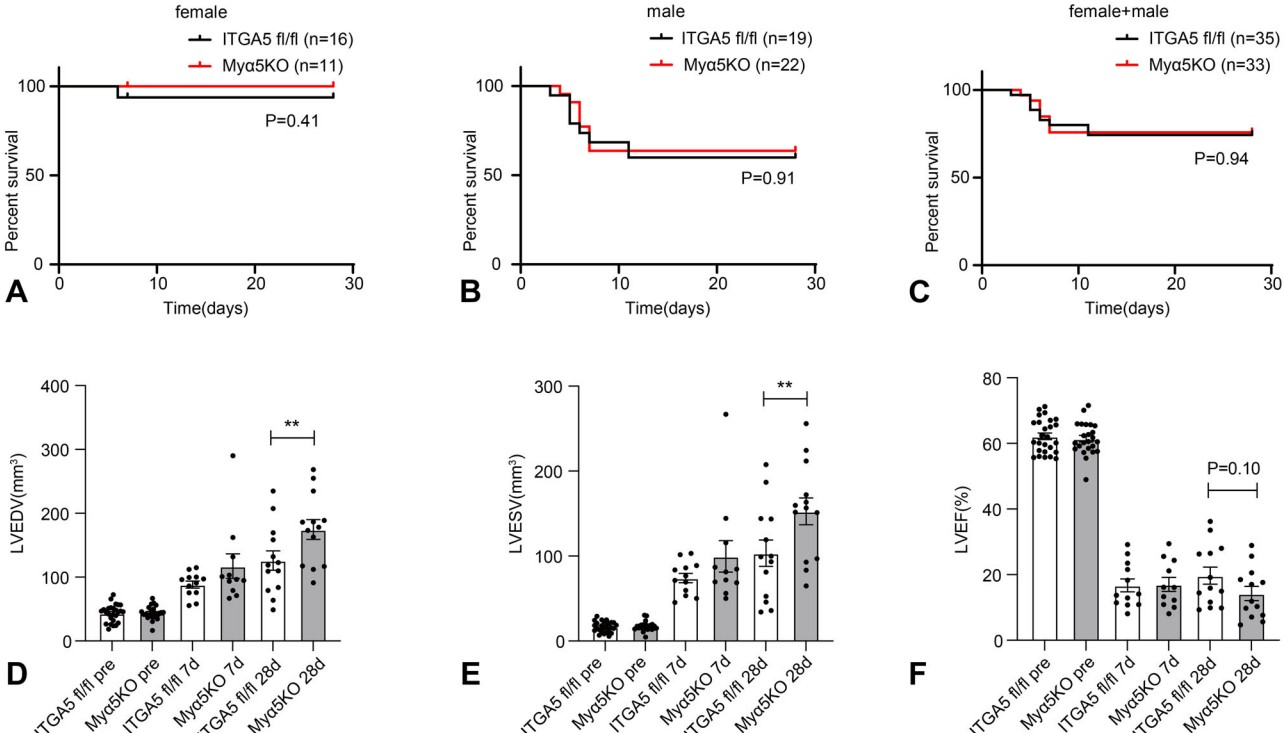

**Fig. 2 | Myeloid cell-specific ITGA5 loss accentuates dilative post-infarction remodeling.** Comparison of the survival curves between ITGA5 fl/fl and Myα5KO (myeloid cell–specific ITGA5 knockout) mice after 7-28 days of permanent coronary occlusion (**A–C**). **A–C**, no significant differences in mortality were noted between ITGA5 fl/fl and Myα5KO mice (ITGA5 fl/fl: $n = 35$, Myα5KO: $n = 33$) in male and female groups. Please note that male mice (**B**) have higher mortality than females (**A**) after myocardial infarction in both genotypes. Statistical analysis was performed using the log-rank test. **D–F** Echocardiographic analysis showed that Myα5KO mice had accentuated dilative remodeling evidenced by increased left ventricular end-diastolic volume (LVEDV; **D**) and left ventricular end-systolic volume (LVESV; **E**) after 28 days. F: Although Myα5KO mice had a trend towards worse systolic dysfunction at the 28-day timepoint, the effects of myeloid cell-specific ITGA5 loss on left ventricular ejection fraction (LVEF; **F**) did not reach statistical significance (**$p < 0.01$, ITGA5 fl/fl pre: $n = 26$, Myα5KO pre: $n = 24$, ITGA5 fl/fl 7d: $n = 12$, Myα5KO 7d: $n = 11$, ITGA5 fl/fl 28d: $n = 13$, Myα5KO 28d: $n = 13$ biologically independent experiments). Statistical analysis was performed using one-way ANOVA, followed by the Sidak post-hoc test. Data are shown as mean values +/- SEM. Source data are provided as a Source Data file.

---

controls, Myα5KO exhibited significantly larger scars 28 days after myocardial infarction (Fig. 3B). The finding reflects impaired scar remodeling in Myα5KO mice that perturbs the contraction of the scar typically noted in WT infarcts. Myα5KO mice also had trends towards lower infarct wall thickness (Fig. 3C) and larger infarct volume (Fig. 3D) at the 28-day timepoint, that did not reach statistical significance.

### Myeloid cell-specific α5 integrin loss does not affect infiltration of the infarct with macrophages

In order to explore the cellular mechanism responsible for the protective effects of myeloid cell-specific ITGA5 signaling, we compared the cellular composition of Myα5KO and ITGA5 fl/fl infarcts. First, we examined the effects of myeloid cell-specific ITGA5 loss on infiltration of the infarcted heart with macrophages using flow cytometry and immunofluorescence. The gating strategy for flow cytometry is shown in Supplementary Fig. 7. Flow cytometry demonstrated that myeloid cell-specific ITGA5 loss did not affect infiltration of the infarct with myeloid cells, neutrophils, macrophages and T cells 7 days after myocardial infarction (Fig. 4). Moreover, immunofluorescent staining demonstrated that Myα5KO and ITGA5 fl/fl animals had comparable macrophage infiltration after 7–28 days of coronary occlusion (Supplementary Fig. 8).

### Myeloid cell-specific ITGA5 loss does not affect myofibroblast infiltration and collagen deposition in the healing infarct

Next, we examined whether myeloid cell-specific ITGA5 loss affects the reparative and fibrotic response after myocardial infarction. We used α-SMA immunofluorescence to label infarct myofibroblasts, as α-SMA

immunoreactive cells located outside the vascular media (Supplementary Fig. 9A). Infarct myofibroblast density in healing infarcts peaks during the proliferative phase of cardiac repair[35,36] (7-day timepoint) and is significantly reduced as the scar matures (28-day timepoint), as myofibroblasts become deactivated and convert to matrifibrocytes[37]. Quantitative analysis showed that myeloid cell-specific ITGA5 loss does not affect myofibroblast density in the infarcted (Supplementary Fig. 9B) and non-infarcted remodeling myocardium (Supplementary Fig. 9C) after 7–28 days of coronary occlusion.

In order to examine whether myeloid cell-specific ITGA5 loss affects collagen deposition in the healing infarct, we performed picrosirius red staining to label collagen fibers (Supplementary Fig. 10). Activated myofibroblasts produce large amounts of collagen in the infarcted heart during the proliferative phase of infarct healing (7-day timepoint). Collagen content in the infarct zone progressively increases as the scar matures (28-day timepoint). Quantitative analysis showed no significant effects of myeloid cell-specific ITGA5 loss on collagen content in the infarct zone and in the remote remodeling myocardium 7–28 days after coronary occlusion (Supplementary Fig. 10B, C).

### Myeloid cell-specific ITGA5 loss perturbs angiogenesis in the infarct and in the border zone

Macrophages are central regulators of the angiogenic response in healing and remodeling tissues[20,38]. In order to examine whether myeloid cell-specific ITGA5 loss affects infarct angiogenesis, we performed immunohistochemical staining for CD31 to identify microvessels (Fig. 5A). Myα5KO mice had significantly reduced microvascular

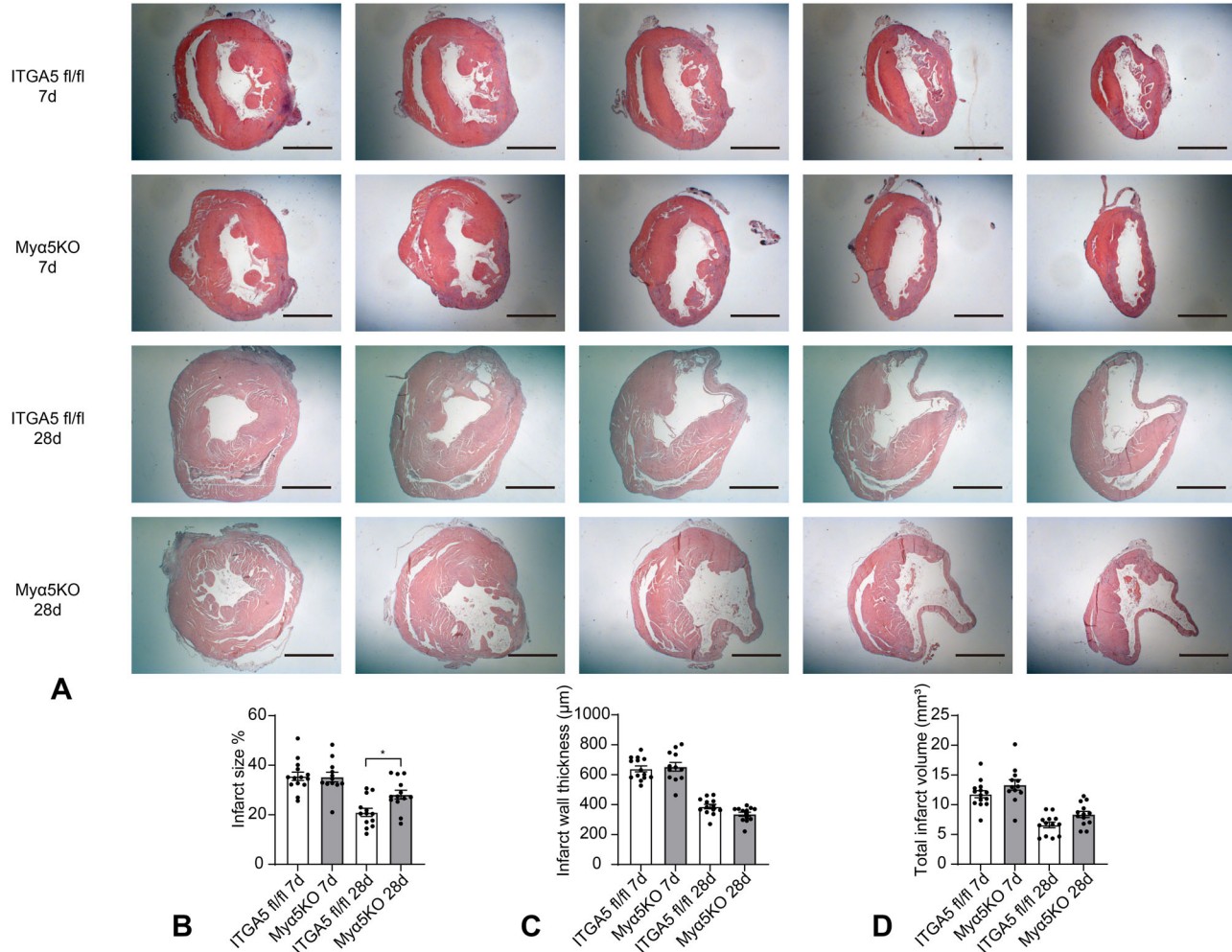

**Fig. 3 | Myeloid cell-specific ITGA5 loss perturbs scar remodeling after myocardial infarction. A** In order to assess scar size after 7-28 days of coronary occlusion, systematic quantification of morphometric parameters was performed by sectioning and staining the entire heart from base to apex. **B** The area of infarcted myocardium was comparable between groups at the 7-day timepoint; however, Myα5KO mice exhibited significantly larger scars 28 days after infarction

(*$p < 0.05$, ITGA5 fl/fl 7d: $n = 14$, Myα5KO 7d: $n = 12$, ITGA5 fl/fl 28d: $n = 13$, Myα5KO 28d: $n = 13$ biologically independent experiments). **C, D** Myα5KO mice had trends towards lower infarct wall thickness and larger infarct volume at the 28-day time-point that did not reach statistical significance. Statistical analysis was performed using one-way ANOVA, followed by the Sidak post-hoc test. Data are shown as mean values +/- SEM. Source data are provided as a Source Data file. Scale bar = 2 mm.

density in the infarct zone at the 7-day timepoint (Fig. 5B), and in the border zone (Fig. 5C) after 28 days of coronary occlusion. Microvascular density in the remote remodeling myocardium was comparable between Myα5KO mice and corresponding ITGA5 fl/fl controls (Fig. 5D). During the maturation phase of infarct healing, microvascular endothelial cells recruit mural cells, resulting in the formation of mature coated vessels[39,40]. In order to identify mature coated vessels in healing infarcts we performed immunofluorescence for α-SMA (Fig. 5E). Quantitative analysis showed that the density of mature coated α-SMA+ microvessels was lower in Myα5KO infarcts after 28 days of coronary occlusion (Fig. 5F). In contrast, the density of mature vessels in the remote remodeling myocardium was not affected by myeloid cell-specific ITGA5 loss (Fig. 5G).

In order to exclude the possibility that the functional and pathologic perturbations in infarcted LyzMCre; ITGA5 fl/fl mice are due to Cre effects, we compared dysfunction and infarct angiogenesis between LyzMCre mice and Cre-negative littermates 7 days after infarction. No significant differences were noted in ejection fraction, LVEDV, the density of microvessels, and mature coated vessels between groups (Supplementary Fig. 11). Thus, myeloid cell-specific ITGA5 loss accentuates adverse remodeling and inhibits infarct angiogenesis.

## Conditional macrophage-specific ITGA5 knockdown using the inducible CX3CR1-Cre driver also accentuates adverse remodeling and inhibits angiogenesis after infarction

LyzM-Cre is a highly effective and widely used Cre driver, but targets all myeloid cells. In order to examine whether the effects of myeloid cell-specific ITGA5 loss are related to abrogation of integrin α5 signaling specifically in macrophages, we used a second Cre driver for inducible macrophage-specific targeting. CX3CR1-CreER mice were used to generate tamoxifen-inducible macrophage-specific ITGA5 KOs (iMaα5KO mice, Supplementary Fig. 12). Loss of ITGA5 was confirmed using immunofluorescent staining of infarct sections from iMaα5KO and ITGA5 fl/fl mice after 7 days of coronary occlusion (Supplementary Fig. 13). iMaα5KO mice were healthy and exhibited no baseline defects. Conditional macrophage-specific ITGA5 loss did not have any effects on body weight and cardiac geometry and function (Supplementary Fig. 14).

After myocardial infarction, iMaα5KO mice exhibited accentuated dilative remodeling, in comparison to ITGA5 fl/fl mice, evidenced by increased LVEDV (Fig. 6A) and LVESV (Fig. 6B) 28 days after coronary occlusion. The effects of macrophage-specific ITGA5 loss on ejection fraction did not reach statistical significance (Fig. 6C). Assessment of microvascular density using CD31 immunohistochemistry (Fig. 6D)

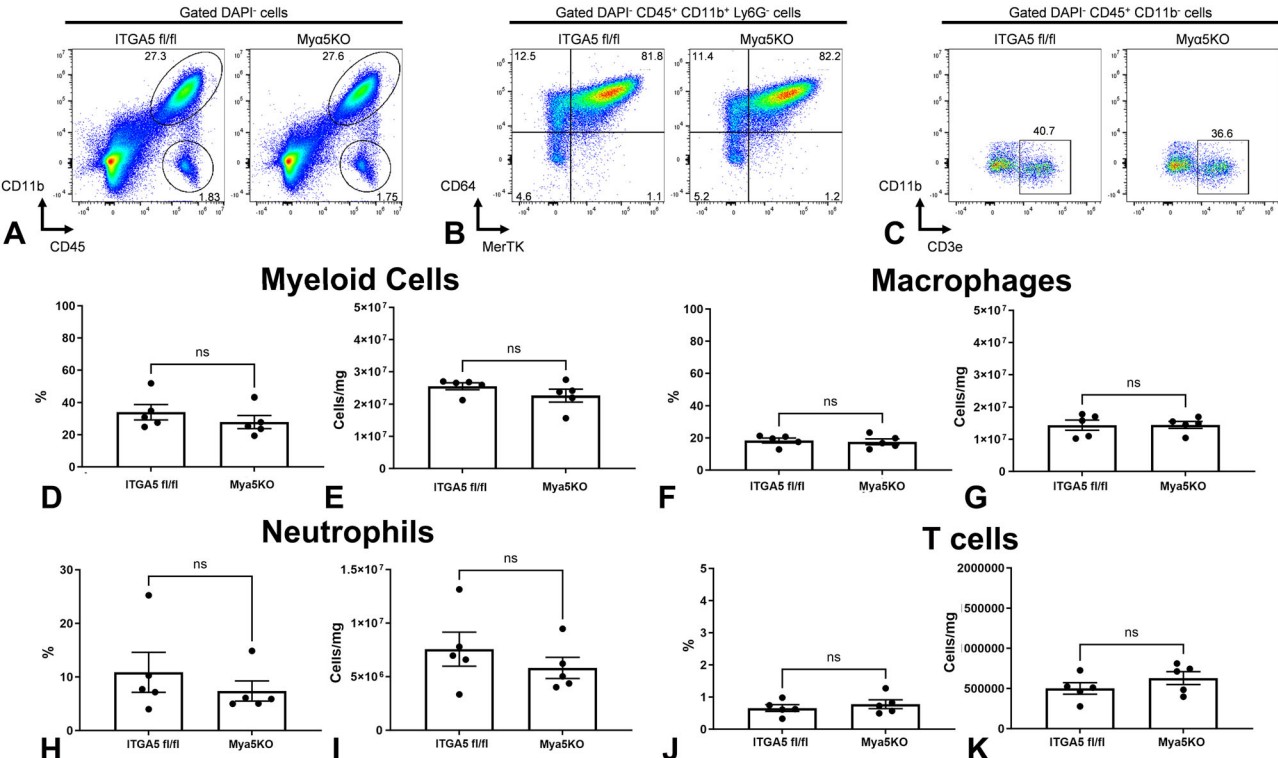

**Fig. 4 | Myeloid cell-specific ITGA5 loss does not affect infiltration of the infarcted heart with myeloid cells, macrophages, neutrophils, and T lymphocytes.** Flow cytometry was used to compare the percentage and absolute number of cells in the infarct, 7 days after myocardial infarction. The gating strategy is shown in Supplementary Fig. 7. DAPI was used to label live cells (DAPI-cells). Myeloid cells were identified as DAPI-/CD45 + /CD11b+ cells (**A**), macrophages were labeled as DAPI-/CD45 + /CD11b + /Ly6G-/CD64 + /MerTK+ cells (**B**), whereas T cells were identified as DAPI-/CD45 + /CD11b-/CD3e+ cells (**C**). Quantitative analysis showed no significant effects of myeloid cell-specific ITGA5 loss on myeloid cell (**D**, **E**), macrophage (**F**, **G**), neutrophil (**H**, **I**), and T cell (**J**, **K**) numbers in the infarct (p = NS, *n* = 5 biologically independent experiments/group). Panels **D**, **F**, **H**, and **J** show the percentage of each population in relation to all live cells (DAPI-cells), whereas panels **E**, **G**, **I**, and **L** show the absolute number of cells per mg of myocardial tissue. Data are shown as mean values +/- SEM, Statistical analysis was performed using unpaired two-tailed Student's t test. Source data are provided as a Source Data file. NS no significance.

showed that inducible macrophage-specific loss of ITGA5 attenuated infarct angiogenesis in the infarct (Fig. 6E) and in the border zone (Fig. 6F) at the 28-day timepoint, without affecting the number of microvessels in the remote non-infarcted myocardium (Fig. 6G). Assessment of vascular maturation using α-SMA immunofluorescence (Fig. 6H) showed that macrophage-specific ITGA5 loss perturbed formation of mature coated vessels in the infarct zone after 28 days of coronary occlusion (Fig. 6I), without affecting the density of mature α-SMA+ microvessels in the remote remodeling myocardium (Fig. 6J).

In order to exclude the possibility that the functional and pathologic perturbations in infarcted iMaα5KO mice are due to Cre effects, we compared dysfunction and infarct angiogenesis between tamoxifen-treated CX3CR1^CreER mice and Cre-negative littermates 28 days after infarction. No significant differences were noted in ejection fraction, LVEDV, microvessel and mature coated vessel density between groups (Supplementary Fig. 15). Thus, macrophage-specific ITGA5 loss accentuates adverse remodeling and inhibits infarct angiogenesis.

In contrast, macrophage-specific ITGA5 loss had no significant effects on post-infarction mortality (Supplementary Fig. 16A, B), and on collagen content in the infarct zone and remote remodeling segments (Supplementary Fig. 16C–E). Moreover, macrophage-specific disruption of ITGA5 signaling did not affect infiltration of the infarcted myocardium with macrophages (Supplementary Fig. 17), but transiently increased myofibroblast density at the 7-day timepoint (Supplementary Fig. 18).

Thus, our in vivo experiments using 2 different Cre drivers demonstrated that macrophage-specific ITGA5 signaling protects the infarcted heart from adverse remodeling, by stimulating infarct angiogenesis.

## Single cell transcriptomic analysis identifies reparative and pro-inflammatory clusters of infarct macrophages

Infarct macrophages exhibit remarkable heterogeneity. In order to examine the potential relation between integrin expression and the profile of macrophage subpopulations infiltrating the infarct, we performed scRNA-seq in CSF1R+ myeloid cells sorted from control CSF1R^EGFP reporter mice, or after 7 days of coronary occlusion. 8,751 CSF1R+ cells from 3 normal hearts and 27,184 cells from 3 different infarcted hearts were analyzed. The summary metrics are shown in Supplementary Table 1.

12 distinct clusters of cells were identified (Supplementary Table 2, Fig. 7A). In control hearts, the predominant cluster of resident macrophages (CMp cells) expressed high levels of *Timd4*, *Lyve1*, and *Siglec1*, consistent with the previously reported phenotypic characteristics of the self-renewing subset of resident cardiac macrophages[7]. CMp cells were abundant in control hearts, representing more than half of CSF1R+ cells; however, their number was significantly reduced in infarcts (accounting for ~1% of CSF1R+ cells in 7-day infarcts). Two additional minor clusters of resident macrophages were identified. Mp12 cells were characterized by high *Ccr2* expression and lower *Cx3cr1* levels and may represent the CCR2+ macrophage subpopulation, known to reside in normal myocardium[12,41]. Another subset of resident macrophages (Mp11) expressed high levels of *Cx3cr1, Lyve1,* and *Timd4* and virtually disappeared after infarction.

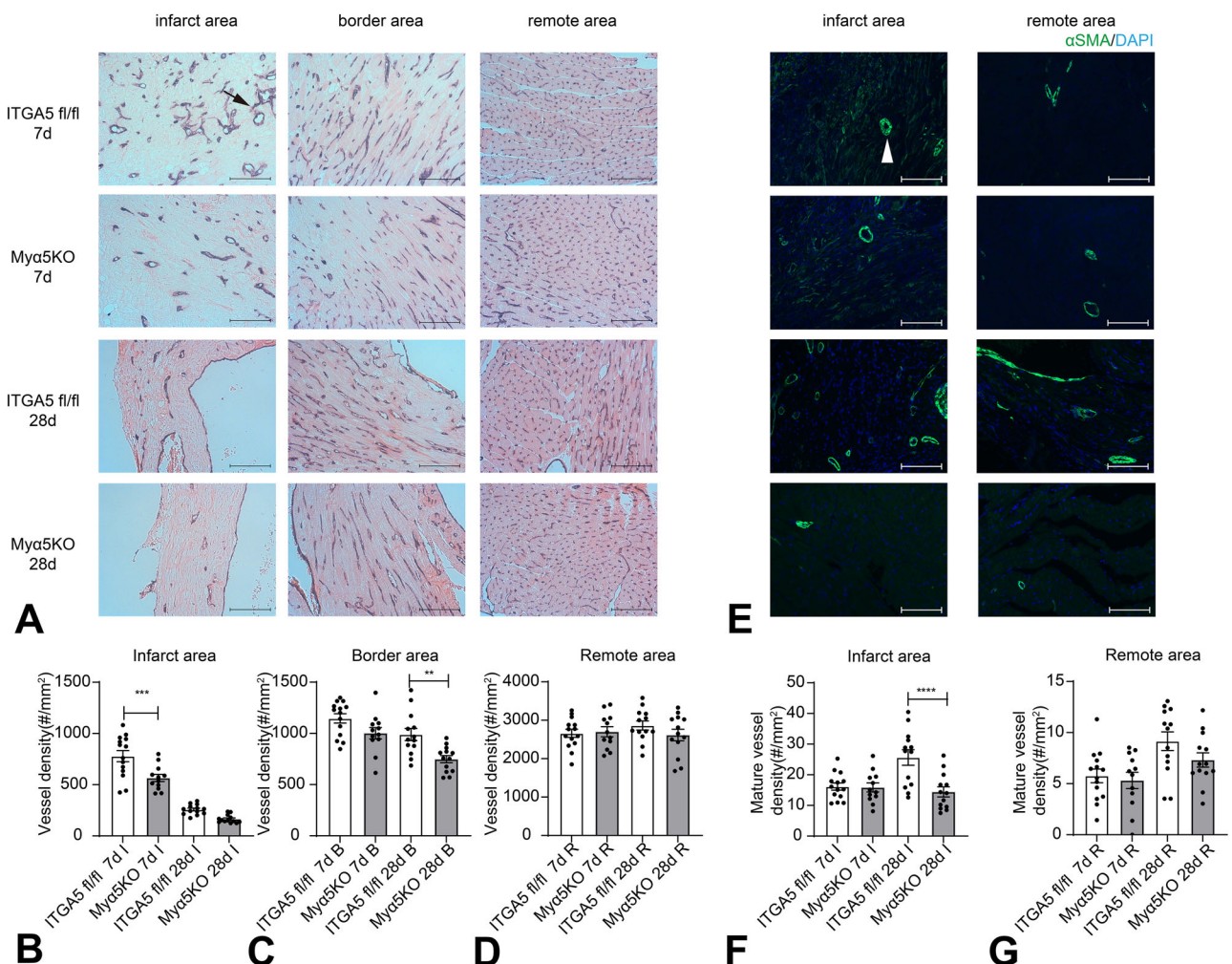

**Fig. 5 | Myeloid cell-specific ITGA5 loss reduces microvascular density and perturbs formation of coated mature vessels in the healing infarct. A** CD31 immunohistochemistry was used to label endothelial cells. Representative images show CD31 staining of the infarcted area, border zone and remote remodeling myocardium from ITGA5 fl/fl and Myα5KO mice after 7 and 28 days of coronary occlusion. The arrow identifies a typical CD31+ microvessel. Quantitative analysis showed that Myα5KO mice have significantly reduced microvascular density in the infarct zone at the 7-day timepoint (**B**), and in the border zone (**C**) after 28 days of coronary occlusion. Microvascular density in the remote remodeling myocardium was comparable between Myα5KO mice and corresponding ITGA5 fl/fl controls (**D**) (**$p < 0.01$, ***$p < 0.001$, ITGA5 fl/fl 7d: $n = 14$, Myα5KO 7d: $n = 12$, ITGA5 fl/fl 28d: $n = 13$, Myα5KO 28d: $n = 13$ biologically independent experiments). Scalebar =

100um. **E** In healing infarcts, microvessels undergo maturation acquiring a coat comprised of α-SMA-expressing mural cells (arrowheads). Representative images show α-SMA immunofluorescence staining of infarcted and remote areas from ITGA5 fl/fl and Myα5KO mice after 7 and 28 days of coronary occlusion. The arrow identifies a typical α-SMA+ vessel. Myα5KO mice had significantly reduced density of mature α-SMA+ microvessels in the infarct zone at the 28-day timepoint (**F**). The density of α-SMA+ vessels in the remote remodeling myocardium was comparable between Myα5KO and ITGA5 fl/fl mice (**G**). (****$p < 0.0001$, ITGA5 fl/fl 7d: $n = 14$, Myα5KO 7d: $n = 12$, ITGA5 fl/fl 28d: $n = 13$, Myα5KO 28d: $n = 13$ biologically independent experiments). Data are shown as mean values +/- SEM. Statistical analysis was performed using one-way ANOVA, followed by Sidak post-hoc test. Source data are provided as a Source Data file. Scalebar = 100um. I infarct area, R remote area.

Several other macrophage clusters were found predominantly in infarcted hearts. 7 days after myocardial infarction, a marked expansion of macrophages with reparative properties (RMp cluster) was noted. These cells became the predominant population of macrophages in the infarct, accounting for approximately two-thirds of the CSF1R+ cells at the 7-day timepoint (Supplementary Table 2). RMp cells had high expression of macrophage markers, such as *Adgre1* (encoding F4/80), *H2-Aa*, *Mertk*, *C1qa*, *C1qb*, *C1qc*, *Cd68* and *Cx3cr1* (Fig. 7B). The reparative phenotype of these cells was evidenced by high expression of growth factor genes (such as *Pdgfa*, *Pdgfb*, *Igf1*, *Hgf*, *Vegfb*, and *Gdf15*), and genes encoding matricellular proteins, such as *Spp1* (encoding osteopontin) and *Sparc* (Fig. 7C). RMp cells also expressed high levels of *Mmp14* (the gene encoding the matrix-bound metalloproteinase MMP-14) suggesting a migratory phenotype (Fig. 7C)[17].

Several additional macrophage clusters expanded following myocardial infarction. A pro-inflammatory macrophage cluster (IMp)

expressed high Ccr2 levels and high levels of cytokines, including *Il1b*, *Il18*, *Ccl9*, and *Vegfa* (Fig. 7B, C). A cluster of macrophages expressing *Irak3* (encoding the anti-inflammatory protein IRAK-M[42]) had high expression of *Hmox* and matrix remodeling genes, including *Thbs1*, *Ecm1*, *Vcan*, *Mmp9* and *Timp1* (Mp5 cluster, Fig. 7A–C). Consistent with the previously reported proliferative activity of macrophages in infarcted and remodeling hearts[9,43], we identified a macrophage cluster with high expression of proliferation-associated genes, such as *Plk1*, *Mybl2*, *Ccnb1*, *Ccnd1* and *Ccne1* (Proliferative Macrophages/PMp, Fig. 7A–C). A cluster with high expression of angiogenic genes, including *Vegfa*, *Vegfb*, *Angpt2*, *Angptl2*, *Angptl4* and *Angptl6* was also identified (Angiogenic macrophages/Amp, Fig. 7A–D), representing ~5% of the CSF1R+ cells in the infarct. A cell population with a monocyte gene expression profile (Mo, monocyte) was also noted, expressing high levels of *Ly6c2*, *Plac8*, and *Spn* (the gene encoding CD43, which is highly expressed in monocytes and

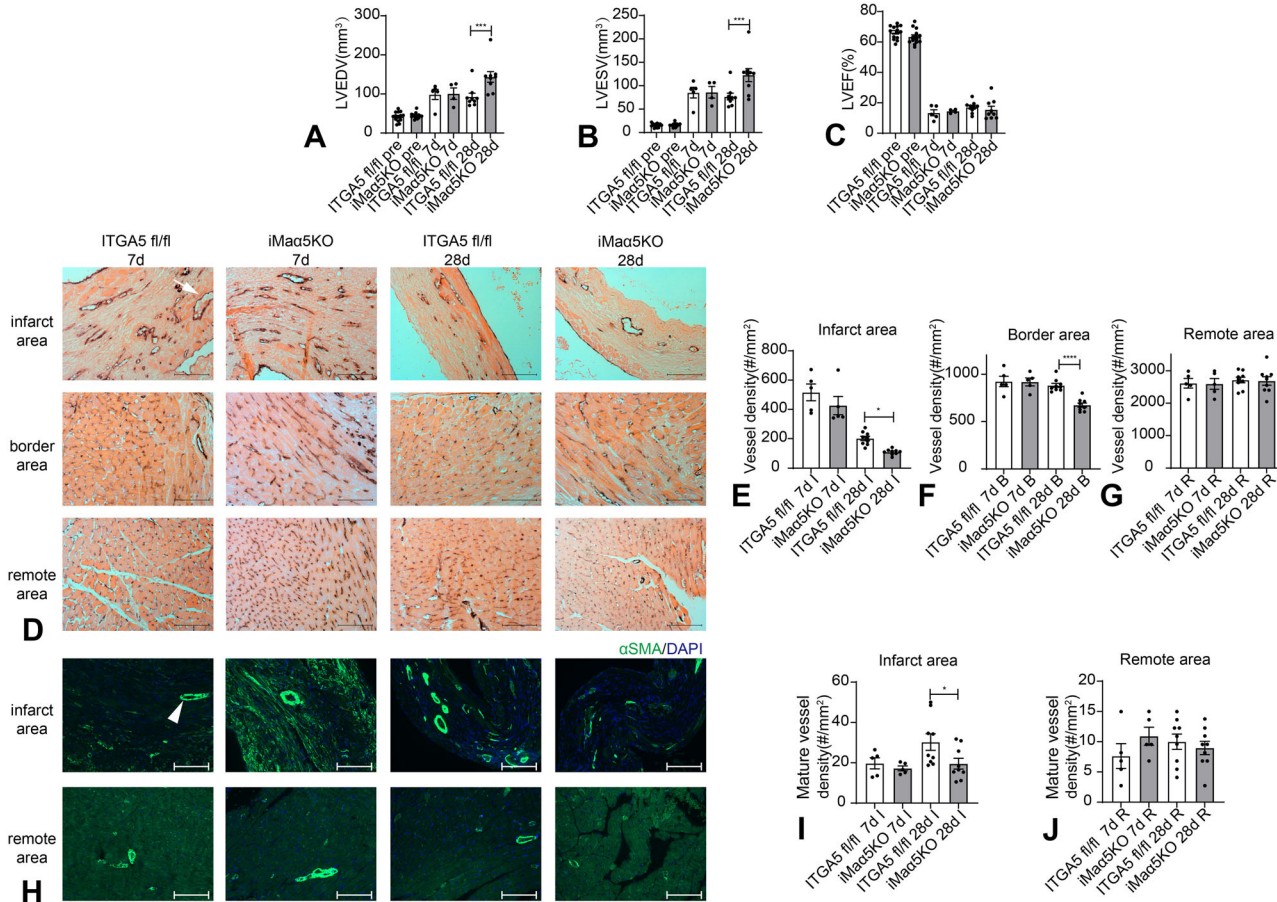

**Fig. 6 | Conditional ITGA5 deletion in CX3CR1+ macrophages accentuates dilative post-infarction remodeling, perturbing infarct angiogenesis and vascular maturation.** Echocardiographic assessment showed that inducible macrophage–specific ITGA5 knockout mice (iMaα5KO) had accentuated dilative remodeling after myocardial infarction, evidenced by increased left ventricular end-diastolic volume (LVEDV; **A**) and left ventricular end-systolic volume (LVESV; **B**) after 28 days of coronary occlusion. **C** The effects of macrophage-specific ITGA5 loss on left ventricular ejection fraction (LVEF; C) did not reach statistical significance (***$p < 0.001$, ITGA5 fl/fl pre: $n = 14$, iMaα5KO pre: $n = 14$, ITGA5 fl/fl 7d: $n = 5$, iMaα5KO 7d: $n = 4$, ITGA5 fl/fl 28d: $n = 9$, iMaα5KO 28d: $n = 9$ biologically independent experiments). **D** Representative images show CD31 immunohistochemical staining of infarcted area, border zone, and remote remodeling myocardium in ITGA5 fl/fl and iMaα5KO mice after 7 and 28 days of coronary occlusion. The arrow identifies a typical microvessel. iMaα5KO mice had significantly reduced microvascular density in the infarct zone (**E**) and in the border zone (**F**) after 28 days

of coronary occlusion. Microvascular density in the remote remodeling myocardium was comparable between iMaα5KO mice and corresponding ITGA5 fl/fl controls (**G**). **H** In healing infarcts, microvessels undergo maturation acquiring a coat comprised of α-SMA-expressing mural cells (arrowheads). Representative images of α-SMA immunofluorescence staining from infarcted area and remote area of ITGA5 fl/fl and iMaα5KO mice after 7 and 28 days of coronary occlusions are shown. The arrow identifies a typical α-SMA+ microvessel. iMaα5KO mice had significantly reduced density of mature α-SMA+ microvessels in the infarct zone at the 28-day timepoint (**I**). The density of α-SMA+ vessels in the remote remodeling myocardium was comparable between groups (**J**) (****$p < 0.0001$, *$p < 0.05$, ITGA5 fl/fl 7d: $n = 5$, iMaα5KO 7d: $n = 5$, ITGA5 fl/fl 28d: $n = 9$, iMaα5KO 28d: $n = 9$ biologically independent experiments). Statistical analysis was performed using one-way ANOVA followed by post-hoc Sidak tests. Data are shown as mean values +/- SEM. Source data are provided as a Source Data file. Scale bar = 100um. I infarct area, R remote area.

---

downregulated in macrophages[44]). These cells had relatively low expression of the macrophage genes *Adgre1, H2-Aa, Mertk, C1qa, C1qb* and *C1qc*, consistent with their monocyte identity (Fig. 7A, B). In addition, clusters with granulocyte (Gr) and dendritic cell profiles (Dc) were also identified and they expressed low levels of macrophage genes (Fig. 7A, B). Gr cells expressed high levels of the granulocyte markers *S100a9, Csf3r, Cxcr2*, and *Lcln2* and were found in low numbers in the infarcted heart, reflecting the reduced neutrophil infiltration of the infarcted heart at the timepoint examined (7 days after infarction)[35]. Although granulocytes do not express CSF1R protein, they exhibit some expression of *Csf1r* mRNA and can be labeled in the CSF1R^EGFP reporter line we used for macrophage identification[45]. Dc cells expressed high levels of the dendritic cell markers *cd209, Siglech* and *Ccr7* (Fig. 7A, B) Finally, a small population of cells with fibroblast characteristics (Fib) was identified in the control heart, expressing *Pdgfra, Pdgfrb* and *Tcf21*, all markers of fibroblast-like interstitial cells (Fig. 7A, B). These cells expressed low

levels of macrophage genes and likely represent a small population of resident cardiac fibroblasts.

## Macrophages with an angiogenic transcriptional profile exhibit high expression of *Itga5*

Integrin gene expression was assessed across the clusters of CSF1R+ cells (Fig. 8A). Supplementary Fig. 19 shows the integrin genes that were upregulated or downregulated in specific clusters of CSF1R+ cells (in comparison to all other clusters). *Itga5* expression was significantly higher in angiogenic AMp macrophages (Fig. 8B), when compared with cells from all other clusters (padj=1E-142, log2FC = 2.82). Proliferative macrophages (PMp) also had significantly higher *Itga5* levels than other clusters (log2FC = 1.27, padj=2.3E-15). Moreover, comparison of expression levels between infarcted and control hearts showed significant *Itga5* upregulation in reparative macrophages following infarction (RMp, log2(Infarct/control)=0.54, padj=4.15E-0.6). The partner chain for ITGA5, ITGB1 was broadly expressed by the majority

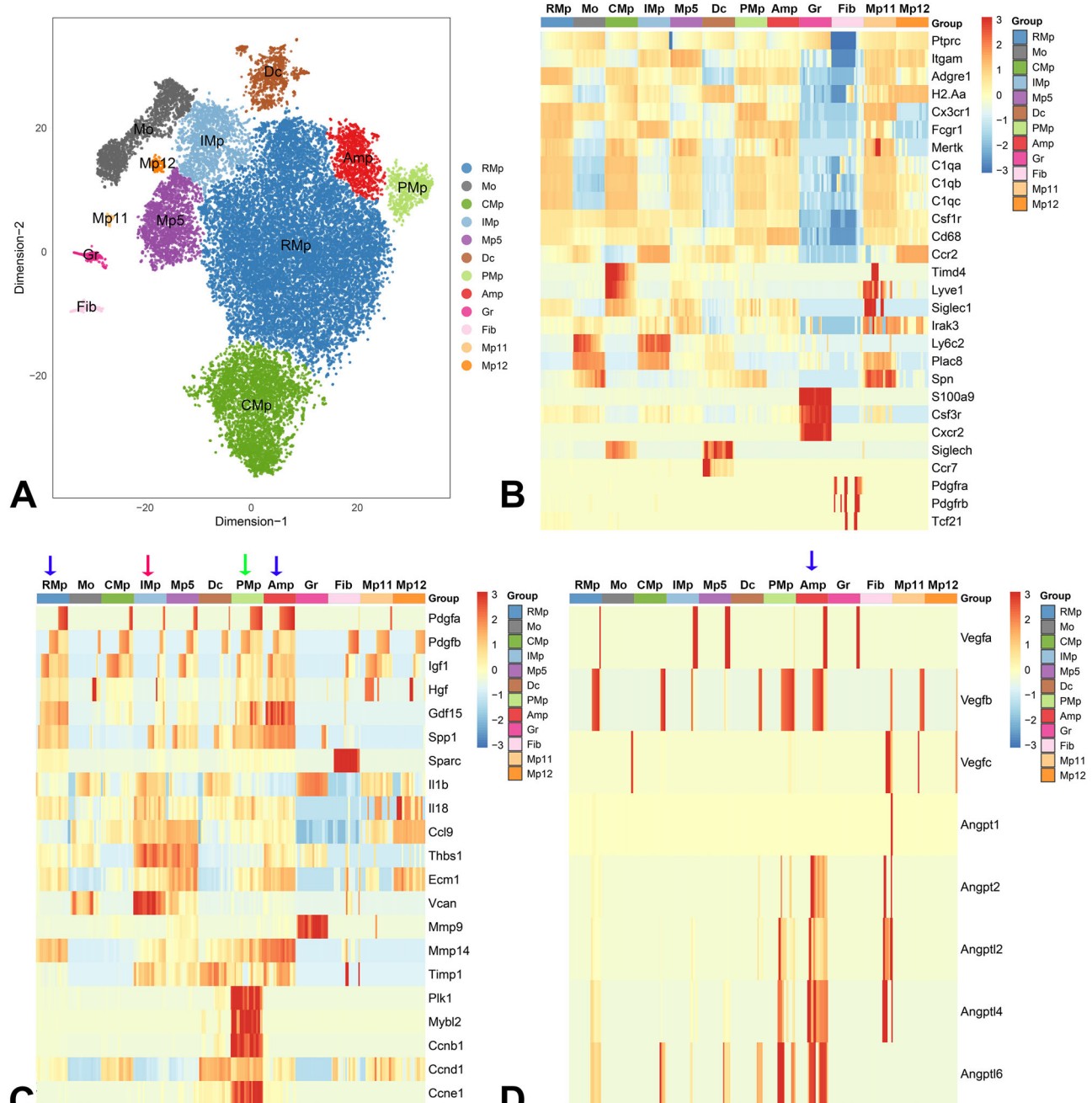

**Fig. 7 | Clusters of CSF1R+ cells in control and infarcted hearts.** ScRNA-seq was used to characterize the transcriptional profile of CSF1R+ cells harvested from control and infarcted hearts. 12 distinct clusters were identified (UMAP, **A**). **B** The heatmap illustrates relative expression of marker genes for various clusters. A cluster of reparative macrophages (RMp) was abundant in infarcted hearts after 7 days of coronary occlusion. Several additional macrophage clusters were identified (see text). In addition, cells with monocyte (Mo), granulocyte (Gr), and dendritic cell profiles (Dc) were also noted. A small cluster exhibited transcriptional characteristics of fibroblasts (Fib). **C** Relative expression of growth factors, inflammatory cytokines, matricellular genes, and proliferation-associated genes by various clusters. RMp and angiogenic macrophages (Amp) exhibited high levels of

growth factor expression (blue arrows). In contrast, inflammatory macrophages (IMp) expressed high levels of inflammatory cytokines (red arrow). A cluster with high expression of proliferation-associated genes (such as *Ccnb1*, *Ccne1*, and *Plk1*) was also noted and may represent proliferative macrophages (PMp, green arrow). **D** Expression of angiogenic growth factors across macrophage clusters. Amp cells (which had high levels of *Itga5*) exhibited high expression of angiogenic genes (blue arrow). PMp proliferative macrophages, CMp resident cardiac macrophages, Mp macrophages. Source data are provided as a Source Data file. Raw scRNA-seq data were deposited in the NCBI's Gene Expression Omnibus under accession number GSE227251.

of cells in all clusters, with significantly higher expression levels in the CMp, PMp, Amp, Mp11, and Mp12 clusters (Fig. 8C). The *Itgam* (encoding ITGAM/CD11b), *Itgb2* and *Itgb5* were also broadly expressed in cells from all macrophage and monocyte clusters (Fig. 8D–F). Other integrins were predominantly expressed by specific clusters. Inflammatory IMp cells had high expression of *ItgaL* and *Itgb7*, in comparison

to other clusters. Resident CMp exhibited higher expression of *Itga9* in comparison to other clusters. *Itgax* was predominantly expressed by dendritic cells, by the angiogenic macrophage cluster (Amp) and by the minor resident macrophage clusters (Mp11 and Mp12), whereas *Itga6* was expressed at higher levels in reparative macrophages (Supplementary Fig. 20). *Itgav* was expressed by subpopulations of cells

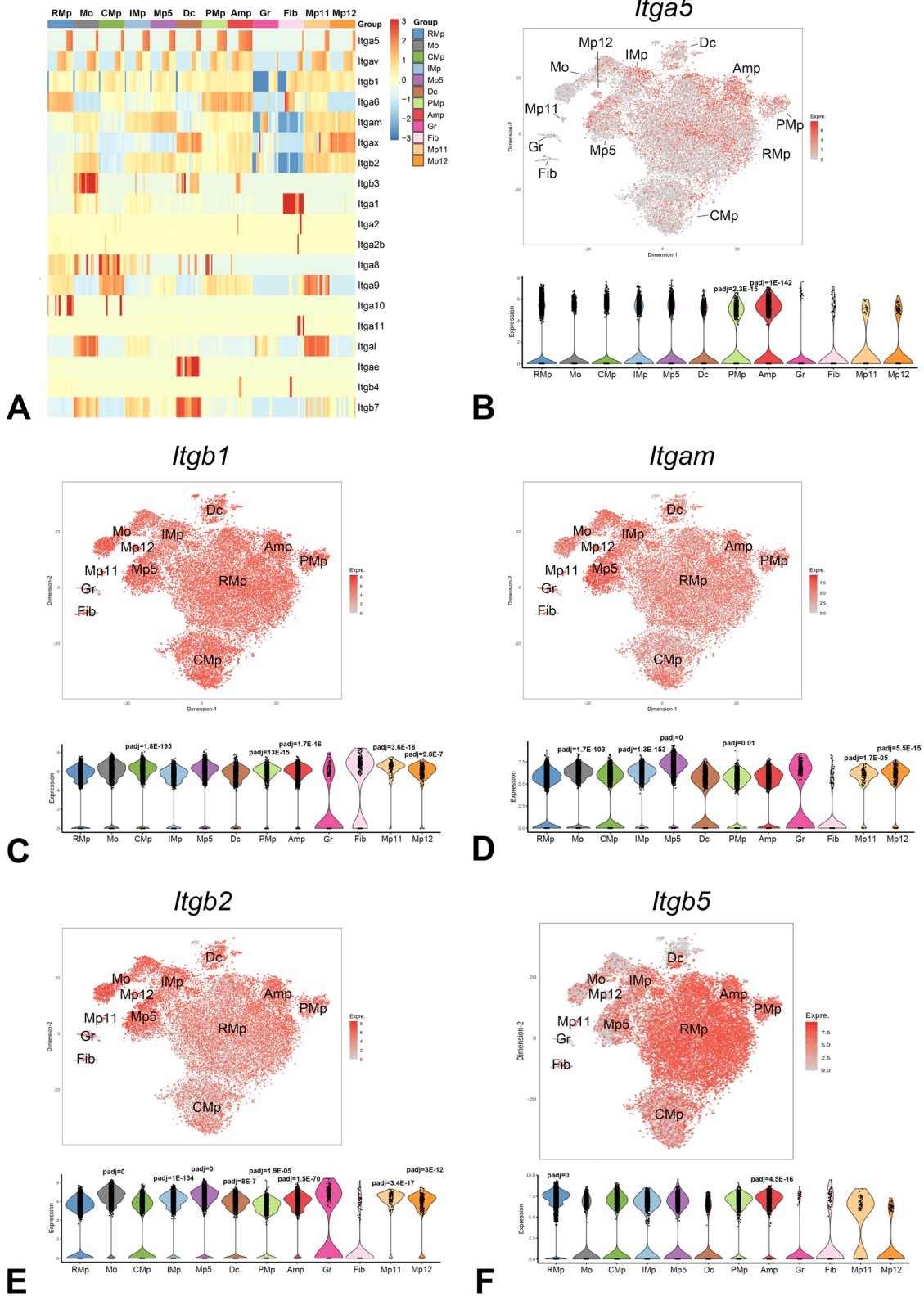

**Fig. 8 | Cluster-specific patterns of integrin expression in infarct macrophages.** **A** The heatmap shows the patterns of integrin gene expression across various clusters of CSF1R+ cells. **B** Although *Itga5* was broadly expressed by subsets of cells from all clusters, angiogenic macrophages (Amp) had higher levels of *Itga5* than other clusters (log2FC = 2.82, padj=1E-142). Amp also had higher expression of *Itgb1* (**C**), encoding the ITGA5 partner chain ITGB1 (log2FC = 0.77, padj=1.7E-16). *Itgb1* (**C**), *Itgam* (**D**), *Itgb2* (**E**), and *Itgb5* (**F**) were broadly expressed by cells from all clusters.

Differentially regulated integrin genes in specific macrophage clusters are shown in Supplementary Fig. 19. PMp proliferative macrophages; CMp resident cardiac macrophages, RMp reparative macrophages, Dc dendritic cells, IMp inflammatory macrophages, Mo monocytes, Gr granulocytes, Fib fibroblasts, Mp macrophages, padj adjusted *p*-value. Source data are provided as a Source Data file. Raw scRNA-seq data were deposited in the NCBI's Gene Expression Omnibus under accession number GSE227251.

from all clusters (Supplementary Fig. 20G). *Itgb4, Itgae, Itga2, Itga2b, Itga3, Itga7, Itga8, Itga10*, and *Itga11* expression was very low in all clusters (Supplementary Fig. 21).

## ITGA5 signaling in macrophages induces synthesis of angiogenic mediators, in vivo and in vitro

Next, we asked which macrophage-derived mediators are responsible for ITGA5-induced angiogenic activity. In order to identify the molecular mediators responsible for the angiogenic effects of ITGA5 in macrophages, we first performed an angiogenesis PCR array using mRNA from macrophages harvested from Myα5KO and ITGA5 fl/fl infarcts (at the 7-day timepoint). Supplementary Fig. 22 shows a heatmap and a volcano plot, illustrating the effects of myeloid cell-specific α5 integrin loss on the angiogenic profile of infarct macrophages. The data identified several secreted angiogenesis regulators that were modulated by ITGA5. Infarct macrophages from Myα5KO mice had significantly lower expression of the potent angiogenic growth factor *Vegfa* (Fig. 9A), and the angiogenic chemokines *Cxcl1* (Fig. 9B) and *Cxcl2* (Fig. 9C). Moreover, levels of *Csf3* (Fig. 9D) and *Tgfb2* (Fig. 9E) were also significantly decreased in Myα5KO infarct macrophages. In contrast, loss of ITGA5 did not affect synthesis of several other angiogenic mediators, such as *Vegfb, Vegfd* and *Igf1* (Fig. 9F–H), the angiopoietins, *Fgf1, Fgf2* and *Hgf* (Supplementary Fig. 22).

Although assessment of gene expression in infarct macrophages provides insights into the role of ITGA5 signaling, the observed changes may not necessarily be due to abrogation of direct actions of ITGA5 on macrophages but may also reflect secondary effects related to the accentuation of adverse remodeling. In order to identify mediators and pathways directly modulated by ITGA5 in macrophages, we performed an in vitro study, examining the in vitro effects of ITGA5 neutralization in isolated macrophages. RNA-seq analysis showed that of the angiogenesis regulators differentially regulated in vivo, *Vegfa* was also markedly downregulated in bone marrow macrophages upon ITGA5 neutralization (Fig. 9I). Reduced *Vegfa* mRNA levels upon ITGA5 blockade were associated with a reduction in VEGFA protein levels in the macrophage supernatant, assessed with ELISA (Fig. 9J). In contrast,

expression of the other angiogenesis regulators that were modulated in infarct macrophages (*Cxcl1, Cxcl2, Csf3, Tgfb2*) was not significantly affected by ITGA5 neutralization (Fig. 9K–N). Moreover, consistent with the in vivo data, expression of *Vegfb, Vegfd*, and *Igf1* by isolated macrophages was comparable in the presence or absence of ITGA5 signaling (Fig. 9O–Q). Thus, taken together the in vivo and in vitro data suggest that ITGA5 signaling promotes an angiogenic VEGFA-expressing macrophage phenotype.

## Bioinformatic analysis of RNA-seq data from in vivo and in vitro experiments identifies candidate intracellular pathways regulated by ITGA5

In order to identify integrin-mediated molecular pathways involved in the induction of angiogenic mediators in macrophages, we performed 2 different RNA-seq experiments: a) in vivo comparing the transcriptomic profile in infarct macrophages from Myα5KO and ITGA5 fl/fl infarcts and b) in vitro comparing gene expression in isolated bone marrow macrophages in the presence or absence of an anti-ITGA5 blocking antibody.

In infarct macrophages, 1506 genes were downregulated and 1177 genes were upregulated upon ITGA5 loss (Supplementary Fig. 23). Ingenuity Pathway Analysis (IPA) identified a wide range of canonical pathways, that were differentially modulated by ITGA5 loss in infarct macrophages (Supplementary Fig. 24), and upstream regulator analysis identified a long list of mediators that were predicted to be inhibited (Supplementary Table 3) or activated (Supplementary Table 4) in ITGA5KO infarct macrophages.

In vitro, ITGA5 inhibition was associated with marked alterations in the macrophage transcriptome in the presence or absence of fibronectin (without fibronectin: 1736 downregulated genes and 1873 upregulated genes; with fibronectin: 1993 downregulated genes and 2196 upregulated genes) (Supplementary Fig. 25). Ingenuity Pathway Analysis (IPA) identified numerous canonical pathways, that were differentially modulated by ITGA5 blockade (Supplementary Fig. 26), and upstream regulator analysis identified a long list of mediators that were predicted to be inhibited or activated upon ITGA5 blockade (Supplementary Tables 5–8).

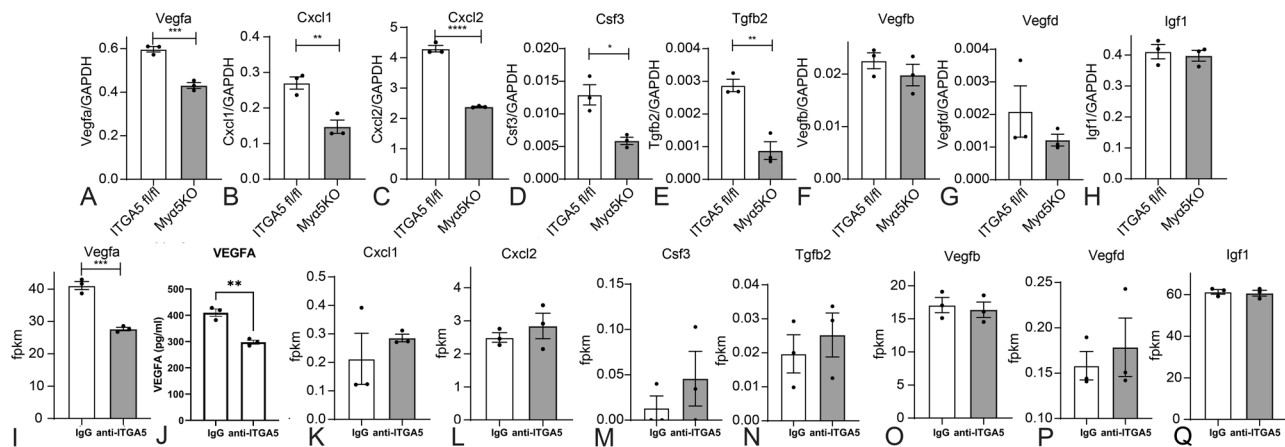

**Fig. 9 | ITGA5 signaling mediates synthesis of angiogenic mediators in infarct and in bone marrow macrophages.** In order to identify specific angiogenic mediators regulated by ITGA5 integrin, we performed an angiogenesis PCR array using RNA extracted from CD11b + /Ly6G- macrophages harvested from ITGA5 fl/fl and Myα5KO infarcts (7 days after infarction) **A–H** Of the 84 angiogenesis regulators assessed, *Vegfa* (**A**), *Cxcl1* (**B**), *Cxcl2* (**C**), *Csf3* (**D**), *Tgfb2* (**E**), were significantly decreased in Myα5KO infarct macrophages. In contrast, the expression of other critical angiogenesis mediators, including *Vegfb* (**F**), *Vegfd* (**G**), and *Igf1* (**H**) was not significantly affected by ITGA5 loss. **I–P** In order to examine whether ITGA5 directly modulates macrophage synthesis of angiogenic mediators, we studied the effects of an anti-ITGA5 neutralizing antibody (clone HMa5-1, Biolegend) in bone marrow

macrophages. RNA-seq showed that only *Vegfa* was markedly downregulated in bone marrow macrophages upon ITGA5 neutralization (**I**). Reduced *Vegfa* mRNA levels upon ITGA5 blockade were associated with a decrease in VEGFA protein levels, assessed through an ELISA (**J**). In contrast the angiogenesis regulators *Cxcl1* (**K**), *Cxcl2* (**L**), *Csf3* (**M**), *Tgfb2* (**N**), *Vegfb* (**O**), *Vegfd* (**P**) and *Igf1* (**Q**) were not significantly affected by ITGA5 blockade (****$p < 0.0001$, ***$p < 0.001$, **$p < 0.01$, *$p < 0.05$, $n = 3$ biologically independent experiments/group). Thus, the in vitro and in vivo experiments suggest that the angiogenic actions of ITGA5 in macrophages involve VEGFA synthesis. Data are shown as mean values +/- SEM Statistical analysis was performed using unpaired two-tailed Student's t test. Source data are provided as a Source Data file. Fpkm fragments per Kilobase Million, IgG immunoglobulin G.

In order to determine which intracellular pathways are directly regulated by ITGA5 in macrophages, and are also modulated in vivo by ITGA5 we identified intracellular mediators predicted to be modulated by ITGA5 in both in vivo and in vitro studies. These pathways (PI-3K/Akt, Erk5, VEGF signaling, MAPK signaling, p38 signaling, HIF1a, Nrf2) were all predicted to be inhibited upon ITGA5 blockade/loss (Supplementary Table 9) and were considered candidate downstream mechanisms responsible for the effects of ITGA5 in macrophages.

### The effects of ITGA5 on macrophage-derived *Vegfa* synthesis are independent of p38, Erk1/2 and Erk5 signaling

We next dissected the intracellular pathways involved in ITGA5-mediated acquisition of an angiogenic phenotype in macrophages. First, we examined the role of p38 MAPK, Erk1/2 MAPK, and Erk5 (all identified as candidate pathways through bioinformatic analysis of RNA-seq data), in mediating ITGA5-induced *Vegfa* synthesis in isolated bone marrow macrophages. ITGA5 neutralization markedly reduced *Vegfa* synthesis; however, incubation with the p38 MAPK inhibitor SB203580 did not affect *Vegfa* expression levels in ITGA5 antibody or IgG-treated macrophages (Supplementary Fig. 27A). Thus, ITGA5-mediated *Vegfa* induction in macrophages does not involve p38 MAPK signaling. Moreover, treatment with 2 different Erk1/2 inhibitors (U0126 and PD98059) and incubation with 2 different Erk5 inhibitors (XMD8-92 and BIX02189) did not affect *Vegfa* expression in cells with intact ITGA5 signaling (Supplementary Fig. 27B, C), suggesting that Erk1/2 and Erk5 are not implicated in mediating the pro-angiogenic effects of ITGA5 in macrophages.

### ITGA5 signaling stimulates *Vegfa* induction in macrophages through PI-3K and FAK-mediated pathways

Next, we examined the role of PI-3K/Akt signaling (one of the candidate pathways identified through RNA-seq) in mediating the ITGA5-induced angiogenic stimulation in macrophages. Treatment with the PI-3K inhibitor LY294002 markedly reduced *Vegfa* synthesis in cells with intact ITGA5 signaling (Fig. 10A). FAK signaling has been extensively implicated in integrin signaling[46] and has been involved in the stimulation of angiogenesis[47]. Thus, we examined the role of FAK in mediating ITGA5-induced *Vegfa* synthesis. Treatment with the FAK inhibitor PF-573228 markedly attenuated *Vegfa* expression in macrophages with intact ITGA5 signaling (Fig. 10B). Thus, ITGA5-induced angiogenic stimulation in macrophages is dependent on PI-3K and FAK signaling. Western blotting showed that expression and activation of PI-3K/Akt are markedly reduced upon ITGA5 inhibition (Fig. 10C–F). Moreover, ITGA5 blockade also had a significant effect on FAK activation without affecting the levels of FAK (Fig. 10C, G–I). In order to examine whether PI3K activation in ITGA5-stimulated macrophages is dependent on FAK, we examined the effects of the FAK inhibitor on PI-3K/Akt expression and activation. The FAK inhibitor had no significant effects on PI-3K/Akt expression and activation in the presence or absence of ITGA5 signaling (Fig. 10J–M). Taken together (Supplementary Fig. 27 and Fig. 10), our findings demonstrate that ITGA5-mediated angiogenic conversion of macrophages does not involve p38 MAPK, Erk MAPK and Erk5, but requires activation of independent PI-3K and FAK cascades.

### ITGA5 mediates the expression of the angiogenic genes *Emilin2* and *Ecm1* in vivo and in vitro

Next, we identified additional secreted mediators that may be induced by ITGA5 in macrophages, and may contribute to angiogenesis and scar maturation. RNA-seq comparing the transcriptomic profile of myeloid cells harvested from Mya5KO and ITGA5 fl/fl infarcts showed that a significant number of proinflammatory genes (including *Il1a*, *Il1b*, *Il6*, *Cxcl3* and *Cxcl5*), anti-inflammatory genes (such as *Il10* and *Tgfb2*), and angiogenic mediators (such as *Nrg1*, *Hbegf*, and *Areg*) were downregulated in infarct macrophages upon ITGA5 disruption

(Supplementary Table 10). Moreover, expression of *Lox* and *Tgm2* (encoding the matrix-crosslinking enzymes lysyl-oxidase and transglutaminase-2 respectively) were also markedly reduced in ITGA5 KO infarct macrophages. However, these genes were not consistently modulated by ITGA5 inhibition in vitro, suggesting that their down-regulation in vivo may reflect secondary consequences of myeloid cell-specific ITGA5 loss. In contrast, expression of *Emilin2*, encoding the angiogenic matricellular protein EMILIN2 (elastin microfibril interface located protein 2)[48], and *Ecm1* (which encodes the matrix protein Extracellular matrix protein 1 (ECM1), a potent stimulus of endothelial cell proliferation[49]) was consistently suppressed upon ITGA5 disruption in vivo and in vitro. ScRNA-seq demonstrated high expression of *Ecm1* and *Emilin2* in macrophage clusters (Supplementary Fig. 28). The angiogenic macrophage cluster (Amp) expressed significantly higher levels of both *Ecm1* and *Emilin2* than other clusters, supporting the notion that ITGA5 may be involved in upregulation of these angiogenic genes. *Ecm1* was also highly expressed in RMp, Mp5, and Mp12 clusters, whereas monocytes, granulocytes, and the macrophage clusters Imp, CMp, Mp11, and Mp5 also expressed higher levels of *Emilin2* than other clusters (Supplementary Fig. 28). Thus, ECM1 and EMILIN2 may be additional ITGA5-dependent angiogenic mediators secreted by infarct macrophages.

## Discussion

We report for the first time, a crucial role for macrophage-specific ITGA5 signaling in tissue repair. Using 2 distinct macrophage-specific loss-of-function models, we demonstrated that ITGA5 activation promotes an angiogenic phenotype in infarct macrophages, stimulating neovessel formation in the infarct and in the border zone, and protecting the infarcted heart from adverse remodeling. ScRNA-seq identified a subpopulation of infarct macrophages expressing angiogenic genes and exhibiting high levels of *Itga5*. Using whole transcriptome sequencing approaches in vitro and in vivo, followed by bioinformatic analysis, we dissected the downstream angiogenic molecular signals activated by ITGA5 in macrophages. Neutralization experiments demonstrated that independent PI-3K and FAK pathways mediate the angiogenic effects of ITGA5 in macrophages, by stimulating VEGFA synthesis. Our findings highlight the critical role of matrix-driven integrin cascades in the activation of a reparative macrophage phenotype in healing tissues.

The adult mouse heart contains a population of resident macrophages, derived predominantly from the yolk sac and fetal monocyte progenitors[50,51]. Following myocardial infarction, massive and sudden necrosis of cardiomyocytes in the ischemic zone stimulates innate immune pathways that activate chemokine synthesis in resident cardiac macrophages and endothelial cells. This local inflammatory reaction triggers a marked expansion of the cardiac macrophage population, mediated initially through the recruitment of circulating monocytes[52]. Induction of CSF-1 stimulates a proliferative program, further increasing the number of macrophages in the healing infarct[9,43]. In response to the changes in the infarct environment, macrophages undergo dynamic phenotypic transitions. Activation of pro-inflammatory and phagocytic macrophages during the early stages of infarct healing is followed by anti-inflammatory transition and by the emergence of angiogenic and fibrogenic subpopulations during the proliferative phase of cardiac repair. Soluble secreted mediators, including neurohumoral factors, inflammatory cytokines, and growth factors have been extensively implicated in the regulation of the functional properties and phenotypic profile of infarct macrophages[13,53–55]. Direct effects of hypoxia on HIF-1 activation may also regulate the inflammatory activity of macrophages in the infarct[56].

In addition to the effects of hypoxia and soluble mediators, infarct macrophages also respond to changes in the ECM network[57]. The infarcted heart is enriched with a wide range of matricellular macromolecules and specialized matrix proteins. These components of the

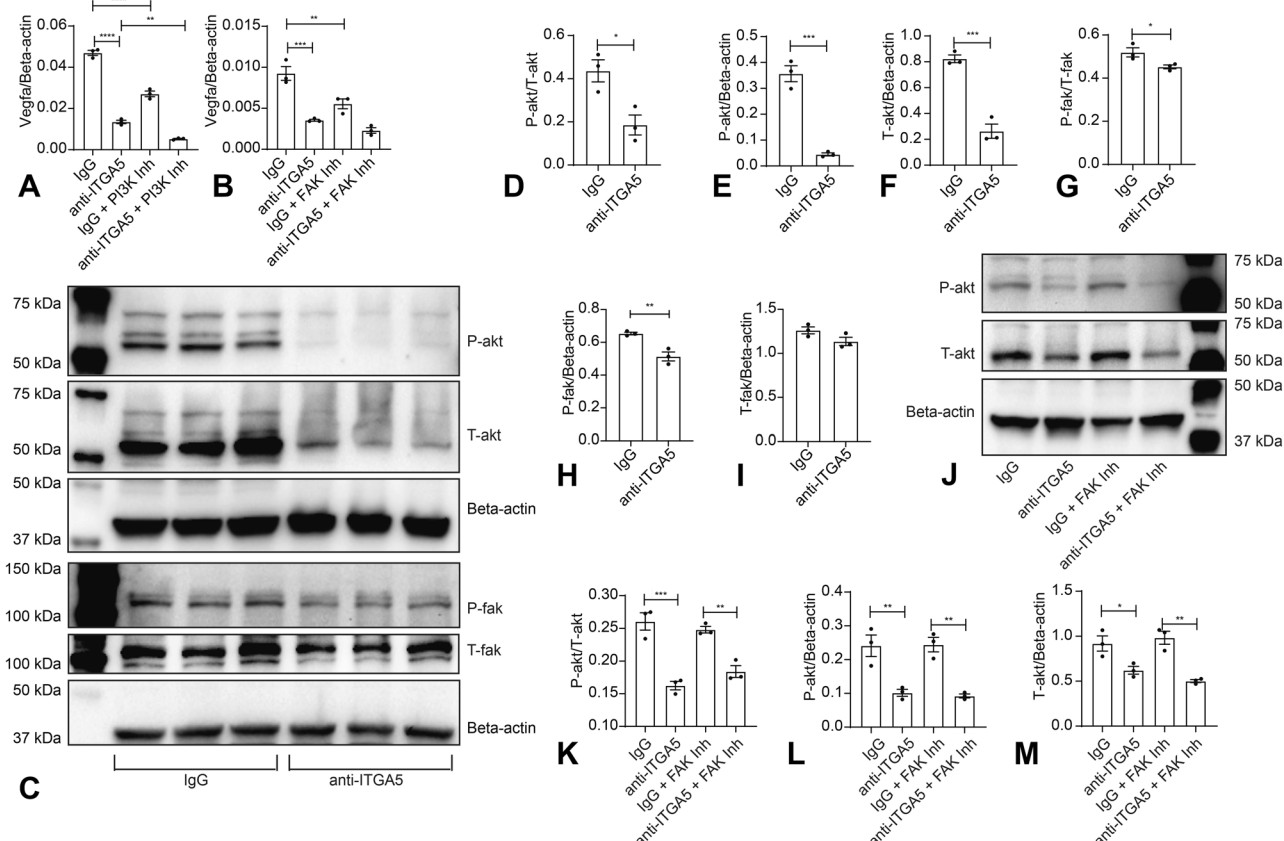

**Fig. 10 | ITGA5-mediated angiogenic stimulation of macrophages is dependent on PI-3K/Akt and FAK signaling. A, B** We used treatment of bone marrow macrophages (BMMs) with pharmacologic inhibitors to examine the role of Phosphoinositide-3 Kinase (PI-3K)/Akt (**A**) and Focal Adhesion Kinase (FAK) (**B**) signaling in ITGA5-mediated *Vegfa* synthesis. ITGA5 blockade markedly reduced *Vegfa* expression in BMMs (**A**, **B**). Treatment with the PI-3K/Akt inhibitor LY294002 (20 μM, **A**), or with the FAK inhibitor PF-573228 (5 μM, **B**) markedly attenuated *Vegfa* synthesis in macrophages with intact ITGA5 signaling (IgG group), but had no significant effects on *Vegfa* levels upon ITGA5 blockade (**A**, **B**). **C–I** Western blotting experiments (**C**) were used to examine the role of ITGA5 signaling on PI-3K/Akt and FAK activation in fibronectin-treated BMMs. PI-3K/Akt expression and activation (**C**, **D–F**) were markedly attenuated upon ITGA5 blockade. Moreover, ITGA5 blockade modestly but significantly attenuated FAK activation, without significantly affecting total FAK levels (**C**, **G–I**). In order to examine whether PI-3K/Akt activation in ITGA5-stimulated macrophages is dependent on FAK, the effects of the FAK inhibitor PF-573228 on PI-3K/Akt expression and activation were examined. Western blotting showed the FAK inhibitor had no significant effects on PI-3K/Akt expression and activation, in the presence or absence of ITGA5 blockade (**J–M**) (****$p < 0.0001$, ***$p < 0.001$, **$p < 0.01$, *$p < 0.05$, $n = 3$ biologically independent experiments/group). Thus, ITGA5-mediated angiogenic activation of macrophages is dependent on independent FAK and PI-3K pathways. Data are shown as mean values +/- SEM. Statistical analysis was performed using one-way ANOVA, followed by Sidak post-hoc test. Source data are provided as a Source Data file. Inh inhibitor.

matrix do not play a major structural role but serve as modulators of cytokine and protease activity, and as transducers of signaling cascades that link the cells to their extracellular milieu[58]. During the inflammatory and proliferative phases of healing, sequential deposition of plasma and cellular fibronectin in the infarct zone contributes to the formation of a provisional matrix network, which regulates inflammation and repair[59]. Matricellular proteins, such as tenascin-C[60], thrombospondins[61], osteopontin[62] and periostin[63] are also induced in the healing infarct and modulate phenotype and function of immune cells, vascular cells and fibroblasts[64].

Much like other cell types, macrophages sense changes in the ECM environment through integrins. In the healing infarct, matrix fragments generated by activated proteases, provisional matrix components (such as fibronectin and fibrin), and matricellular proteins transduce integrin-dependent signals on macrophages, modulating their inflammatory and reparative properties. Our systematic study of integrin gene expression by macrophages (Supplementary Fig. 1) suggests that the responsiveness of macrophages to the provisional ECM may be accentuated by increased expression of several members of the integrin family (including *Itga5, Itgav, Itga6, ItgaM, Itgax, Itgb2, and Itgb5*). ITGA5 is one of the highest-expressed integrin chains in

infarct macrophages. Flow cytometry and immunofluorescence showed that ITGA5 protein expression is markedly upregulated in infarct macrophages during the proliferative phase of cardiac repair (Fig. 1). ITGA5 associates with the ITGB1 subunit, forming a receptor which transduces fibronectin-mediated signals through the short sequence Arg-Gly-Asp (RGD)[65]. In addition to fibronectin, the best-characterized α5β1 ligand, several other components of the ECM, including fibrin[66], fibulin-5[67], vitronectin[68] and the matricellular proteins osteopontin[28], SPARC[69], and periostin[70] have been suggested to interact with the ITGA5/ITGB1 complex.

Published evidence suggests an important role for ITGA5 in the regulation of inflammatory responses. In vascular remodeling, ITGA5 has been implicated in transducing fibronectin-mediated pro-inflammatory signals[71]. Moreover, in a model of atherosclerosis α5β1 was found to mediate inflammation[30]. However, these pro-inflammatory actions were attributed to integrin-dependent effects on the endothelium that trigger NF-κB activation and subsequent Vascular Cell Adhesion Molecule (VCAM)-1 upregulation, thus promoting macrophage recruitment[30]. Direct evidence demonstrating a role for macrophage-specific ITGA5 signaling in the regulation of inflammation is lacking. Associative data from a model of sepsis have suggested that

expression of ITGA5 may correlate with a pro-inflammatory macrophage phenotype[72]. Moreover, in vitro studies have demonstrated that in peripheral blood mononuclear cells and in HL-60 cells, α5β1 signaling was implicated in cytokine-mediated regulation of matrix metalloproteinase (MMP) production[29,73]. Our experiments did not show any significant effects of myeloid cell-specific ITGA5 loss on recruitment of macrophages and lymphocytes in the healing infarct (Fig. 4). Although in vivo, infarct macrophages harvested from myeloid cell-specific ITGA5 KOs had lower expression of both pro- and anti-inflammatory genes, in vitro ITGA5 inhibition did not modulate the inflammatory profile of cultured macrophages, in the presence or absence of fibronectin (Supplementary Table 10). Thus, the in vivo consequences of ITGA5 loss on the inflammatory phenotype of macrophages may reflect secondary effects, rather than direct actions of ITGA5 signaling on macrophages.

Our experiments, using both constitutive and inducible macrophage-specific loss-of-function models demonstrated a critical role for ITGA5 in stimulating angiogenic macrophage activation. The angiogenic actions of macrophage ITGA5 protected the infarcted heart from adverse remodeling, highlighting the central role of macrophage-driven formation of reparative neovessels in the repair of the infarcted heart. α5β1 integrin has been previously suggested to stimulate angiogenesis and lymphangiogenesis; however, these actions have been attributed to direct effects on endothelial cells[74–76]. Our findings demonstrate for the first time that macrophage ITGA5 plays a crucial role in mediating angiogenic effects in healing tissues.

Which macrophage-derived molecular signals trigger ITGA5-induced angiogenesis in the healing infarct? A systematic study of the levels of angiogenesis regulators in vitro and in vivo identified VEGFA, as a central mediator involved in ITGA5-stimulated angiogenic conversion of macrophages (Fig. 9). This finding is consistent with published observations, suggesting that myeloid cells are a major source of VEGFA in the infarcted myocardium, serving as key cellular effectors of infarct angiogenesis[20]. Two additional ECM-related angiogenic mediators, ECM1[49] and EMILIN2[48] were also found to be suppressed upon ITGA5 disruption in both in vivo and in vitro experiments and may contribute to the angiogenic actions of ITGA5 signaling.

To dissect the integrin-mediated cascades responsible for ITGA5-induced angiogenic conversion, we used unbiased whole genome transcriptomic analysis in vitro and in vivo. The candidate pathways identified through bioinformatic analysis were then tested using pharmacologic inhibition strategies. We demonstrated that ITGA5-stimulated angiogenic macrophage activation is independent of p38 MAPK, Erk1/2, and Erk5 activation (Supplementary Fig. 27) but involves FAK and PI-3K signaling pathways (Fig. 10). A large body of evidence shows that, upon activation, integrins undergo conformational changes that induce cytoplasmic protein binding, and downstream signaling through cytosolic kinases[77]. In macrophages ITGA5/ITGB1-induced activation of FAK has been implicated in regulation of cell motility[78]. Our findings demonstrate that integrin-stimulated FAK activation in macrophages also has prominent effects on their functional profile, stimulating an angiogenic phenotype.

The data also suggest a potential link between integrin expression and macrophage heterogeneity in the healing infarct. Our scRNA-seq experiments identified a macrophage subpopulation with an angiogenic transcriptional profile that expands following infarction and expresses high levels of *Itga5* (Fig. 8, Supplementary Fig. 19). This cluster of angiogenic macrophages expressed high levels of *Vegfa*, *Ecm1* and *Emilin2*, the 3 angiogenic genes found to be induced through ITGA5-dependent pathways in vitro and in vivo. Thus, stimulation of angiogenesis through macrophage-specific ITGA5 signaling may reflect integrin activation in a specific subpopulation of macrophages with angiogenic potential. Additional functions of macrophages may be regulated by other integrins. Thus, spatial and temporal changes in

growth factor and matrix content may induce cluster-specific patterns of integrin upregulation and activation, thus playing a central role in the functional and phenotypic diversification of macrophages in the infarcted heart.

We demonstrate for the first time a crucial role for macrophage-specific activation of ITGA5 in cardiac repair, associated with stimulation of an angiogenic macrophage phenotype. Our findings suggest an integrin-mediated macrophage-dependent mechanism that stimulates angiogenesis in tissue repair. Considering their critical role in reparative cellular responses, and their localization on the cell surface, which facilitates pharmacologic interventions, integrins are attractive therapeutic targets in a wide range of diseases. Macrophage-specific ITGA5 activation through the administration of activating peptides may be a promising strategy to enhance angiogenesis in myocardial infarction and in ischemic cardiomyopathy.

## Methods

### Generation of myeloid cell-specific and macrophage-specific ITGA5 KO mice and breeding of macrophage reporter mice

All animal studies were approved by the Institutional Animal Care and Use Committee at Albert Einstein College of Medicine. All animals were housed in the Albert Einstein College of Medicine animal institute (Ullman Building), a disease-free rodent facility. Animal care was in strict accordance with AAALAC and NIH guidelines. In order to study the role of α5 integrin (ITGA5) in macrophages in vivo, we generated 2 lines of macrophage-specific ITGA5 knockout mice. First, we used the lysozyme-M Cre driver (Jackson #004781)[79], an effective and widely used tool for gene deletion in myeloid cells[13] to generate myeloid cell-specific ITGA5 KO mice (Myα5KO). LyzM-Cre mice were bred with ITGA5 fl/fl (Jackson #032299) mice[80] to generate LyzM-Cre; ITGA5 fl/fl mice (Myα5KO) and ITGA5 fl/fl littermates. Next, we used the CX3CR1-CreER driver (Jackson #020940)[81] to generate mice with inducible loss of ITGA5 in CX3CR1+ macrophages. CX3CR1-CreER mice were bred with ITGA5 fl/fl animals to generate tamoxifen-inducible CX3CR1-CreER; ITGA5 fl/fl mice (inducible macrophage-specific ITGA5 knockouts, iMaα5KO) and corresponding ITGA5 fl/fl control littermates. Tamoxifen (Sigma, T5648) was dissolved in 5% ethanol and 95% corn oil to make a 20 mg/mL stock solution. Tamoxifen was administered intraperitoneally every 24 h for 5 consecutive days (100 mg/kg/day) in iMaα5KO mice and corresponding ITGA5 fl/fl animals. Mice were 10 weeks old during initiation of tamoxifen treatment. In order to study the time course of ITGA5 activation in macrophages we used the "MacGreen" CSF1R-EGFP reporter mice[82]. These transgenic mice (Jackson labs #018549) express the enhanced green fluorescent protein (EGFP), under the control of the mouse *Csf1r*. 81 ITGA5 fl/fl mice, 46 Myα5KO mice, 33 iMaα5KO mice, and 29 CSF1R-EGFP reporter mice (both male and female) underwent non-reperfused myocardial infarction protocols.

Mice were maintained in a controlled environment with a 12:12 light-dark cycle, room temperature of 21–23 °C and humidity of 55% ± 10 and had ad libitum access to dry laboratory food and water. All mice were transferred to clean cages once weekly. Mice were fed with Laboratory Rodent Diet 5001 (LabDiet Catalogue Number #00006505). Food and clean water were available ad libitum for the mice at all times. Feed and water were available on a free-choice basis in wire feeders above the floor of the cage.

### Mouse model of non-reperfused myocardial infarction

Animal studies were approved by the Institutional Animal Care and Use Committee at Albert Einstein College of Medicine and conform with the Guide for the Care and Use of Laboratory Animals published by the National Institutes of Health. A model of non-reperfused myocardial infarction was induced by permanent ligation of the left anterior descending coronary artery, as previously described by our group[13]. ITGA5 fl/fl mice, Myα5KO mice, iMaα5KO mice, and CSF1R[EGFP] mice were used

for the in vivo study. 3-4-month-old male and female mice were anesthetized using inhaled isoflurane (3% for induction, 2% for maintenance). Intraoperatively, heart rate, respiratory rate, and electrocardiogram were continuously monitored, and the depth of anesthesia was assessed using the toe pinch method. The left anterior descending coronary artery was occluded for 24 h, 3 days, 7 days, or 28 days. To assess cardiac function and remodeling following myocardial infarction, animals underwent echocardiographic analysis at baseline and after 7 and 28 days of myocardial infarction. At the end of the experiment, euthanasia was performed using 2% inhaled isoflurane followed by cervical dislocation. Early euthanasia was performed with the following criteria, indicating suffering of the animal: weight loss>20%, vocalization, dehiscent wound, hypothermia, signs of heart failure (cyanosis, dyspnea, tachypnea), lack of movement, hunched back, ruffled coat, lack of food or water ingestion. At the end of the experiment, the animals were sacrificed, and the hearts and lungs were excised and weighed. Hearts were fixed in zinc-formalin (Z-fix, Anatech, Battle Creek, MI) and embedded in paraffin for histological studies (7 days group: ITGA5 fl/fl $n = 19$, Myα5KO $n = 12$, iMaα5KO $n = 5$; 28 days group: ITGA5 fl/fl $n = 22$, Myα5KO $n = 13$, iMaα5KO $n = 9$). CSF1R[EGFP] mice were used for histological studies (control group: $n = 6$; 24 h group: $n = 6$; 3 days group: $n = 5$; 7 days group: $n = 6$, 28 days group: $n = 6$). Additional experiments were performed to exclude the effects of Cre on the echocardiographic endpoints and on angiogenesis. LyzM Cre mice and corresponding Cre-negative controls underwent 7-day coronary occlusion protocols. Echocardiography was performed at baseline and prior to sacrifice; at the end of the experiments, the hearts were used for histological processing ($n = 6$-$11$/ group). Moreover, we compared function and infarct angiogenesis between tamoxifen-injected CX3CR1[CReER] animals and corresponding Cre-negative controls undergoing 28 days of coronary occlusion ($n = 10$-$11$/group). The surgical protocols were performed by an investigator (R.L.) blinded to the genotype of the animals.

### Echocardiography
Echocardiographic studies were performed at baseline and after 7 and 28 days of permanent coronary occlusion to assess cardiac function and remodeling following myocardial infarction using the Vevo 2100 system (VisualSonics. Toronto ON). Baseline echocardiographic parameters were compared between male and female ITGA5 fl/fl and Myα5KO (or iMaα5KO) mice at 3-4 months of age (ITGA5 fl/fl $n = 41$, Myα5KO $n = 25$, iMaα5KO $n = 14$). Because male mice had very high mortality in both ITGA5 fl/fl and iMaα5KO groups after tamoxifen injection, a comparison of echocardiographic parameters in iMaα5KO mice following coronary occlusion was performed in female mice (7-day group: ITGA5 fl/fl $n = 19$, Myα5KO $n = 12$, iMaα5KO $n = 5$; 28-day group: ITGA5 fl/fl $n = 22$, Myα5KO $n = 13$, iMaα5KO $n = 9$). Long-axis B-mode was used to assess the geometric characteristics of the LV after myocardial infarction. Short-axis and long-axis M-mode were used for the measurement of systolic and diastolic ventricular and wall diameters. The left ventricular end-diastolic diameter (LVEDD), left ventricular end-systolic diameter (LVESD), left ventricular end-systolic volume (LVESV), and left ventricular end-diastolic volume (LVEDV) were measured as indicators of dilative remodeling. LV mass was measured as an indicator of hypertrophic remodeling. The echocardiographic off-line analysis was performed by an investigator (R.L.) blinded to the study groups.

### Quantitative morphometry, scar size measurements and Artificial Intelligence (AI)-based assessment of collagen content
For histopathological analysis, murine hearts were fixed in zinc-formalin (Z-fix; Anatech, Battle Creek, MI), and embedded in paraffin. Infarcted hearts were sectioned from base to apex at 250 μm intervals, thus reconstructing the whole heart, as previously described[33]. 20 sections (5 μm thick) were cut at each level. In order to assess scar

size, the first section at each partition was stained with hematoxylin and eosin (H&E) (HE staining: 7 days group: ITGA5 fl/fl $n = 19$, Myα5KO $n = 12$; 28-day group: ITGA5 fl/fl $n = 22$, Myα5KO $n = 13$). Morphometric parameters were quantitatively assessed using Zen 2.6 Pro software (Carl Zeiss). The infarcted and non-infarcted areas were measured at each level and the volume of the infarct and of the non-infarcted remodeling myocardium at each level was calculated as: Infarct volume = infarct area x 350 μm (250 μm + 20 sections x 5 μm = 350 μm) and non-infarcted myocardium volume =non-infarcted area x 350 μm. The total volume of the infarcted and non-infarcted myocardium was calculated as the sum of the volumes of each partition. Scar size was measured by dividing the volume of the infarct by the total volume of the left ventricle (infarct volume + volume of non-infarcted myocardium) and was expressed as a percentage. Picrosirius red staining was used to label the collagen-based scar, and collagen content was quantitatively assessed in the infarct and remote myocardium using Intellesis and Image Analysis modules of Zen 2.6 Pro software. An artificial intelligence (AI) model was trained to segment and measure the surface areas of collagen fibers. At least 10 different fields from two non-adjacent stained sections per mouse at 2 different levels (mid-myocardial, and apical levels) were scanned and analyzed per heart sample (Picrosirius red: 7 days group: ITGA5 fl/fl $n = 19$, Myα5KO $n = 12$, iMaα5KO $n = 5$; 28-day group: ITGA5 fl/fl $n = 22$, Myα5KO $n = 13$, iMaα5KO $n = 9$).

### Immunohistochemistry and assessment of microvascular density
For histopathological analysis, murine hearts were fixed in 10% aqueous buffered zinc-formalin (Z-fix, Anatech, 170) and embedded in paraffin. Hearts were sectioned from base to apex at 250-μm intervals, thus reconstructing the whole heart, as previously described[33]. Ten sections (5 μm thick) were cut at each interval, corresponding to an additional 50-μm segment. Antibodies used for immunohistochemical studies are listed in Supplementary Table 11. Microvessels in the infarct, border zone, and remote remodeling myocardium after 7 and 28 days of coronary occlusion were identified by using immunohistochemistry with an anti-CD31 antibody (Cell Signaling Technology, #77699). Staining was performed with the use of a biotinylated secondary antibody against rabbit (Vectorlabs, 1:500 BA-1000) followed up by a streptavidin peroxidase-based technique with the Vectastain ELITE ABC-HRP kit, Peroxidase (Vector Laboratories Inc., PK6100). In brief, sections were deparaffinized in Xylene (Sigma-Aldrich, 534056), rehydrated in consecutive graded washes of ethanol and followed up with heat-mediated antigen retrieval with citrate buffer, pH 6.0 (Sigma-Aldrich, C9999) for 30 min. Sections were pretreated with a solution of 3% hydrogen peroxide (Sigma-Aldrich, H1009) to inhibit endogenous peroxidase activity and incubated with 2% bovine serum albumin (Sigma-Aldrich, A8531) to block nonspecific protein binding. Sections were incubated overnight at 4 °C with anti-CD31 antibody prepared in 1X Tris Buffered Saline (TBS, Boston Bioproducts, BM-300X). Sections were then washed three times (5 mins each) with 1X TBS and incubated with biotinylated anti-rabbit for 1 h at room temperature. Sections were washed three times with TBS (5 mins each) to proceed with secondary antibody detection by the streptavidin-peroxidase-based technique Vectastain ELITE rabbit kit, following manufacturer protocol. Peroxidase activity was detected with diaminobenzidine with nickel. Slides were counterstained with eosin (Sigma-Aldrich, HT1102128) and mounted with Cytoseal medium (Thermofisher, 23-244257). At least 10 different fields from two non-adjacent stained sections at 2 different levels (mid-myocardial, and apical levels) were scanned and analyzed from each heart sample (CD 31 staining: 7 days group: ITGA5 fl/fl $n = 19$, Myα5KO $n = 12$, iMaα5KO $n = 5$; 28-day group: ITGA5 fl/fl $n = 22$, Myα5KO $n = 13$, iMaα5KO $n = 9$). Scar maturation following infarction is associated with the formation of mature neovessels, coated with α-smooth muscle actin (α-SMA) + mural cells[39].

Mature vessels in the infarct and remote remodeling myocardium were identified by using a similar immunofluorescent staining protocol as described above, with an antibody to α-SMA (Sigma, F3777) after 7 and 28 days of coronary occlusion. At least 10 different fields from two non-adjacent stained sections per mouse at 2 different levels (mid-myo-cardial, and apical levels) were scanned and analyzed per heart sample. (α-SMA staining: 7 days group: ITGA5 fl/fl $n = 19$, Myα5KO $n = 12$, iMaα5KO $n = 5$; 28-day group: ITGA5 fl/fl $n = 22$, Myα5KO $n = 13$, iMaα5KO $n = 9$). Negative controls were performed to confirm the specificity of the antibodies using sections in which the primary anti-body was substituted by isotype-matched IgG (Rat: R&D, 6-001-A; Rabbit: Merck Millipore, NI01-100UG) at the same concentration. Quantitative histological studies were performed by an investigator (R.L. or H.V.) blinded to the study groups.

## Immunofluorescence and assessment of ITGA5 expression, macrophage, and myofibroblast density

Immunofluorescence was used to study the time course of ITGA5 localization in infarct macrophages and to assess the effects of myeloid and macrophage-specific ITGA5 loss on macrophage and myofibro-blast density and on the number of mature coated vessels In order to study the time course, and cellular localization of ITGA5 activation in infarct macrophages, ITGA5 (Abcam, ab150361), and anti-GFP (Abcam, ab6662), or Wheat Germ Agglutinin, Alexa Fluor™ 633 (WGA AF633, Thermo Fisher Scientific, Waltham, MA) staining was performed on sections from infarcted CSF1R-EGFP hearts at baseline, and after 24 h, 3 days, 7 days and 28 days of coronary occlusion (control: $n = 6$; 24 h: $n = 6$; 3 days, $n = 5$; 7 days, $n = 6$; 28 days, $n = 6$). Sections were depar-affinized in Xylene, rehydrated in consecutive graded washes of ethanol, and exposed to heat-mediated antigen retrieval for 30 min with citrate buffer, pH 6.0. Sections were washed three times with TBS (5 min each) and then permeabilized with 0.1% Triton (Sigma-Aldrich, T8787) in TBS (T-TBS) for 10 min, followed up with non-specific pro-tein blocking via incubation for 1 h in 10% donkey serum (Sigma Aldrich, D9663) in TBS. Slides were then incubated overnight at 4 °C with anti-ITGA5 and anti-GFP primary antibodies. Sections were then washed with TBS and incubated with fluorescently labeled secondary antibodies for 1 h at room temperature. Antibodies dilutions were prepared in TBS. Autofluorescence quenching was performed using TrueBlack Lipofuscin Autofluorescence Quencher (Biotium, #3007) and mounted using Fluoro-Gel II with DAPI (EMS, 50-246-93). Fluor-escently stained sections were scanned using Zen 2.6 Pro software and the Zeiss Imager M2 microscope (Carl Zeiss, White Plains, NY). In addition, Leica SP8 confocal microscope (Leica Microsystems, Wetzlar, Germany) was employed for the acquisition of high-resolution images. The microscope is housed in a controlled environment to minimize vibrations in the Analytical Imaging facility of Albert Einstein College of Medicine. Leica LAS X software was utilized to control the microscope settings and for image acquisition. Laser power and gain settings were optimized to ensure minimal photobleaching, while maintaining ade-quate signal-to-noise ratios. Quantitative analysis was performed by counting the number of ITGA5-positive and GFP-positive macrophages in 15 fields from 2 different sections from each animal. To assess the number of macrophages in the infarcted and remodeling myocardium, sections were stained with Mac-2 antibody (Cedarlane, CL8942AP) in the infarcted ITGA5 fl/fl, Myα5KO and iMaα5KO hearts (Mac-2 staining: 7 days group: ITGA5 fl/fl $n = 19$, Myα5KO $n = 12$, iMaα5KO $n = 5$; 28-day group: ITGA5 fl/fl $n = 22$, Myα5KO $n = 13$, iMaα5KO $n = 9$). Macrophage quantitation was performed, using an AI model (Zen 2.6 Pro software) in which macrophages were identified as the DAPI-positive nuclei surrounded by Mac2+ profiles. In order to assess myofibroblast den-sity, we performed immunofluorescent staining with an anti-α-SMA antibody (Sigma, F3777). Myofibroblasts were identified as spindle-shaped α-SMA-positive cells located outside the vascular media. An AI-based model (Zen 2.6 Pro software) was trained on multiple fields of

different regions of the myocardium to identify myofibroblasts. Objects of interest were defined as the DAPI-positive nuclei sur-rounded by α-SMA profiles, excluding vascular smooth muscle cells. Quantitative analysis was performed using 10 fields from 2 different sections from each animal (α-SMA staining: 7 days group: ITGA5 fl/fl $n = 19$, Myα5KO $n = 12$, iMaα5KO $n = 5$; 28-day group: ITGA5 fl/fl $n = 22$, Myα5KO $n = 13$, iMaα5KO $n = 9$). Negative controls were performed to confirm the specificity of the antibodies using sections in which the primary antibody was substituted by species-specific IgG at the same concentration. Quantitative histological studies were performed by an investigator (R.L.) blinded to the study groups. Cell density was expressed as cells/mm$^2$.

## Isolation of infarct macrophages from Myα5KO and ITGA5 fl/fl mice

Infarcted Myα5KO, and ITGA5 fl/fl mice were sacrificed to harvest myeloid cells, 7 days after coronary occlusion. Myocardial tissue from control or infarcted hearts of individual mice was finely minced and placed into a digestion buffer cocktail of 0.25 mg/ml Liberase Blend-zyme 3 (Roche Applied Science), 20 U/ml DNase I (Sigma-Aldrich), 10 mmol/l HEPES (Invitrogen), and 0.1% sodium azide in HBSS with Ca2$^+$ and Mg2$^+$ (Invitrogen), and shaken at 37 °C for 20 min. Cells were then passed through a 40-μm nylon mesh. The cell suspension was then centrifuged (10 min, 500 g, 4 °C) and up to 10$^8$ cells were resus-pended in 200 μL of MACS buffer (Miltenyi Biotec, 130-091-376). Cells in MACS buffer were incubated with 50 μL of Anti-Ly-6G MicroBeads UltraPure (Miltenyi Biotec, 130-120-337) for 10 min at 4 °C, washed once and centrifuged. Cells were resuspended in MACS buffer and passed through a MS column (Miltenyi Biotec, 130-042-201) in a MACS separator (Miltenyi Biotec, 130-090-312). The magnetically labeled Ly-6G+ cells were retained on the column. The unlabeled cells (Ly-6G- flow through) were collected, washed with MACS buffer (Ly-6G- cells), and centrifuged at 500 g for 10 min. Subsequently, the cell pellet (per 10$^8$ Ly-6G- cells) was again resuspended in 90 μL of MACS buffer, incubated with 10 μL of CD11b microBeads (Miltenyi Biotec, 130-049-601) at 4 °C for 15 min, washed with MACS buffer and centrifuged as previously. Cells resuspended in 500 μL MACS buffer went through an MS column placed in a MACS separator. The mag-netically labeled CD11b+ cells were retained on the column. Ly6G-CD11b+ cells (macrophages) were flushed out and collected for RNA isolation. The RNA extracted from these cells was used for PCR array and RNA-seq experiments.

## Isolation, culture, and stimulation of mouse bone marrow macrophages

Femoral and tibial BMMs were obtained from 2- to 3-month-old Wild-type C57BL/6 J mice, as previously described[83]. Briefly, bone marrow cells were incubated for 24 h in DMEM/F-12 medium (ThermoFisher) supplemented with 15% fetal bovine serum (GIBCO), and 12 ng/ml CSF-1 (R&D systems). Non-adherent primitive mononuclear phagocytes were reseeded in media containing 15% fetal bovine serum and CSF-1 (120 ng/ml) for 48 h, in order to allow their differentiation to adherent mononuclear phagocytes. The resulting adherent macrophages were passaged once into new plates. Cells used for experiments were cul-tured until confluent. In order to examine the effects of ITGA5 signaling on macrophage phenotype, bone marrow macrophages were cultured for 24 h in the presence or absence of a neutralizing anti-ITGA5 anti-body (10 μg/ml, Biolegend, 103910), or isotype-matched IgG control (10 μg/ml, Biolegend, 400902). After 24 h of stimulation, RNA was harvested from the cells. PCR array and RNA-seq were performed to explore the effects of ITGA5 on the transcriptomic profile of the mac-rophages. The antibodies used for the in vitro integrin neutralization experiments are listed in Supplementary Table 12.

In order to investigate signaling pathways responsible for ITGA5-induced *Vegfa* synthesis in macrophages, confluent macrophages were

passaged into fibronectin (Sigma, F1141) pre-coated plates, then incubated with IgG (10 μg/ml, Biolegend, 400902) or anti-ITGA5 neutralizing antibody (10 μg/ml, Biolegend, 103910), in the presence or absence of a P38 inhibitor (SB203580 20 μM, Selleckchem, S1076), or one of two different ERK1/2 inhibitors (U0126 5 μM, Cell Signaling Technology, 9903; PD98059 50 μM, Cell Signaling Technology, 9900), or one of two different ERK5 inhibitors (XMD8-92 5 μM, Selleckchem, S7525; BIX02189 5 μM, Selleckchem, S1531), or a PI-3K/Akt inhibitor (LY294002 20 μM, Selleckchem, S1105), or a FAK inhibitor (PF573228 5 μM, Sigma, PZ0117) for 24 h. RNA was extracted from the macrophages and used for *Vegfa* qPCR.

### RNA extraction, qPCR, and PCR array

Total RNA was extracted from cells and mouse hearts using QIAzol Lysis Reagent (Qiagen, 79306) and reverse transcription was performed using iScript™ Reverse Transcription Supermix kit (Bio-Rad Laboratories, 1708840), according to the manufacturer's instructions. Quantitative real-time PCR was performed in a CFX384 Touch™ Real-Time PCR Detection System (Bio-Rad Laboratories) with SsoFast EvaGreen Supermix reagent (Bio-Rad Laboratories, 1725204). The following primer pairs were used: *β-Actin* forward 5′- CTACCTCATGAAG ATCCTGACC-3′, *β-Actin* reverse 5′- CACAGCTTCTCTTTGATGTCAC-3′; ITGA5 forward 5′- CTTCAACCTAGACGCGGAG-3′, ITGA5 reverse 5′- CG GTAAAACTCCACGGAGAAG-3′; *Vegfa* forward 5′- TAGAGTACATC TTCAAGCCGTC -3′, *Vegfa* reverse 5′- CTTTCTTTGGTCTGCATTCACA -3′. The housekeeping gene β-Actin was used as an internal control. The qPCR procedure was repeated three times in independent runs; gene expression levels were calculated using the ΔΔCT method. For PCR array, total RNA was extracted using the RNeasy mini kits (Qiagen, 74104). A total of 0.5 μg of RNA was transcribed into cDNA using the RT² first strand kit (Qiagen, 330404). Quantitative PCR was then performed using the RT² Profiler mouse angiogenesis PCR array (PAMM-024ZE-4) from Qiagen according to the manufacturer's protocol. The same thermal profile conditions were used for all primers sets: 95 °C for 10 min, 40 cycles at 95 °C for 15 s, and 60 °C for 1 min. The data obtained were exported to the SABiosciences PCR array web-based template where it was analyzed using the ΔΔCt method.

### Library preparation for transcriptome sequencing

RNA isolated from bone marrow macrophages stimulated in the presence or absence of anti-ITGA5 antibody, and from macrophages harvested from Myα5KO and ITGA5 fl/fl infarcts (7-day timepoint) was used for RNA-sequencing. RNA was sent to Novogene (Sacramento, California) for the construction of a total of eighteen cDNA libraries by using NEBNext® Ultra™ RNA Library Prep Kit for Illumina® (NEB, Ipswich, MA, USA). In brief, the process of library construction consisted of i) mRNA purification and enrichment from total RNA using oligo(dt)-attached magnetic beads, ii) fragmentation of purified mRNA using divalent cations exposed to elevated temperatures in NEBNext First Strand Synthesis Reaction Buffer, iii) double-stranded cDNA synthesis, using Rnase H- reverse transcriptase (first strand) and DNA polymerase I, dNTP and RNase H (second strand); iv) terminal cDNA ends reparation by exonucleases/polymerases; and poly adenylation of the 3′ ends of the DNA fragments, v) Sequencing adaptors ligation and finally vi) cDNA fragments size selection (150–200 bp in length) which underwent PCR. PCR was performed using Phusion High-Fidelity DNA Polymerase, universal PCR primers, and Index (X) Primer. PCR products were purified (AMPure XP system), and the library concentration (>2 nM) and quality were assessed using a Bioanalyzer 2100 system (Agilent, Santa Clara, CA, USA).

### Quality analysis, mapping, and assembly

The library preparations were sequenced on Illumina Novaseq 6000 devices, generating 150 bp paired-end reads. Adapters, poly-N, and low-quality reads from the raw data were excluded to purify the data

analysis. Quality Phred Scores, Q20%, Q30%, and GC contents of the clean data were calculated, showing high accuracy of reads (>99%, with 0.02 error rate). Filtered reads were aligned to the C57BL/6 J reference genome (Mus Musculus reference genome release December 2011 (*mm10*; GRCm38.p6) from Ensembl) using TopHat2 algorithm alignment program v.2.0.9[84]. Mapped reads were assembled using both Scripture (beta2) and Cufflinks v.2.1.1 algorithm[85].

### Gene expression, differential expression, enrichment, and co-expression analysis

HTSeq software v.0.6.1[86] was used to count the number of reads mapped to each gene. The read count of fragments per kilobase of transcript sequence per millions base pairs sequenced (FPKM) was used to calculate gene expression level, which considered the effects of both sequencing depth and gene length[85]. Read counts obtained from the gene expression analysis were used for differential expression analysis. Cluster differential expression analysis for every gene in the above different conditions was performed using the DESeq2 R software package (v.1.10.1)[87]. Genes with an adjusted *P*-value ≤ 0.05 were considered to be differentially expressed. The List of genes was ranked by differential gene expression as log2 (fold change) between each comparison group. Positive values were upregulated genes, whereas negative values were downregulated genes. Ingenuity pathway analysis (IPA) was used to identify the pathways which are regulated by α5 integrin in macrophages. All RNA-seq processed data have been deposited in NCBI's Gene Expression Omnibus and are accessible through GEO SuperSeries accession number GSE190837 (GSE190835 in vitro study and GSE190836 in vivo study).

### Analysis of integrin gene expression in infarct macrophages

RNA-seq data from an experiment comparing gene expression between macrophages from sham hearts and infarcted hearts after 3 and 7 days of coronary occlusion were used to determine the time course of integrin gene expression in infarct macrophages (GSE187701) In this experiment, CD11b + /Ly6G- macrophages were harvested from infarcted Smad7 fl/fl animals (Jackson Laboratory, stock No: 017008)[88,89]. These animals served as control mice in a study examining the role of macrophage Smad7 in myocardial infarction. The following genes were analyzed: *Itga1, Itga2, Itga3, Itga4, Itga5, Itga6, Itga7, Itga8, Itga9, Itga10, Itga2b, Itgad, Itgae, Itgal, Itgam, Itgav, Itgax, Itgb1, Itgb2, Itgb3, Itgb4, Itgb5, Itgb6, Itgb7, and Itgb8*. A comparison of gene expression levels between groups was performed using FPKM normalized counts.

### Flow cytometry

Flow cytometry was used to study the time course of ITGA5 expression in infarct macrophages. Male and female WT C57Bl6/J mice underwent non-reperfused infarction protocols (7 or 28 days of coronary occlusion). Sham controls underwent thoracotomy without coronary occlusion (Sham; n = 5, Myocardial Infarction (MI) 7 days; n = 8, MI 28 days; n = 11). Mice were sacrificed and intracardially perfused with 25 mL of ice-cold phosphate-buffered saline to remove blood cells. Whole hearts were quickly minced and digested by shaking the tissue fragments (15 min at 37 °C water bath, repeated 3 times) in 3.0 ml of digestion buffer (RPMI 1640-medium (Sigma-Aldrich) containing 120 μg/ml of Liberase TH (Roche, 5401151001) and 70 μg/ml of DNase I (Roche, 10104159001). After digestion, single-cell suspensions were incubated in hemolytic buffer (pH 7.4; 0.15 M Ammonium Chloride, 0.01 M Potassium bicarbonate, 0.0001 M EDTA) for 2 min at room temperature to remove red blood cells and washed with ice-cold phosphate-buffered saline containing 0.5% BSA and 0.002 M EDTA. After red blood cell removal, cells were harvested by passing through a 30-μm cell strainer and by washing with ice-cold phosphate-buffered saline. After Fc receptors were blocked with anti-CD16/32 antibody (BD Pharmingen, 553142) at room temperature for 15 min, cells collected

from each heart were used for assessment of ITGA5 expression in macrophages. Half the cells from each heart were incubated with an anti-CD49e/ITGA5 antibody along with antibodies necessary to identify macrophages (Supplementary Table 13) at 4 °C for 15 min. The other half of the cells were incubated with a mixture of antibodies containing isotype control (instead of the anti-CD49e/ITGA5 antibody), along with the antibodies needed for macrophage identification. Subsequently, cells were incubated with DAPI (300 nM, Thermo Fisher Scientific, MP01306) for 5 min at 4 °C to identify live cells (as DAPI-negative). After staining, cells were analyzed by flow cytometry (Aurora, Cytek) with FlowJo software (BD Biosciences). After removing doublets and identifying DAPI- live cells, we calculated the MFI for CD49e/ITGA5 and isotype control in macrophages, which were identified as CD45 + CD11b + Ly6G- CD64+ MerTK+ cells. Finally, CD49e expression in macrophages was assessed by subtracting the isotype control MFI from the MFI for CD49e/ITGA5.

Flow cytometry was also used to compare infiltration of the infarct with leukocytes between Mya5KO mice and corresponding ITGA5 fl/fl controls. Male and female Mya5KO mice and ITGA5 fl/fl animals (Mya5KO mice; $n = 5$, ITGA5 fl/fl mice; $n = 5$) underwent non-reperfused infarction protocols (7 days coronary occlusion). Single-cell suspensions were harvested as described above. Isolated cells were incubated with anti-CD16/32 antibody (BD Pharmingen, 553142) at room temperature for 15 min to block Fc receptors. After staining with a mixture of antibodies (Supplementary Table 14) at 4 °C for 15 min, cells were incubated with DAPI (Thermo Fisher Scientific, MP01306) for 5 min at 4 °C, to identify live cells (as DAPI-negative cells). Subsequently, the cells were analyzed by flow cytometry (Aurora, Cytek) with FlowJo software (BD Biosciences) to evaluate the percentage and the number of immune cells. After removing doublets and identifying DAPI-live cells, we evaluated the percentage and the absolute number of DAPI- CD45 + CD11b+ cells (myeloid cells), DAPI-CD45 + CD11b + Ly6G- CD64+ MerTK+ cells (macrophages), and DAPI- CD45 + CD11b- CD3e+ cells (T cells).

## Fluorescence-activated cell sorting (FACS) of infarct macrophages and single cell RNA-sequencing (scRNA-seq)

The transcriptome of isolated macrophages from 12–16 week-old CSF1R$^{EGFP}$ mice was studied using scRNA-seq. Macrophage suspensions were obtained from 3 control CSF1R$^{EGFP}$ hearts (pooled in one sample) and from 3 infarcted CSF1R$^{EGFP}$ hearts after 7 days of coronary occlusion (as independent samples). Briefly, infarcted heart tissue was minced and placed into a cocktail of Liberase TH (125 µg/mL) (Roche) and DNase (20 U/ml) (Sigma) in HBSS + + solution (Hanks balanced salt solution, Gibco) for 10 min at 37 °C in an orbital shaker. After enzymatic incubation, tissue was mechanically digested by slowly pipetting to reach a single-cell suspension. The supernatant was filtered through a cell strainer (40 µm, nylon; Falcon), the digestion was repeated once or twice with the remaining pieces, and the supernatants were pooled together. The total time for enzymatic digestion was around 30–45 min. Cell suspension was centrifuged for 20 min at 200 $g$ with centrifuge brakes deactivated to remove cell debris. Erythrocytes were removed using RBC lysis buffer (eBioscience). The cell suspension was then centrifuged at 500 $g$ for 5 min at 4 °C and resuspended in 100 µL of PBS containing 2% FBS and 2.5 mM EDTA (sorting buffer). Cells in single-cell suspensions were blocked with anti-mouse CD16/CD32 (1:250, BD Biosciences) for 15 min at 4 °C. In order to identify neutrophils, the cell suspension was incubated for 30 min at 4 °C in the dark with anti-Ly6G- PerCP/Cyanine5.5 (1:100, BioLegend). The cell suspension was washed twice with PBS/5% FBS, centrifuged at 500 $g$ for 5 min at 4 °C, and the final pellet was resuspended in 200 µL of sorting buffer. Standard, strict forward scatter width *versus* area criteria were used to discriminate doublets and gate-only singleton cells. Non-neutrophils (Ly6G-) were gated to identify EGFP+ macrophages, which were sorted with the FACSAria Sorter (BD Biosciences).

The isolated cells were prepared for sequencing using Single Cell 3′ Reagent Kits v3.1 (10X Genomics) according to the manufacturer's instructions. Briefly, the total of EGFP$^+$/Ly6G$^-$ events were sorted in 1X PBS/2% FBS, and the number of cells was quantified in a Neubauer chamber. A maximum of 17,000 cells were loaded at a concentration of 1000 cells/µL on a Chromium Controller instrument (10X Genomics) to generate single-cell gel bead-in-emulsions (GEMs). In this step, each cell was encapsulated with primers containing a fixed Illumina Read 1 sequence, followed by a cell-identifying 16 bp 10X barcode, a 12 bp Unique Molecular Identifier (UMI) and a poly-dT sequence. A subsequent reverse transcription yielded full-length, barcoded cDNA. This cDNA was then released from the GEMs, PCR-amplified and purified with magnetic beads (SPRIselect, Beckman Coulter). Enzymatic Fragmentation and Size Selection was used to optimize cDNA size prior to library construction. Illumina paired-end adaptor sequences (P5 beginning and P7 end) were added and the resulting library was amplified via end repair, A-tailing, adaptor ligation and PCR. Library quality control and quantification was performed using Qubit 2.0 Fluorometer (ThermoFisher Scientific) and Agilent's 4200 TapeStation System (Agilent), respectively. The libraries were also quantified by real time PCR (KAPA biosystems). Sequencing was performed in an Illumina HiSeq at the recommended configuration (Read1: 150 bp; Index i7: 8 bp each; Read2: 150 bp) at an average depth of 50,000 reads/cell. Raw sequence data (.bcl files) generated from the sequencer were converted into fastq files and de-multiplexed using the 10X Genomics' cellranger mkfastq command.

## Single cell RNA-seq data analysis
The 10X scRNA-seq data were processed with the CellRanger software (v3.1.0, 10X Genomics) with the mouse mm10 reference genome, using the default setting. The subsequent analysis was performed with the RISC package (v1.0)[90]. The gene expression matrices were first filtered for each of the four samples independently, selecting cells with 500 to 40000 unique molecular identifiers (UMIs) and expressing >200 genes. Genes expressed in <3 cells were also removed, and mitochondrial genes were excluded for data normalization across cells. The integration of the four sets of scRNA-seq data used the control sample as the reference and the top 20 principal components (PCs). Dimension reduction and clustering used top 20 PCs and 25 nearest neighbors. Differential expression for defining cluster markers (selected clusters versus all other clusters) or comparison of gene expression between normal and infarcted heart macrophages was performed with the Wilcoxon Rank Sum test, or the QuasiPoisson model for statistical testing using adjusted $p$-value (padj) <0.01. Genes expressed in <10% of cells and <10 cells of the targeted clusters were not subject to statistical test. Single-cell RNA-seq data are available in the NCBI's Gene Expression Omnibus under accession number GSE227251.

## Protein extraction and western blotting
Protein from macrophages was extracted in RIPA lysis buffer (Thermofisher, 89900) along Halt™ Protease Inhibitor Cocktail (Thermofisher, 78429). An equal amount of protein from each experimental group was fractionated by 10% Mini-PROTEAN® TGX™ Gel (Bio-Rad Laboratories, 456−1036) and transferred onto a PVDF membrane (Bio-Rad Laboratories, 1620177). Membranes were incubated with anti-ITGA5 (Abcam, ab150361), p-AKT (Cell Signaling Technology, 4060), AKT (Cell Signaling Technology, 9272), p-P38 (Cell Signaling Technology, 4511), P38 (Cell Signaling Technology, 8690), p-ERK1/2 (Cell Signaling Technology, 4370), ERK1/2 (Cell Signaling Technology, 4695), p-FAK (Cell Signaling Technology, 3283), FAK (Cell Signaling Technology, 3285), beta Actin (Cell Signaling Technology, 4970) antibodies in the appropriate dilution, followed by the application of appropriate horse radish peroxidase (HRP)-conjugated secondary antibodies. Proteins were developed using Pierce™ ECL Western blot detection reagents (Thermofisher, 32106) and imaged

by the ChemiDoc™ MP System (Bio-Rad Laboratories). Bands were densitometrically quantified by Image Lab 3.0 software (Bio-Rad Laboratories) and normalized to appropriate loading controls. The antibodies used for Western blotting experiments are listed in Supplementary Table 15.

## Assessment of VEGFA protein levels

Mouse bone marrow macrophages were cultured for 24 h in the presence or absence of a neutralizing anti-ITGA5 antibody (10 μg/ml, Biolegend, 103910), or isotype-matched IgG control (10 μg/ml, Biolegend, 400902). After 24 h of stimulation, the cell supernatants were collected, and VEGFA levels were quantitatively assessed using an ELISA kit (R&D, catalog #: MMV00), according to the manufacturer's instructions.

## Statistical analysis

For comparisons of two groups unpaired two-tailed Student's t-test was used. For comparisons of multiple groups, one-way ANOVA was performed followed by the Sidak or Bonferroni multiple comparison tests. The Kruskal-Wallis test, followed by Dunn's multiple comparison posttest was used for non-Gaussian distributions. The Shapiro-Wilk or the Kolmogorov-Smirnov test was used to test normality. Survival analysis was performed using the Kaplan–Meier method. Mortality was compared using the log-rank test. Data are expressed as means ± SEM. Statistical significance was set at $p < 0.05$.

## Reporting summary

Further information on research design is available in the Nature Portfolio Reporting Summary linked to this article.

## Data availability

All RNA-seq processed data have been deposited in NCBI's Gene Expression Omnibus and are accessible through GEO SuperSeries accession number GSE190837 (GSE190835 in vitro study and GSE190836 in vivo study). Single cell RNA-seq data are available in the NCBI's Gene Expression Omnibus under accession number GSE227251. All other data needed to reproduce the results presented here are contained within the manuscript, figures and supplementary information. Source data are provided with the paper. Source data are provided with this paper.

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

## Acknowledgements

Dr. Frangogiannis' laboratory is supported by NIH grants R01 HL76246, R01 HL85440, and R01 HL149407, and by Department of Defense grants PR181464 and PR211352. Dr. Ruoshui Li and Dr. Jun Li were supported by the Chinese Scholarship Council. Dr. Humeres, Dr. Chen, Dr Hanna, and Dr Venugopal were supported by American Heart Association postdoctoral awards. Dr Kubota was supported by the Japanese Heart Association. Dr Tuleta was supported by a post-doctoral grant from the Deutsche Forschungsgemeinschaft (TU 632/1-1). Dr Ruoshui Li is currently located in the Department of Cardiology, The Affiliated Hospital of Xuzhou Medical University, Xuzhou, China.

## Author contributions

R.L. and N.G.F. performed study concept and design, R.L., B.C., A.K., A.H., C.H., S.C.H., I.T., S.H., H.V., F.Z., K.S., and J.L. performed the experiments, R.L., B.C., A.K., A.H., C.H., S.C.H., Y.L., R.M., I. T., S.H., H.V., J.Z., D.Z., and N.G.F. analyzed and interpreted the data, R.L. and N.G.F. wrote the manuscript, all authors critically revised the manuscript and approved the final version.

## Competing interests

The authors declare no competing interests.
