## [Peer Review File · Nature Communications]

The protective effects of macrophage-specific integrin $\alpha 5$ in myocardial infarction are associated with accentuated angiogenesisEditorial Note: Parts of this Peer Review File have been redacted as indicated to remove third-party material where no permission to publish could be obtained.

REVIEWER COMMENTS

Reviewer #1 (Remarks to the Author):

This manuscript by Li et al. comes from a well established laboratory and deals with a timely and interesting topic, the role of macrophages in post-MI healing. The tested hypotheses are relevant and novel, and the provided data support the drawn conclusions. The work is technically sound. The authors describe that the integrin receptor $\alpha 5$, expressed by macrophages, has a role in infarct healing and post-MI recovery. Using two different cre drivers, the receptor was deleted in macrophages and interesting phenotypes reported. The selection of the deleted receptor was guided by initial transcription screening. The used Cre lines make sense, and their agreement is encouraging. KO led to larger ventricles over time, indicating worse infarct healing (this information, shown in 3D and 4B should be added to the abstract, which may in general profit from some editing as it is fairly confusing, with some seemingly irrelevant information on methodology that could be cut). Mechanistically, the impaired healing is due to reduced Vegfa production by macrophages, which compromises angiogenesis and recovery. There is also data on the intracellular signaling that leads to the observed phenotypes. Overall, this is a fairly complete, data rich paper that deals with an important topic. I don't see any major shortcomings.

Given the amount of data already in this manuscript, I am hesitant to request more experiments. One suggestion however would be to study the number of macrophages and their blood progenitors (monocytes) by FACS. The authors could also put this as a suggestion into the discussion.

Reviewer #2 (Remarks to the Author):

In the manuscript, "Macrophage-specific $\alpha 5$ integrin protects the infarcted heart from adverse remodeling, by stimulating angiogenesis through PI-3K and FAK pathways," Li et al.

demonstrate the integral role of integrin $\alpha 5$ in macrophages in response to cardiac injury. The authors show that integrin $\alpha 5$ is upregulated on the 7th day following myocardial infarction and that upregulation is partly due to the expression of integrin $\alpha 5$ on the infiltrating macrophages. The authors use two different mouse models to demonstrate that macrophage-specific integrin $\alpha 5$ regulates vascularization of the infarct zone by regulating the expression of an important angiogenic factor, VEGFA. The authors show that PI3K and FAK signaling downstream of integrin $\alpha 5$ regulate VEGFA transcription. Overall, this is a nice and well-done study that will greatly interest the public.

Major Comments:

- 1) Can the authors show whether or not VEGF A protein is secreted by macrophages upon stimulation of integrin $\alpha 5$ (e.g. perform ELISA for VEGFA following integrin $\alpha 5$ stimulation by Fn1 or any other way)?
- 2) Can authors exclude the role of Cre expression in their phenotype? There are no Cre+ control mice in their study.

Minor Comments:

- 1) Figure 2: Expression of Itga5 in macrophages in Fig 2A is not clearly shown. Not clear what R designates in Fig. 2A. Not clear what green and yellow stainings mark in panels 7d R and 28d R.
- 2) Supplemental Figure VI: Itga5 expression and deletion are not clearly shown in panel C. Maybe the authors can show magnified panels as well as show separate channels. The legend to this figure does not mention arrows in panel C.
- 3) Supplemental Figure XI: Please mention whether or not p values in the table of panel C were adjusted for multiple testing.

Reviewer #3 (Remarks to the Author):

Review Li et al. „ Macrophage-specific alpha5 integrin protects the infarcted heart from adverse remodeling, by stimulating angiogenesis through PI-3K and FAK pathways”

In this study the authors report on Integrin alpha5 (ITGA5) as one of the most induced integrins in cardiac macrophages shortly after myocardial infarction in mice.

Mice with myeloid cell-specific genetic deletion (LyzM-Cre) of *Itga5* show modest effects on cardiac function and infarct size, and reduced angiogenesis at later time points (28d) upon infarction. The authors report a similar phenotype in a similar mouse model, where *Cx3cr1-Cre* is used to delete *Itga5* in macrophages. Mechanistically, the authors report that either genetic deletion of *Itga5* in macrophages or application of an antibody to the latter suppresses activation of the PI3kinase and FAK-pathway and VEGFA mRNA and protein. To this reviewer the study contains a number major limitations, that do not justify the major claims posed by this study.

Major points:

- 1) The authors should define more clearly what cells express ITGA5 and whether the cells sequenced in Fig.1 are indeed pure macrophages. Cd11b⁺/Ly6G⁻ magnetic bead selection is used to purify cardiac macrophages. As this one step procedure often leads to contaminations/noise from other cell types, was the cell fraction controlled for impurities? Flow cytometry measurement should be able to dissect the level of EC (CD31), FB (collagens, *Pdgfrb*) or cardiomyocyte contamination. This reviewer could not find data for the purity of the cell fractions studied. This point appears of considerable relevance, as the composition of the myeloid cell fraction drastically changes upon myocardial infarction. So perhaps the ‘upregulation’ of ITGA5 may turn out also result from different cell populations infiltrating the myocardium.
- 2) For their RNA-Seq experiment in Figure 1, they use CD11b as a macrophage marker, for immunofluorescence staining, however, they use CSF1R, yet this receptor is also expressed on monocytes? If the authors want to make the distinction, a macrophage-specific gene/protein such as Cd68 or F4/80 should be stained as well.
- 3) For analysis of infiltration parameters, *Mac2*/*Galectin3* was used to stain the macrophage

population. While Galectin3 is an important molecule for migration and infiltration, it also might stain/identify monocytes. Why was this staining chosen instead of CSF1R or a more Mac-specific protein?

4) The experiments in figure 6 use another Cre-line (CX3CR1) to recapitulate the findings of the knockout using LyzM-Cre. It is unclear why the authors didn't use this model from the start as they want to analyze the role of ITGA5 in macrophages? To what extent are the other myeloid cells affected from ITGA5 loss in the first (myeloid cell) model? What is the (mature) macrophage content within the LyzM positive myeloid cell fraction in the myocardial infarction model?

5) Capillary rarefaction has been reported during cardiac remodeling in many studies. Whether the observed (modest) effects on remodelling are causally related to macrophage-mediated effects on angiogenesis appears not clear and the authors should tone down the respective claims.

6) Figure 5: CD31 immunohistochemistry and alphaSMA immunofluorescence are used to show a reduction of angiogenesis after ITGA5 depletion in myeloid cells. For analysis, two non-adjacent sections per mouse were used. As vessels are difficult to quantify with sections, it is unclear, if two sections yield enough statistical power. Please comment.

7) The in vitro experiments would benefit from a control group without Itga5 activation, i.e. in the absence of fibronectin.

8) The authors may comment, why they chose to study ITGA5 and not other molecules upregulated to a similar or even greater extent (ITGB5)?

Reviewer #4 (Remarks to the Author):

Li and colleagues have explored the role of alpha5 integrin (ITGA5) expressed by macrophages during infarct healing. They have used two Cre deleter lines (constitutive LyzM-Cre and tamoxifen-inducible CX3CR1-CreER) to delete ITGA5 specifically in macrophages and found that macrophage ITGA5 promotes angiogenesis and attenuates adverse left ventricular remodeling (dilation) after acute myocardial infarction (AMI). Mechanistically, they propose that ITGA5 augments VEGF expression in macrophages via activation of PI3K and FAK.

My main concern with this paper is that the ITGA5 expressing “macrophage” cell population(s) remain poorly defined (e.g. resident vs. recruited). Defining macrophages as CD11b+/Ly6G- (Figure 1) is too simplistic. Immunostaining images presented in Figure 2 are poor quality. FACS should be used to better define monocyte/macrophage subsets expressing ITGA5 during AMI healing (time course). Considering that ITGA5 is a cell surface protein it might be included in more sophisticated FACS panels. Moreover single (myeloid) cell RNAseq should be used to better define ITGA5-expressing myeloid cell subsets.

Similar considerations apply when studying ITGA5 effects on post AMI inflammation (Mac2 immunofluorescence staining alone is not sufficient; Suppl. Figure III). The authors should use established flow cytometry panels to assess the impact of macrophage-specific ITGA5 deletion on the inflammatory response after AMI (studying diverse myeloid and lymphoid cell subsets).

Considering that the majority of patients with AMI undergoes reperfusion therapy, the impact of LyzM-Cre deletion on LV remodeling and angiogenesis should be briefly explored/confirmed in a transient coronary artery ligation mouse model.

Is a ca. 20% reduction in VEGF expression in ITGA5 KO infarct macrophages (detected by angiogenesis PCR array in Figure 7A) really sufficient to explain the more adverse left ventricular remodeling after AMI in the KO mice? How does the ITGA5 KO impact overall left ventricular VEGF expression levels?

Are non-angiogenesis related growth factors/genes affected by macrophage ITGA5 KO? For example, genes that have a direct impact on scar maturation (Figure 4)?

Minor:

Line 53: authors should not speculate about ischemic cardiomyopathy as this is not examined in this manuscript.

Line 176: LyzM-Cre does not enable “macrophage-specific” gene deletion.

Line 295: grammar?

Figure 1A: is each line representing 1 mouse?

Suppl. Figure XI: there appears to be a large interindividual variability within the WT and KO groups (3 each). This requires a comment.

RESPONSE TO THE REVIEWERS:

We are grateful to the reviewers for their comments that improved the quality of our work. We have performed extensive new experiments to address the reviewers' criticisms:

1. In order to better define the macrophage identity of ITGA5⁺ cells, we have performed **new flow cytometry experiments** that clearly demonstrate increased ITGA5 levels in infarct macrophages, 7 days after myocardial infarction. The new data are shown in Figure 1A-C (the gating strategy is illustrated in Supplemental Figure II).

New flow cytometry experiments were also performed to examine the effects of myeloid cell-specific ITGA5 loss on infiltration of the infarct with leukocytes. The new data are shown in Figure 4, and the gating strategy is presented in Supplemental Figure VI. The findings are consistent with the immunofluorescence studies, suggesting no significant effects of myeloid cell-specific ITGA5 loss on myeloid cell, macrophage and lymphocyte infiltration.

2. **New single cell RNAseq experiments** were performed to better define the macrophage clusters that express various integrins and to identify relations between integrin expression and specific phenotypes. We have performed scRNA-seq on cells sorted from control and infarcted CSF1R^{EGFP} macrophage reporter mice. The new data are shown in Figures 7 and 8, in Supplemental Figures XVII and XVIII, and in Supplemental Tables I-III. The findings show cluster-specific patterns of integrin expression in infarct macrophages. Importantly, the cluster exhibiting high expression of *Itga5*, has high expression levels of several angiogenic genes, including *Vegfa*, *Vegfb*, *Angpt2*, *Angptl2*, *Angptl4* and *Angptl6*. The high *Itga5* expression in the angiogenic cluster of macrophages further supports our myeloid cell-specific and macrophage-specific in vivo studies on the role of ITGA5 in stimulation of an angiogenic macrophage program.

3. We have explored additional mediators that may be responsible for the effects of myeloid cell and macrophage-specific ITGA5 loss. We used our RNAseq analysis of the profile of infarct macrophages to identify mediators with roles in inflammation, repair and matrix remodeling, which are differentially regulated by ITGA5 loss. Because many of these genes may reflect secondary consequences of the effects on remodeling (rather than primary targets of ITGA5 in macrophages), we examined which of these genes are modulated in a similar manner by ITGA5 blockade in vitro. The data are presented in Supplemental Table XI and presented in the results section. We found that expression of *emilin2*, encoding the angiogenic matricellular protein EMILIN2 (elastin microfibril interface located protein 2), and *Ecm1* (which encodes the matrix protein Extracellular matrix protein 1 (ECM1), a potent stimulus of endothelial cell proliferation) was consistently suppressed upon ITGA5 disruption in vivo and in vitro. Thus, ECM1 and EMILIN2 may be additional ITGA5-dependent angiogenic mediators secreted by infarct macrophages.

4. We have addressed all other concerns raised by the reviewers. We present new immunofluorescence images to clearly show the ITGA5⁺ macrophages (Figure 1D, Supplemental Figure III, Supplemental Figure XI). We show new data excluding Cre effects on macrophage-mediated actions in our genetic models (Supplemental Figure X, Supplemental Figure XIII). We

show new data on VEGFA protein levels (Figure 9J). We have also clarified important points made by the reviewers.

Overall, the revised version contains several new figures (Figures 1, 4, 7, 8, Supplemental Figures II, III, VI, X, XIII, XVII, XVIII) and Tables (Supplemental Tables I, II, III, XI, XII, XIII). Drs Yang Liu, Richard Ma, Izabela Tuleta, Harikrishnan Venugopal, Fenglan Zhu, Kai Su, Jinghang Zhang and Deyou Zheng who contributed to the new scRNA-seq, flow cytometry, and immunofluorescence experiments presented in the revised version were included as co-authors. We have addressed the specific concerns raised by the reviewers as follows:

REVIEWER COMMENTS

Reviewer #1 (Remarks to the Author):

This manuscript by Li et al. comes from a well established laboratory and deals with a timely and interesting topic, the role of macrophages in post-MI healing. The tested hypotheses are relevant and novel, and the provided data support the drawn conclusions. The work is technically sound. The authors describe that the integrin receptor $\alpha 5$, expressed by macrophages, has a role in infarct healing and post-MI recovery. Using two different cre drivers, the receptor was deleted in macrophages and interesting phenotypes reported. The selection of the deleted receptor was guided by initial transcription screening. The used Cre lines make sense, and their agreement is encouraging. KO led to larger ventricles over time, indicating worse infarct healing (this information, shown in 3D and 4B should be added to the abstract, which may in general profit from some editing as it is fairly confusing, with some seemingly irrelevant information on methodology that could be cut). Mechanistically, the impaired healing is due to reduced Vegfa production by macrophages, which compromises angiogenesis and recovery. There is also data on the intracellular signaling that leads to the observed phenotypes. Overall, this is a fairly complete, data rich paper that deals with an important topic. I don't see any major shortcomings.

Thank you very much for your kind and insightful comments, and for your helpful recommendations. We have revised the abstract to clarify the message and also to comply to the journal's word limits.

Given the amount of data already in this manuscript, I am hesitant to request more experiments. One suggestion however would be to study the number of macrophages and their blood progenitors (monocytes) by FACS. The authors could also put this as a suggestion into the discussion.

Thank you for your comment. We have performed new flow cytometry experiments, in order to study the time course of ITGA5 expression in macrophages, and to compare infiltration of

leukocytes, macrophages and lymphocytes between myeloid cell-specific ITGA5 KO and control infarcts.

Flow cytometry for ITGA5 and macrophage markers is shown in Figure 1 (the gating strategy is illustrated in Supplemental Figure II). The findings show that the number of ITGA5+ macrophages peaks 7 days after infarction. The findings are consistent with the immunofluorescent staining (Figure 1) and RNA-seq data (Supplemental Figure I), which suggests a similar time course of ITGA5 protein expression in macrophages infiltrating the infarct.

Moreover, we have performed new flow cytometry experiments to examine the effects of myeloid cell-specific ITGA5 loss on macrophage and lymphocyte infiltration. The new data are shown in Figure 4, and the gating strategy for this experiment is illustrated in Supplemental Figure VI. The findings show no significant effects of myeloid cell-specific ITGA5 loss on recruitment of myeloid cells, macrophages and T lymphocytes in the healing infarct. Thus, these observations are in agreement with the immunofluorescence data shown in Supplemental Figure VII.

Reviewer #2 (Remarks to the Author):

In the manuscript, "Macrophage-specific $\alpha 5$ integrin protects the infarcted heart from adverse remodeling, by stimulating angiogenesis through PI-3K and FAK pathways," Li et al. demonstrate the integral role of integrin $\alpha 5$ in macrophages in response to cardiac injury. The authors show that integrin $\alpha 5$ is upregulated on the 7th day following myocardial infarction and that upregulation is partly due to the expression of integrin $\alpha 5$ on the infiltrating macrophages. The authors use two different mouse models to demonstrate that macrophage-specific integrin $\alpha 5$ regulates vascularization of the infarct zone by regulating the expression of an important angiogenic factor, VEGFA. The authors show that PI3K and FAK signaling downstream of integrin $\alpha 5$ regulate VEGFA transcription. Overall, this is a nice and well-done study that will greatly interest the public.

Thank you very much for your kind and insightful comments, and for your helpful recommendations.

Major Comments:

1) Can the authors show whether or not VEGF A protein is secreted by macrophages upon stimulation of integrin $\alpha 5$ (e.g. perform ELISA for VEGFA following integrin $\alpha 5$ stimulation by Fn1 or any other way)?

We have performed new experiments examining the effects of ITGA5 neutralization on VEGFA protein levels. Consistent with the mRNA data, the protein analysis shows that VEGFA secretion by isolated macrophages, cultured in the presence of fibronectin, is dependent on ITGA5. The new data are shown in Figure 9J.

2) Can authors exclude the role of Cre expression in their phenotype? There are no Cre+ control mice in their study.

Thank you very much for raising this important point. To address this question, we have performed new experiments examining the effects of Cre (described below). However, first we would like to explain the basis for our initial use of floxed controls. There is extensive evidence that Cre expression has effects on cardiomyocyte function; for this reason, Cre controls are routinely included in most investigations examining cardiomyocyte-specific manipulations, especially when the MCM system is used. In contrast, in investigations examining the role of macrophages through the use of the Cre-lox system, the vast majority of studies have used floxed controls, as there has been no evidence on Cre actions on macrophage phenotype. To illustrate this, we have reviewed manuscripts examining macrophage-specific actions using the LysM^{Cre} and CX3CR1^{CreER} drivers that were recently (2019-2022) published in *Nature Communications* and in other high-impact journals. Virtually all the studies we identified (Reviewer Table I and II) used floxed controls, ignoring any possible actions of Cre recombinase. The absence of significant Cre effects is also supported by our own published work comparing effects of Smad2 and Smad3 deletion in myeloid cells using the LysM Cre driver. Our studies showed that although myeloid cell-specific Smad3 loss perturbed phagocytosis and impaired suppression of post-infarction inflammation¹, myeloid cell-specific Smad2 deletion had no significant effects². The absence of significant differences between LysM^{Cre};Smad2fl/fl and Smad2 fl/fl animals following infarction² suggests that Cre expression in myeloid cells has no major impact on their phenotype.

Reviewer Table I: Control groups in studies using the LyzM Cre driver to explore myeloid cells-specific actions in disease

Gene targeted	Model	Controls	Journal/Year	Reference
Lgmn	Myocardial infarction	Lgmn fl/fl	Circulation/2022	³
Bcn1	Tumorigenesis	Bcn1 fl/fl	J Clin Invest /2019	⁴
Smad3	Myocardial infarction	Smad3 fl/fl	Circ Res/2019	¹
Cd40	Atherosclerosis model	CD40 fl/fl	Cardiovasc Res/2022	⁵
Grx1, Fabp5	Acute lung injury model	Grx1 fl/fl , FABP5 fl/fl	Nat Commun/2021	⁶
Arg1	Lung tumor model	Arg1 fl/fl	Nat Commun/2022	⁷
Noc4l	Model of obesity-related insulin resistance	Noc4l fl/fl	Nat Commun/2021	⁸
Alk5	Colorectal cancer model	C57B16J	Nat Commun/2020	⁹
Mettl3	Infection and tumor model	Mettl3 fl/fl	Sci Adv/2021	¹⁰

Reviewer Table II: Control groups in studies using the Cx3cr1 CreER driver to explore myeloid cells-specific actions in disease

Gene targeted	Model	Controls	Journal/Year	Reference
Lgmn	Myocardial infarction	Lgmn fl/fl	Circulation/2022	³
Ask1	Neuroinflammation model	ASK1 fl/fl	PNAS/2022	¹¹
Dlg1	Neuroinflammation model	Dlg1 fl/fl	Neurosci Bull/2021	¹²
IL-10	Memory impairment studies	IL-10 fl/fl	J Neuroinflammation/2021	¹³
Tsc1	Tuberous sclerosis complex model	Tsc1 fl/fl	Epilepsia/2018	¹⁴
Tak1	Stroke model	Tak1 fl/fl	J Mol Med /2020	¹⁵
IL-6	Traumatic brain injury model	IL-6 fl/fl	Glia/2020	¹⁶

However, as the reviewer points out, Cre effects should be considered in all genetic manipulation studies. Thus, in order to exclude that the observed effects are due to Cre expression (rather than ITGA5 loss) we have performed new experiments examining the effects of Cre on the main positive endpoints of our study: adverse remodeling and infarct angiogenesis in both the LysMCre and CX3CR1CreER models. LysMCre/+ animals and Cre-/- controls underwent infarction protocols and were sacrificed 7 days after coronary occlusion (the timepoint in which myeloid cell-specific ITGA5 KO mice had worse remodeling and reduced microvascular density). Echocardiography showed that Cre expression did not affect systolic function and dilative remodeling. Moreover, microvascular density, assessed through CD31 immunohistochemistry was comparable between Cre+ and Cre-negative animals. The data are shown in Supplemental Figure X.

Similarly, we performed new experiments using tamoxifen-treated CX3CR1CreER mice and Cre-negative controls. There were no significant differences in echocardiographic parameters, in microvascular density and in the density of mature coated vessels between groups. The data are shown in Supplemental Figure XIII.

Minor Comments:

1) Figure 2: Expression of Itga5 in macrophages in Fig 2A is not clearly shown. Not clear what R designates in Fig. 2A. Not clear what green and yellow stainings mark in panels 7d R and 28d R.

2) Supplemental Figure VI: Itga5 expression and deletion are not clearly shown in panel C. Maybe the authors can show magnified panels as well as show separate channels. The

legend to this figure does not mention arrows in panel C.

We have improved the quality of the immunofluorescence data and we have performed new flowcytometry experiments to study the time course of ITGA5 expression in macrophages. Figure 1 in the revised version of the manuscript shows new flow cytometry data demonstrating an increased number of ITGA5+ macrophages in the infarcted myocardium 7 days after myocardial infarction. Moreover, new immunofluorescence images clearly show the ITGA5+ macrophages, by including the 2 channels and the merged image (Figure 1D). Supplemental Figure III shows the complete time course.

We have also revised Supplemental Figure XI to show separate channels that clearly demonstrate ITGA5 loss in macrophages of tamoxifen-treated CX3CR1^{CreER};ITGA5^{fl/fl} mice

3) Supplemental Figure XI: Please mention whether or not p values in the table of panel C were adjusted for multiple testing.

No adjustment for multiple comparisons was used for the array data presented in this figure. This is now clearly indicated in the figure legend.

Reviewer #3 (Remarks to the Author):

Review Li et al. „ Macrophage-specific alpha5 integrin protects the infarcted heart from adverse remodeling, by stimulating angiogenesis through PI-3K and FAK pathways“

In this study the authors report on Integrin alpha5 (ITGA5) as one of the most induced integrins in cardiac macrophages shortly after myocardial infarction in mice.

Mice with myeloid cell-specific genetic deletion (LyzM-Cre) of Itga5 show modest effects on cardiac function and infarct size, and reduced angiogenesis at later time points (28d) upon infarction. The authors report a similar phenotype in a similar mouse model, where Cx3cr1-Cre is used to delete Itga5 in macrophages. Mechanistically, the authors report that either genetic deletion of Itga5 in macrophages or application of an antibody to the latter suppresses activation of the PI3kinase and FAK-pathway and VEGFA mRNA and protein. To this reviewer the study contains a number major limitations, that do not justify the major claims posed by this study.

Thank you very much for the insightful comments and criticisms. We have extensively revised our study following your recommendations.

Major points:

1) The authors should define more clearly what cells express ITGA5 and whether the cells sequenced in Fig.1 are indeed pure macrophages. Cd11b+/Ly6G- magnetic bead selection is used to purify cardiac macrophages. As this one step procedure often leads to contaminations/noise from other cell types, was the cell fraction controlled for impurities? Flow cytometry measurement should be able to dissect the level of EC (CD31), FB (collagens, Pdgfrb) or cardiomyocyte contamination. This reviewer could not find data for the purity of the cell fractions studied. This point appears of considerable relevance, as the composition of the myeloid cell fraction drastically changes upon myocardial infarction. So perhaps the 'upregulation' of ITGA5 may turn out also result from different cell populations infiltrating the myocardium.

2) For their RNA-Seq experiment in Figure 1, they use CD11b as a macrophage marker, for immunofluorescence staining, however, they use CSF1R, yet this receptor is also expressed on monocytes? If the authors want to make the distinction, a macrophage-specific gene/protein such as Cd68 or F4/80 should be stained as well.

We agree with the reviewer that immunomagnetic sorting can be associated with contamination with other cell types. Thus, although the vast majority of the cells used for RNA-seq are macrophages, we cannot exclude contamination with other cell types that may have influenced the results. For this reason, we have performed extensive new experiments to examine the time course of ITGA5 expression in infarct macrophages.

a. We have examined the time course of ITGA5 expression in macrophages using flow cytometry. The findings support the immunohistochemical data, demonstrating that expression of ITGA5 in CD11b+/Ly6G-/CD64+/MerTK+ macrophages peaks 7 days after MI. The new data are shown in Figure 1. The gating strategy for this experiment is shown in Supplemental Figure II.

b. We have also performed new scRNAseq data examining the transcriptomic profile of sorted CSF1R+ cells harvested from sham and infarcted hearts. We identified 12 clusters of CSF1R+ cells. The data are presented in figures 7 and 8, and in Supplemental Figures XVII and XVIII and in Supplemental Tables I, II and III. Our findings showed cluster-specific patterns of Itga5 expression in infarct macrophages. Interestingly, a cluster with high expression of angiogenic genes, including *Vegfa*, *Vegfb*, *Angpt2*, *Angptl2*, *Angptl4* and *Angptl6* was identified, representing ~5% of the CSF1R+ cells in the infarct. These cells exhibited higher Itga5 expression levels than other clusters (Figure 8, Supplemental Table III). These findings support our in vivo targeting data, which suggest a role for macrophage ITGA5 in mediating an angiogenic macrophage phenotype.

3) For analysis of infiltration parameters, Mac2/Galectin3 was used to stain the macrophage population. While Galectin3 is an important molecule for migration and infiltration, it also might stain/identify monocytes. Why was this staining chosen instead of CSF1R or a more

Mac-specific protein?

We have previously performed a systematic study to identify the optimal macrophage marker for staining of formalin-fixed paraffin embedded sections, using 2 different reporter lines (CSF1REGFP and CX3CR1EGFP mice) and several different antibodies¹⁷. Of the antibodies we tested, Mac2 had the highest reproducibility for paraffin-embedded samples with complete absence of artifacts (supported by the use of Gals3 KO samples) and excellent quality of staining. In contrast, other markers (such as F4/80 antibodies) had poor performance in paraffin-embedded tissues. Although Mac2 is not the most specific of the anti-macrophage antibodies, the reproducibility of staining (when paraffin-embedded samples are studied) make it highly suitable for quantitative analysis.

In order to strengthen our analysis of the effects of myeloid cell-specific ITGA5 loss on macrophage recruitment, we have performed new experiments using flow cytometry for assessment of macrophage infiltration in myeloid cell-specific ITGAA5 KO and control infarcts. The new data are shown in Figure 4 (the gating strategy is illustrated in Supplemental Figure VI. The flow cytometry data support the immunofluorescence experiments, as no significant effects of myeloid cell-specific ITGA5 loss on recruitment of myeloid cells, macrophages and T lymphocytes were noted.

4) The experiments in figure 6 use another Cre-line (CX3CR1) to recapitulate the findings of the knockout using LyzM-Cre. It is unclear why the authors didn't use this model from the start as they want to analyze the role of ITGA5 in macrophages? To what extent are the other myeloid cells affected from ITGA5 loss in the first (myeloid cell) model? What is the (mature) macrophage content within the LyzM positive myeloid cell fraction in the myocardial infarction model?

We initiated breeding with the 2 different Cre drivers concurrently; however, as myeloid cell-specific KO mice became available first, we performed extensive work with this line. The 2 Cre drivers provide complementary information. The LyzM Cre driver is a robust tool for dissection of myeloid cell-specific effects, but lacks specificity for macrophages. The inducible CX3CR1CreER is more specific for macrophages, but may be less sensitive in targeting the broad range of macrophages infiltrating the infarct. We feel that the 2 lines provide complementary information. Moreover, the consistent results obtained with the 2 different drivers increase the level of confidence on the observed angiogenic effect of macrophage ITGA5.

5) Capillary rarefaction has been reported during cardiac remodeling in many studies. Whether the observed (modest) effects on remodelling are causally related to macrophage-mediated effects on angiogenesis appears not clear and the authors should tone down the respective claims.

We agree with the reviewer that it is unclear whether the observed perturbation in angiogenesis and in vascular maturation are responsible for the effects on adverse remodeling. However, we have performed a systematic study of alternative possibilities. Scar size at 7 days was not affected, suggesting no significant effects on cardiomyocyte loss. Myofibroblast density and collagen content were comparable between groups. No significant effects on macrophage and lymphocyte recruitment were noted by flow cytometry and immunofluorescence. Microvascular density and vascular maturation are consistently perturbed upon ITGA5 loss in myeloid cells and macrophages. A ~30% reduction in microvascular density may have a significant impact on perfusion, repair and function after myocardial infarction. To address the reviewer's concern, we have revised the discussion to indicate the limitations.

6) Figure 5: CD31 immunohistochemistry and alphaSMA immunofluorescence are used to show a reduction of angiogenesis after ITGA5 depletion in myeloid cells. For analysis, two non-adjacent sections per mouse were used. As vessels are difficult to quantify with sections, it is unclear, if two sections yield enough statistical power. Please comment.

For assessment of microvascular density and vascular maturation, we assessed more than 10 fields (200X magnification) from 2 different levels for each mouse. The high quality of the staining (which as illustrated in the figures is free of artifacts and provides highly reproducible results), the large number of fields (>20/mouse) and the large number of animals studied (ITGA5 fl/fl n=19, Myα5KO n=12, iMacα5KO n=5; 28-day group: ITGA5 fl/fl n=22, Myα5KO n=13, iMacα5KO n=9) increase the power of the quantitative analysis, which is sufficient to detect a 30% difference between groups,

7) The in vitro experiments would benefit from a control group without Itga5 activation, i.e. in the absence of fibronectin.

In our early studies (and in our RNA-seq experiments) we studied macrophages in the presence or absence of fibronectin. Supplemental Figures XXII and XXIII show these findings, illustrating comparisons both in the presence and absence of fibronectin. However, the effects of ITGA5 neutralization were similar in the presence or absence of fibronectin. This may reflect, at least in part, the production of endogenous fibronectin by the cells. Thus, the neutralization experiments examining the mechanisms for ITGA5 actions were performed in fibronectin-treated cells.

8) The authors may comment, why they chose to study ITGA5 and not other molecules upregulated to a similar or even greater extent (ITGB5)?

We are interested in the role of integrins in regulating macrophage phenotype and function. As the reviewer astutely points out, several other integrin chains may be of significant interest, including ITGB5. We have initiated our integrin studies with a focus on ITGB1 partners (thus, the work on ITGA5). We are currently studying the role of ITGAV and its β chain partners (including ITGB5).

Reviewer #4 (Remarks to the Author):

Li and colleagues have explored the role of alpha5 integrin (ITGA5) expressed by macrophages during infarct healing. They have used two Cre deleter lines (constitutive LyzM-Cre and tamoxifen-inducible CX3CR1-CreER) to delete ITGA5 specifically in macrophages and found that macrophage ITGA5 promotes angiogenesis and attenuates adverse left ventricular remodeling (dilation) after acute myocardial infarction (AMI). Mechanistically, they propose that ITGA5 augments VEGF expression in macrophages via activation of PI3K and FAK.

My main concern with this paper is that the ITGA5 expressing "macrophage" cell population(s) remain poorly defined (e.g. resident vs. recruited). Defining macrophages as CD11b+/Ly6G- (Figure 1) is too simplistic. Immunostaining images presented in Figure 2 are poor quality. FACS should be used to better define monocyte/macrophage subsets expressing ITGA5 during AMI healing (time course). Considering that ITGA5 is a cell surface protein it might be included in more sophisticated FACS panels. Moreover single (myeloid) cell RNAseq should be used to better define ITGA5-expressing myeloid cell subsets.

Thank you very much for the insightful comments and criticisms. We have extensively revised our study following your recommendations.

We have performed both flow cytometry and single cell RNA-seq experiments to address the concerns regarding the macrophage identity of the ITGA5+ cells:

a) We have examined the time course of ITGA5 expression in macrophages using flow cytometry. The findings demonstrate that expression of ITGA5 in CD11b+/Ly6G-/CD64+/MerTK+ macrophages peaks 7 days after MI. The new data are shown in Figure 1. The gating strategy for this experiment is shown in Supplemental Figure II.

b. We have performed new scRNAseq experiments, examining the transcriptomic profile of sorted CSF1R+ cells harvested from control and infarcted hearts (7 days after coronary occlusion). We identified 12 clusters of CSF1R+ cells. The data are presented in the results section and in figures 7 and 8, Supplemental Figures XVII and XVIII, and in Supplemental Tables I, II and III. The findings suggest cluster-specific patterns of integrin expression in infarct macrophages. Supplemental Table III shows all integrin genes that were specifically upregulated in one or more of the CSF1R+ cell clusters (in comparison to all other clusters). *Itga5* expression was significantly higher in a cluster of macrophages that exhibited an angiogenic transcriptional profile (Angiogenic macrophages, Amp, Fig 8B), when compared with cells from all other clusters (padj=0, logFC=0.87). The angiogenic macrophages significantly expanded after infarction, and were characterized by high expression of angiogenic genes, including *Vegfa*, *Vegfb*, *Angpt2*, *Angptl2*, *Angptl4* and *Angptl6*.

Moreover, comparison of integrin expression levels between infarcted and control hearts showed significant *Itga5* upregulation in a large cluster of macrophages that exhibited high expression of growth factors and matricellular genes (reparative macrophages, RMp)

The partner chain for ITGA5, ITGB1 was broadly expressed by the majority of cells in all clusters (Fig 8C). The *Itgam* (encoding ITGAM/CD11b), *Itgb2* and *Itgb5* genes were also broadly expressed in cells from all macrophage and monocyte clusters (Figure 8B-F). Other integrins were predominantly expressed by specific clusters. Inflammatory macrophages (a cluster with high expression of pro-inflammatory genes that – Imp) had high expression of *Itga1*, *ItgaL* and *Itgb7*, in comparison to other clusters. Resident cardiac macrophages (CMp) exhibited higher expression of *Itga9* in comparison to other clusters. *Itgax* was predominantly expressed by the angiogenic macrophage cluster and by the minor resident macrophage clusters (Mp11 and Mp12), whereas *Itga6* was expressed at higher levels by reparative macrophages. *Itgav* was expressed by significant subpopulations of cells from all clusters (Supplemental Fig XVII). In contrast, *Itgb4*, *Itgae*, *Itga2*, *Itga2b*, *Itga3*, *Itga7*, *Itga8*, *Itga10*, and *Itga11* expression was very low in all clusters (Supplemental Fig XVIII).

Thus, the scRNA-seq data showing that angiogenic macrophages have high expression of *Itga5* are consistent with the pro-angiogenic effects of ITGA5 in macrophages, suggested by the myeloid cell and macrophage-specific KO experiments.

c. We have improved the quality of the immunofluorescence images to illustrate infiltration of the infarct with ITGA5+ macrophages. New immunofluorescence images clearly show the ITGA5+ macrophages at the 7 and 28-day timepoints, by including the 2 channels and the merged image (Figure 1D). Supplemental Figure III shows the complete time course.

Similar considerations apply when studying ITGA5 effects on post AMI inflammation (Mac2 immunofluorescence staining alone is not sufficient; Suppl. Figure III). The authors should use established flow cytometry panels to assess the impact of macrophage-specific ITGA5 deletion on the inflammatory response after AMI (studying diverse myeloid and lymphoid cell subsets).

We have performed new experiments using flow cytometry to quantitatively assess the effects of myeloid cell-specific ITGA5 loss on the numbers of myeloid cells, macrophages and T cells. The findings are shown in Figure 4, whereas the gating strategy is illustrated in Supplemental Figure XI. The data show that myeloid cell-specific ITGA5 loss does not affect infiltration of the infarct with myeloid cells, macrophages and T lymphocytes. These findings are consistent with the Mac2 immunofluorescence (Supplemental Figure VII)

Considering that the majority of patients with AMI undergoes reperfusion therapy, the impact of *LyzM-Cre* deletion on LV remodeling and angiogenesis should be briefly explored/confirmed in a transient coronary artery ligation mouse model.

Because the focus of our study is on repair, the non-reperfused infarction model is optimal for testing the hypothesis, as it results in formation of a transmural scar, which is dependent on neovessel formation in order for healing to occur. In contrast, the reperfused infarction model induces a mid-myocardial infarct, in which the role of neovessels is more limited (as the coronary is reperfused after a brief ischemic interval). In addition to this, the small mid-myocardial infarcts

in ischemia/reperfusion models make assessment of microvascular density more challenging, and perhaps less meaningful. Reviewer Figure I illustrates this.

We agree with the reviewer that the reperfused MI model could provide valuable information; however, this information would address a different question: whether integrin-dependent activation of macrophages has an impact on acute cardiomyocyte death following ischemia (which is not affected in the non-reperfused infarction model, as all cardiomyocytes in the area at risk die).

Reviewer Figure I: H&E section illustrating the midmyocardial localization of infarction in reperfused infarcts (60 min coronary occlusion/7 days of reperfusion). Rapid restoration of flow in the infarct zone in reperfused infarction would be expected to limit any effects of changes in angiogenesis. Moreover, the non-transmural nature of the infarct (arrows) makes assessment of infarct angiogenesis challenging. Considering that the goal of the study is to examine the role of macrophage ITGA5 signaling in cardiac repair, we felt that the non-reperfused infarction model is optimal to test the hypothesis. The reperfused MI model could provide valuable information to understand any effects of integrin macrophage signaling on acute cardiomyocyte death (which is not affected in the non-reperfused infarction model, as all cardiomyocytes in the area at risk die).

Is a ca. 20% reduction in VEGF expression in ITGA5 KO infarct macrophages (detected by angiogenesis PCR array in Figure 7A) really sufficient to explain the more adverse left ventricular remodeling after AMI in the KO mice? How does the ITGA5 KO impact overall left ventricular VEGF expression levels? Are non-angiogenesis related growth factors/genes affected by macrophage ITGA5 KO? For example, genes that have a direct impact on scar maturation (Figure 4)?

These are excellent points; we greatly appreciate the recommendations. As a general rule, we feel it is unlikely that a single mediator can explain the functional consequences of deletion of a key upstream signal with broad effects on cellular phenotype. However, VEGFA is a very good candidate, due to its high level of expression, consistent changes in vivo and in vitro, and important

function in a relevant cellular response (angiogenesis). Thus, we have used *Vegfa* as a reliable readout to dissect the mechanisms responsible for ITGA5-mediated actions in macrophages. To address the reviewers' comments, we have performed additional experiments to assess VEGFA protein levels and to identify additional ITGA5-dependent macrophage-derived mediators that may explain the in vivo phenotype.

First, we have performed new experiments examining the effects of ITGA5 neutralization on VEGFA protein levels. Consistent with the mRNA data, the protein analysis shows that VEGFA secretion by isolated macrophages, cultured in the presence of fibronectin, is dependent on ITGA5. The new data are shown in Figure 9J.

Second, we have identified additional ITGA5-mediated reparative genes that may explain the effects of macrophage-specific ITGA5 loss, using our in vivo and in vitro RNA-seq data. Comparison of the transcriptomic profile of myeloid cells harvested from *Mya5KO* vs ITGA5 fl/fl infarcts showed that a significant number of proinflammatory genes (including *Il1a*, *Il1b*, *Il6*, *Cxcl3* and *Cxcl5*), anti-inflammatory genes (such as *Il10* and *Tgfb2*), and angiogenic mediators (such as *Nrg1*, *Hbegf*, and *Areg*) were downregulated in infarct macrophages upon ITGA5 disruption (Supplemental Table XI). Moreover, expression of *Lox* and *Tgm2* (encoding the matrix-crosslinking enzymes lysyl-oxidase and transglutaminase-2 respectively) was also markedly reduced in ITGA5 KO infarct macrophages. However, differential gene expression in vivo in ITGA5 KO macrophages only in part reflects direct effects of ITGA5, as expression of various genes may be affected indirectly by the consequences of cell-specific ITGA5 loss on adverse remodeling and dysfunction. In order to identify genes specifically modulated by ITGA5 signaling, we examined which of the differentially expressed genes in vivo were also modulated in vitro. We found that expression of *emilin2*, encoding the angiogenic matricellular protein EMILIN2 (elastin microfibril interface located protein 2)¹⁸, and *Ecm1* (which encodes the matrix protein Extracellular matrix protein 1 (ECM1), a potent stimulus of endothelial cell proliferation¹⁹) was consistently suppressed upon ITGA5 disruption in vivo and in vitro. Thus, ECM1 and EMILIN2 may be additional ITGA5-dependent angiogenic mediators secreted by infarct macrophages. The new data are presented in the results section and in Supplemental Table XI.

Minor:

Line 53: authors should not speculate about ischemic cardiomyopathy as this is not examined in this manuscript.

We have deleted the sentence (as the abstract was rewritten to comply with journal restrictions).

Line 176: LyzM-Cre does not enable "macrophage-specific" gene deletion.

We have corrected this statement indicating that LyzM-Cre enables myeloid cell-specific gene deletion.

Line 295: grammar?

We have corrected the sentence.

Figure 1A: is each line representing 1 mouse?

Yes, we have clarified this in the figure legend (this figure is now in the Supplement as Supplemental Figure I)

Suppl. Figure XI: there appears to be a large interindividual variability within the WT and KO groups (3 each). This requires a comment.

The significant variability is typical of the *in vivo* model, in which gene expression in myeloid cells is affected not only by the specific manipulation, but also by any consequences of the genetic intervention on cardiac remodeling and dysfunction. An additional factor is that the heatmap illustrates z scores (in comparison to the mean expression for the same gene in all samples). For low expression genes, this results in the appearance of high variability at expression levels; however (due to the very low expression), biological significance is unclear. We have added a brief comment in the figure legend (Supplemental Figure XIX) indicating this. Moreover, for this reason, identification of candidate ITGA5-mediated signals and pathways was based on consistent findings in both *in vivo* and *in vitro* data.

REFERENCES

1. Chen B, Huang S, Su Y, Wu YJ, Hanna A, Brickshawana A, Graff J, Frangogiannis NG. Macrophage Smad3 Protects the Infarcted Heart, Stimulating Phagocytosis and Regulating Inflammation. *Circ Res*. 2019; 125:55-70.
2. Chen B, Li R, Hernandez SC, Hanna A, Su K, Shinde AV, Frangogiannis NG. Differential effects of Smad2 and Smad3 in regulation of macrophage phenotype and function in the infarcted myocardium. *J Mol Cell Cardiol*. 2022; 171:1-15.
3. Jia D, Chen S, Bai P, Luo C, Liu J, Sun A, Ge J. Cardiac Resident Macrophage-Derived Legumain Improves Cardiac Repair by Promoting Clearance and Degradation of Apoptotic Cardiomyocytes After Myocardial Infarction. *Circulation*. 2022; 145:1542-1556.
4. Tan P, He L, Xing C, Mao J, Yu X, Zhu M, Diao L, Han L, Zhou Y, You MJ, et al. Myeloid loss of Beclin 1 promotes PD-L1hi precursor B cell lymphoma development. *J Clin Invest*. 2019; 129:5261-5277.
5. Bosmans LA, van Tiel CM, Aarts S, Willemsen L, Baardman J, van Os BW, den Toom M, Beckers L, Ahern DJ, Levels JHM, et al. Myeloid CD40 deficiency reduces atherosclerosis by impairing macrophages' transition into a pro-inflammatory state. *Cardiovasc Res*. 2022.
6. Guo Y, Liu Y, Zhao S, Xu W, Li Y, Zhao P, Wang D, Cheng H, Ke Y, Zhang X. Oxidative stress-induced FABP5 S-glutathionylation protects against acute lung injury by suppressing inflammation in macrophages. *Nature communications*. 2021; 12:7094.
7. Fu Y, Pajulas A, Wang J, Zhou B, Cannon A, Cheung CCL, Zhang J, Zhou H, Fisher AJ, Omstead DT, et al. Mouse pulmonary interstitial macrophages mediate the pro-tumorigenic effects of IL-9. *Nature communications*. 2022; 13:3811.
8. Qin Y, Jia L, Liu H, Ma W, Ren X, Li H, Liu Y, Li H, Ma S, Liu M, et al. Macrophage deletion of Noc4l triggers endosomal TLR4/TRIF signal and leads to insulin resistance. *Nature communications*. 2021; 12:6121.

9. Gunderson AJ, Yamazaki T, McCarty K, Fox N, Phillips M, Alice A, Blair T, Whiteford M, O'Brien D, Ahmad R, et al. TGFbeta suppresses CD8(+) T cell expression of CXCR3 and tumor trafficking. *Nature communications*. 2020; 11:1749.
10. Tong J, Wang X, Liu Y, Ren X, Wang A, Chen Z, Yao J, Mao K, Liu T, Meng FL, et al. Pooled CRISPR screening identifies m(6)A as a positive regulator of macrophage activation. *Sci Adv*. 2021; 7.
11. Guo X, Kimura A, Namekata K, Harada C, Arai N, Takeda K, Ichijo H, Harada T. ASK1 signaling regulates phase-specific glial interactions during neuroinflammation. *Proc Natl Acad Sci U S A*. 2022; 119.
12. Peng Z, Li X, Li J, Dong Y, Gao Y, Liao Y, Yan M, Yuan Z, Cheng J. Dlg1 Knockout Inhibits Microglial Activation and Alleviates Lipopolysaccharide-Induced Depression-Like Behavior in Mice. *Neurosci Bull*. 2021; 37:1671-1682.
13. Huo S, Ren J, Ma Y, Ozathaley A, Yuan W, Ni H, Li D, Liu Z. Upregulation of TRPC5 in hippocampal excitatory synapses improves memory impairment associated with neuroinflammation in microglia knockout IL-10 mice. *J Neuroinflammation*. 2021; 18:275.
14. Zhang B, Zou J, Han L, Beeler B, Friedman JL, Griffin E, Piao YS, Rensing NR, Wong M. The specificity and role of microglia in epileptogenesis in mouse models of tuberous sclerosis complex. *Epilepsia*. 2018; 59:1796-1806.
15. Zeyen T, Noristani R, Habib S, Heinisch O, Slowik A, Huber M, Schulz JB, Reich A, Habib P. Microglial-specific depletion of TAK1 is neuroprotective in the acute phase after ischemic stroke. *J Mol Med (Berl)*. 2020; 98:833-847.
16. Sanchis P, Fernandez-Gayol O, Vizueta J, Comes G, Canal C, Escrig A, Molinero A, Giralto M, Hidalgo J. Microglial cell-derived interleukin-6 influences behavior and inflammatory response in the brain following traumatic brain injury. *Glia*. 2020; 68:999-1016.
17. Chen B, Li R, Kubota A, Alex L, Frangogiannis NG. Identification of macrophages in normal and injured mouse tissues using reporter lines and antibodies. *Sci Rep*. 2022; 12:4542.
18. Paulitti A, Andreuzzi E, Bizzotto D, Pellicani R, Tarticchio G, Marastoni S, Pastrello C, Jurisica I, Ligresti G, Bucciotti F, et al. The ablation of the matricellular protein EMILIN2 causes defective vascularization due to impaired EGFR-dependent IL-8 production affecting tumor growth. *Oncogene*. 2018; 37:3399-3414.
19. Han Z, Ni J, Smits P, Underhill CB, Xie B, Chen Y, Liu N, Tylzanowski P, Parmelee D, Feng P, et al. Extracellular matrix protein 1 (ECM1) has angiogenic properties and is expressed by breast tumor cells. *FASEB J*. 2001; 15:988-994.

REVIEWER COMMENTS

Reviewer #1 (Remarks to the Author):

No further comments

Reviewer #2 (Remarks to the Author):

The manuscript “Macrophage-specific integrin $\alpha 5$ protects the infarcted heart from adverse remodeling by stimulating angiogenesis through PI3K and FAK pathways” reports an interesting observation that the deletion of integrin $\alpha 5$ in macrophages exacerbates cardiac response to injury. The authors performed a large number of experiments and supplied additional data in response to reviewers’ critics. This reviewer believes that the injury data is interesting and compelling. However, the mechanistic aspect of the paper is weak and provided data does not fully support authors’ conclusions that integrin $\alpha 5$ functions in macrophages to regulate VEGF A expression through the activation of FAK and PI3K.

Major:

The quality of IF is insufficient to convince the reviewer that stained cells express Itga5 and/or CSFR1. Thin confocal slices are necessary to prove the localization of Itga5 protein and CSFR1GFP signal in the same cell. In addition:

- 1) Itga5 is a transmembrane protein. However, the Itga5 signal in IF panels (Fig. 2 and Sup Fig. III) appears cytoplasmic and appears to co-localize with the cytoplasmic GFP from the CSFR1RGFP reporter. Can authors explain this?
- 2) There appears to be either a leaky expression of CSFR1RGFP or too much autofluorescence making it difficult to discern a legitimate signal from the background in Sup. Fig. III (Compare IIIA with IIIE, and IIIF). Arrows in IIIF point to different cells in different panels. For example, the right arrow in the middle panel points to a round green cell. This is not the same cell in the right-most (merged) panel.
- 3) Arrow placements in IF panels are sloppy and should be checked (see additional specific comments below).

4) Could the authors confirm the presence of Itga5 on the macrophage cell surface? If not, the authors should comment on the apparent cytoplasmic localization of Itga5 in macrophages and explain how cytoplasmic Itga5 regulates signaling to induce VEGFA

5) Methods should be included describing how the IF staining was done (for FACS and slides), including the buffers, stating the inclusion of detergents to permeabilize cells. Antibody information and dilutions need to be stated.

Figure 8B. Cluster labels are shown right on top of the data points in the UMAP plot making it difficult to evaluate Itga5 expression. The differential expression of Itga5 among clusters is not obvious from the violin plot. Please provide p-values.

Arrows in Fig. 1B (28dl) cover the cells to which they are meant to point.

Arrows in XIC are sloppily positioned and point to different cells in different panels. Only part of Mac2+ cell in the top row appears to have cytoplasmic Itga5 expression.

Full Western blots:

A single western blot is shown for each condition. It is not clear how many times WB analyses were performed for each condition. For example:

t-Akt blot for Fig. 10C. Additional bands appear in treated samples at ~20kDa that are not present in controls. Is this reproducible? There also seems to be more protein stuck at the stacking/running gel interface for the mutant samples. This potentially could alter the quantification of tAkt.

Similarly, more protein reactive with pFAK is accumulated at the stacking/running gel interface, and this is likely to alter the quantitative analysis. The authors should adjust gel loading conditions.

These experiments are challenging but need to be done better to be convincing.

Reviewer #3 (Remarks to the Author):

The authors have sufficiently answered most of the questions of this reviewer. With regard to the new data, there are a few points that remain to be addressed though.

1. The title and abstract suggest a direct mechanism of macrophages regulating angiogenesis and thereby cardiac remodeling. In the absence of a clearly defined mechanism and the limitations discussed in the first round of review, the authors need to tone down these statements in the title and abstract. I do not think that the changes to the discussion section are sufficient here. So, my original concern still holds and needs to be addressed: 5) Capillary rarefaction has been reported during cardiac remodeling in many studies. Whether the observed (modest) effects on remodelling are causally related to macrophage-mediated effects on angiogenesis appears not clear and the authors should tone down the respective claims.
2. It is appreciated that the authors performed a single cell sequencing experiment to address some of the previous questions. The experiment is carried out on Csf1r-positive immune cells. What was the rationale to exclude Csf1r-negative cells?
3. A feature plot depicted in Figure 8B (and heatmap in Fig 8A) indicates that the purported subgroup of angiogenic macrophages express modestly higher levels of ITGA5 compared with other macrophage subpopulations. They appear to make up only 5% of the total Csf1r positive cells. Is that level of expression significantly different from other populations? Can the authors provide feature plots for secreted factors Ecm1 and emilim1? Can this be due to contaminating cells?
4. It is appreciated that the authors performed new flow cytometry experiments to demonstrate the upregulation of ITGA5 in macrophages (Figure 1). As this is not immediately visible with the histogram representation, authors are suggested to present this as a density plot.
5. Figure 4: were there differences in the neutrophil population?

Reviewer #4 (Remarks to the Author):

Minor:

- 1) The authors should use the same color code to identify specific cell populations in the t-SNE plot (Figure 7A) and the heatmaps (Figure 7B-D & 8A).

2) Figure design needs to be improved (e.g. hamonizing font sizes, and placement of the panel labels etc. etc.). Most panels look like the 'raw data', visually not appealing. I think this is important too.

RESPONSE TO THE REVIEWERS:

We are grateful to the reviewers and editors for their comments. We have revised our manuscript following their recommendations.

In the revised version of our manuscript, we have addressed all remaining concerns. Specifically:

- 1. To address concerns raised by reviewer 2, we have performed new experiments and new confocal imaging studies to show the localization of ITGA5 in macrophages. Confocal imaging and flow cytometry clearly demonstrate ITGA5 localization on the cell membrane. Cytoplasmic localization was also noted in immunofluorescence studies and likely reflects the de novo synthesis of ITGA5 in infarct macrophages. The new data are shown in Figure 1 and in Supplemental Figures III and IV.*
- 2. To address comments by Reviewers 2, 3 and 4, we have expanded the scRNA-seq analysis to show the cluster-specific patterns of integrin expression in macrophages and to address important presentation problems. The p-values are clearly shown, a new supplemental figure shows all the integrins which are differentially expressed in each cluster (Supplemental Figure XIX), color coding is now consistent and the clusters exhibiting Itga5 expression are clearly shown. The data are shown in the revised versions of Figures 7, 8, Supplemental Figure XIX, and XX.*
- 3. To address comments by Reviewer 3, we show cluster-specific patterns of expression of the angiogenic matricellular proteins ECM1 and EMILIN2. The data show that the angiogenic macrophage cluster has higher expression levels of Ecm1 and Emilin2 than other clusters. The new data are shown in Supplemental Figure XXVIII.*
- 4. We have included the quantitative data on the effects of myeloid cell-specific ITGA5 loss on neutrophil density (Figure 4) and we have revised Figure 1 to show density plots.*
- 5. In response to comments by Reviewer 3, we have revised the title, abstract and discussion to delete statements suggesting a causative relation between and increased capillary density and attenuated remodeling, while indicating the association.*
- 6. We have also addressed concerns by Reviewer 1 regarding bands in the stacking/running gel interface present in 2 of the unedited Western blots. We explain why these bands do not affect the quantitative analysis and the conclusions of these experiments. We also point out (and show ample evidence) that similar imperfections in Western blots are commonly found in the majority of papers published in top-quality journals (including Nature family publications, such as Nature, Nat Cell Biol, Nat Commun).*
- 7. We have addressed all other comments raised by the reviewers.*

8. *Dr Shuaibo Huang who contributed to the revisions has been added to the authors' list.*

Specifically, we have addressed the reviewers' comments as follows:

Reviewer #1 (Remarks to the Author):

No further comments.

Thank you very much for your careful review of our manuscript and for your kind and insightful comments.

Reviewer #2 (Remarks to the Author):

The manuscript "Macrophage-specific integrin $\alpha 5$ protects the infarcted heart from adverse remodeling by stimulating angiogenesis through PI3K and FAK pathways" reports an interesting observation that the deletion of integrin $\alpha 5$ in macrophages exacerbates cardiac response to injury. The authors performed a large number of experiments and supplied additional data in response to reviewers' critics. This reviewer believes that the injury data is interesting and compelling. However, the mechanistic aspect of the paper is weak and provided data does not fully support authors' conclusions that integrin $\alpha 5$ functions in macrophages to regulate VEGF A expression through the activation of FAK and PI3K.

Thank you very much for your careful review of our manuscript and for your kind and insightful comments. We have revised our manuscript following your recommendations.

Major:

The quality of IF is insufficient to convince the reviewer that stained cells express Itga5 and/or CSFR1. Thin confocal slices are necessary to prove the localization of Itga5 protein and CSFR1GFP signal in the same cell.

We have replaced all previous images with better quality figures and we have performed new confocal imaging experiments to show the localization of ITGA5 in macrophages. Specifically, the following revisions were implemented:

a. ITGA5 localization in CSF1R+ macrophages is now clearly demonstrated in triple fluorescence images (ITGA5/CSF1R(EGFP)/DAPI, 7 days after MI). The new images are shown in Figure 1D-I of the revised version of the manuscript, and in Reviewer Figure 1 in the current response.

Reviewer Figure I: Dual immunofluorescence was performed in myocardial sections from infarcted $CSF1R^{EGFP}$ macrophage reporter mice (7 days coronary occlusion), combining ITGA5 staining (red) and CSF1R (GFP) labeling. Panels D-F show low magnification images (scalebar=50 μ m). Panels G-I show high magnification images (Scalebar=30 μ m, panel G shows the rectangular area indicated in panel D). Abundant ITGA5+ macrophages were noted in 7-day infarcts (D-I, arrows). CSF1R-negative cells with vascular, or fibroblast morphology (D-F, yellow arrow) also expressed ITGA5. Quantitative analysis is shown in Figure 1J-K. The complete time course of macrophage ITGA5 expression after MI is shown in Supplemental Figure III.

b. the figure showing the time course of ITGA5 expression has been replaced with a new one with better quality images (Supplemental Figure III)

c. new confocal images are shown to demonstrate ITGA5 localization in CSF1R+ cells. Confocal images are shown in Figure 1L-N and in this response as Reviewer Figure II.

Reviewer Figure II: Confocal microscopy shows that ITGA5 is localized not only on the macrophage surface (white arrows), but also in the cytoplasm (yellow arrow), likely reflecting *de novo* synthesis of ITGA5, followed by subsequent shuttling to the cell membrane. Scalebar=10 μ m.

Moreover, additional confocal images of ITGA5/CSF1R(EGFP)/WGA/DAPI-stained sections are shown in Supplemental Figure IV. WGA binds to glycoproteins on the cell membrane and extracellular matrix, and was added to provide additional evidence of cell surface localization.

1) Itga5 is a transmembrane protein. However, the Itga5 signal in IF panels (Fig. 2 and Sup Fig. III) appears cytoplasmic and appears to co-localize with the cytoplasmic GFP from the CSF1RGFP reporter. Can authors explain this?

4) Could the authors confirm the presence of Itga5 on the macrophage cell surface? If not, the authors should comment on the apparent cytoplasmic localization of Itga5 in macrophages and explain how cytoplasmic Itga5 regulates signaling to induce VEGFA.

*Our flow cytometry studies were performed in the absence of permeabilization protocols. Thus, our flow cytometry data (Figure 1) and the confocal imaging studies (Figure 1L-N and Supplemental Figure IV) clearly show that ITGA5 is localized on the macrophage cell surface. The figures in the revised version of the manuscript clearly illustrate the cell membrane localization. As the reviewer points out, some cells exhibit cytoplasmic labeling. This likely reflects *de novo* synthesis of ITGA5. α and β integrin chains form heterodimers in the endoplasmic reticulum after their synthesis(1). Heterodimers are glycosylated in the ER(2), then undergo additional processing in the Golgi, until delivered to the cell membrane(3). Moreover, cell surface integrins are recycled through endocytosis(4). Thus, cytoplasmic localization of ITGA5 in infarct macrophages may reflect *de novo* synthesis (which is followed by shuttling to the membrane), or endocytotic cycling.*

2) There appears to be either a leaky expression of CSF1RGFP or too much autofluorescence making it difficult to discern a legitimate signal from the background in Sup. Fig. III (Compare IIIA with III E, and III F). Arrows in III F point to different cells in different panels. For example, the right arrow in the middle panel points to a round green cell. This is not the same cell in the right-most (merged) panel.

3) Arrow placements in IF panels are sloppy and should be checked (see additional specific comments below).

The figures have been replaced. The new IF and confocal imaging figures (Figure 1D-N, Supplemental Figures II and III) clearly show the co-localization.

5) Methods should be included describing how the IF staining was done (for FACS and slides), including the buffers, stating the inclusion of detergents to permeabilize cells. Antibody information and dilutions need to be stated.

We have added this information in the Supplement. Supplemental methods describe in detail the buffers used for flow cytometry (Pages 14-15). Information on the antibodies (including the dilutions used) is provided in Supplemental Tables XII and XIII. No detergents were used to permeabilize the cells in the flow cytometry studies.

Detailed information on the immunofluorescence is provided in pages 6-8 of the Supplement. A table with the antibodies used for IF and IHC studies including the dilutions is provided in Supplemental Table XI. Triton X was used in the immunofluorescence studies.

Figure 8B. Cluster labels are shown right on top of the data points in the UMAP plot making it difficult to evaluate Itga5 expression. The differential expression of Itga5 among clusters is not obvious from the violin plot. Please provide p-values.

We have revised panel 8B to avoid overlap of the cluster labels with the data points. We have also added indicators of statistical significance in all violin plots in Figure 8. Moreover, we have added a new supplemental figure (Supplemental Figure XIX), showing the cluster-specific patterns of integrin expression in macrophages.

Arrows in Fig. 1B (28dl) cover the cells to which they are meant to point.

We have replaced the figure.

Arrows in XIC are sloppily positioned and point to different cells in different panels. Only part of Mac2+ cell in the top row appears to have cytoplasmic Itga5 expression.

We have revised the figure to address these concerns (Supplemental Figure XIII in the revised version of the manuscript).

Full Western blots:

A single western blot is shown for each condition. It is not clear how many times WB analyses were performed for each condition.

The unedited gel file now includes all the Western Blots that were performed in the study and included in the quantitative analysis. Each sample represents an independent experiment. The use of scatter/box plots in the corresponding bar graphs and the information provided in the figure legends clearly indicates the number of experiments performed.

t-Akt blot for Fig. 10C. Additional bands appear in treated samples at ~20kDa that are not present in controls. Is this reproducible?

The low molecular weight band was noted in all 3 ITGA5 inhibition experiments.

There also seems to be more protein stuck at the stacking/running gel interface for the mutant samples. This potentially could alter the quantification of tAkt.

The reviewer points out to bands close to the stacking/running gel interface, noted in both IgG and anti-ITGA5 samples. (Reviewer Figure III) The average intensity of these bands is not different between the IgG and anti-ITGA5 groups. In contrast, there is a marked difference in the intensity of the specific t-Akt band between antibody-treated and IgG-treated samples. Thus, these bands should not influence the validity of the quantitative analysis. Moreover, the differences in p-Akt levels (also shown in the figure) are also quite impressive, supporting the effect of ITGA5 on Akt activation

Full unedited gel for Figure 10C

T-Akt antibody (Cell Signaling Technology, #9272)

Reviewer Figure III: *The unedited gel for the total Akt Western blot shown in Figure 10C. Bands in the stacking/running gel interface are noted. These bands are not different between groups and do not influence quantitation of the specific Akt band. Moreover, the marked difference in the intensity of the specific t-Akt band between anti-ITGA5 samples and IgG controls supports a major effect of ITGA5 on Akt levels that is further supported by similar differences in p-Akt levels (shown in Figure 10C).*

Similarly, more protein reactive with pFAK is accumulated at the stacking/running gel interface, and this is likely to alter the quantitative analysis. The authors should adjust gel loading conditions.

The reviewer points out bands close to the stacking/running gel interface in the p-FAK western blot (Reviewer Figure IV). Again, the intensity of these bands is very similar between the antibody and IgG groups and would not influence the quantitative analysis.

Full unedited gel for Figure 10C

P-fak antibody (Cell Signaling Technology, #3283)

Reviewer Figure IV: *The unedited gel for the p-FAK Western blot shown in Figure 10C. Bands in the stacking/running gel interface are noted. These bands are not different between groups and do not influence quantitation of the specific Akt band.*

To further address the concerns raised by the reviewer regarding the presence of signals close to the stacking/running gel interface, we reviewed unedited gels from recent papers published in Nature family journals (Nature, Nature Cell Biol, etc). We noted similar “imperfections” in western blotting images in the majority of published papers. In fact, in more than 50% of the published manuscripts the unedited gel images are truncated and

do not even show the stacking/running gel interface. In the manuscripts that show the full gel, the majority include one or more blots with similar issues. Examples are shown below in Reviewer Figures V-IX:

Reviewer Figure V: *Unedited gel in the supportive data from Ranek et al (Nature 2019)(5). Bands in the stacking/running gel interface are shown in the green rectangle.*

[EDITORIAL NOTE: figure redacted]

Reviewer Figure VI: *unedited gel in the supportive data from Ranek et al (Nature 2019)(5). Bands in the stacking/running gel interface are shown by the green arrows. Intensity is highly variable between samples.*

[EDITORIAL NOTE: figure redacted]

Reviewer Figure VII: *Unedited gel in the supportive data from Ranek et al (Nature 2019)(5). Bands in the stacking/running gel interface are shown by the green arrow. Intensity is highly variable between samples.*

[EDITORIAL NOTE: figure redacted]

Reviewer Figure VIII: *Unedited gel from Lee et al (Nature 2020) (6). Bands in the stacking/running gel interface are clearly noted. Intensity is highly variable between samples*

[EDITORIAL NOTE: figure redacted]

Reviewer Figure IX: *Similar issues (green arrow) in study by Juste et al (Nat Cell Biol 2021) (7).*

[EDITORIAL NOTE: figure redacted]

Reviewer #3 (Remarks to the Author):

The authors have sufficiently answered most of the questions of this reviewer. With regard to the new data, there are a few points that remain to be addressed though.

Thank you very much for your careful review of our manuscript and for your kind and insightful comments. We have revised our manuscript following your recommendations.

1. The title and abstract suggest a direct mechanism of macrophages regulating angiogenesis and thereby cardiac remodeling. In the absence of a clearly defined mechanism and the limitations discussed in the first round of review, the authors need to tone down these statements in the title and abstract. I do not think that the changes to the discussion section are sufficient here. So, my original concern still holds and needs to be addressed: 5) Capillary rarefaction has been reported during cardiac remodeling in many studies. Whether the observed (modest) effects on remodeling are causally related to macrophage-mediated effects on angiogenesis appears not clear and the authors should tone down the respective claims.

*We agree that the data do not document a causative relation between the macrophage-mediated ITGA5-dependent effects on angiogenesis and the improved post-MI remodeling. Thus, we have revised the title, abstract, introduction (page 4) and discussion (page 32) to indicate the association. The revised title is: **“Macrophage-specific integrin $\alpha 5$ exerts protective actions on the infarcted heart, associated with stimulation of angiogenesis”**.*

2. It is appreciated that the authors performed a single cell sequencing experiment to address some of the previous questions. The experiment is carried out on Csf1r-positive immune cells. What was the rationale to exclude Csf1r-negative cells?

The rationale for the focus on the CSF1R+ cells was to obtain a high-resolution analysis of the myeloid cell transcriptome in control and infarcted hearts.

3. A feature plot depicted in Figure 8B (and heatmap in Fig 8A) indicates that the purported subgroup of angiogenic macrophages express modestly higher levels of ITGA5 compared with other macrophage subpopulations. They appear to make up only 5% of the total Csf1r positive cells. Is that level of expression significantly different from other populations?

We have revised figure 8 to show the statistically significant differences in integrin expression between clusters. Itga5 expression was significantly higher in the cluster of angiogenic macrophages (Amp) in comparison to all other clusters (logFC=2.82, padj=1E-142). Moreover, we have added Supplemental Figure XIX to show in detail the cluster-specific patterns of integrin expression, listing all statistically significant differences. Supplemental Figure XIX is also shown in this response as Reviewer Figure X

	LogFC	padj		LogFC	padj		LogFC	padj		LogFC	padj
RMp						PMp				Fib	
Itga6	4.67	0	Imp	2.96	1.95E-302	Itga4	-0.73	1.1E-24	Itga1	7.88	0
Itga4	-1.45	0	Itgal	2.48	4.54E-237	Itgb1	0.82	1.3E-15	Itgb1	-0.85	4.6E-11
Itgb2	-1.13	0	Itga6	-3.72	1.59E-207	Itga5	1.27	2.3E-15			
Itgb5	4.18	0	Itgam	2.02	1.31E-153	Itga6	2.02	4.7E-15	Mp11		
Itgam	-0.93	4.93E-261	Itgb2	1.84	1.01E-134	Itgax	1.03	1.4E-13	Itgal	6.78	1E-99
Itgav	0.99	2.38E-94	Itgb5	-1.29	5.56E-77	Itgb2	0.51	1.9E-05	Itga4	3.94	1.1E-41
			Itgb1	-0.52	5.62E-67	Itgam	0.52	0.01	Itgb1	1.47	3.6E-18
Mo			Itga4	1.21	5.93E-58				Itgb2	2.6	3.4E-17
Itga4	4.47	0	Itga6	-0.66	1.85E-16	Amp			Itgax	2.61	2.6E-11
Itgal	7.42	0				Itgax	2.64	6E-143	Itgb7	2.02	1.4E-07
Itgb2	2.93	0	Mp5			Itga5	2.82	1E-142	Itga9	1.93	1.6E-06
Itgb7	3.86	0	Itgam	3.52	0	Itga4	-1.57	5E-88	Itga6	-2.02	1.1E-05
Itgb5	-4.85	0	Itgb2	2.19	0	Itgb2	2.02	1.5E-70	Itgam	2	1.7E-05
Itgam	1.31	1.72E-103	Itgb5	-2.2	5.28E-129	Itgav	2.12	1.6E-55	Itgb5	-0.71	2.7E-05
			Itga6	-2.02	4.94E-58	Itga6	2.4	2.3E-52	Itgav	2	0.0001
CMp						Itgb1	0.77	1.7E-16			
Itga9	2.09	1.64E-292	Dc			Itgb5	1.79	4.5E-16	Mp12		
Itgb5	-2.28	1.12E-223	Itgax	4.78	0				Itgax	4.98	1.3E-98
Itga6	-2.64	1.91E-219	Itgb7	5.85	0	Gr			Itga4	2.06	9E-18
Itgb2	-2.04	1.49E-204	Itgav	-2.06	6.22E-65	Itgb1	-5.09	8.4E-60	Itgam	2.06	5.5E-15
Itgb1	0.55	1.77E-195	Itgam	-1.29	1.48E-44	Itgal	2.03	2.2E-16	Itgb2	1.87	3E-12
Itgav	-1.45	2.06E-71	Itga4	1.03	5.29E-25				Itgb1	0.87	9.8E-07
Itgam	-1.23	3.07E-53	Itgb2	0.72	0.000000795						

Reviewer Figure X: Cluster-specific patterns of integrin expression in CSF1R+ myeloid cells. Integrin genes upregulated or downregulated in specific clusters, in comparison with all other clusters are shown. The angiogenic macrophage cluster (Amp) and the proliferative macrophage cluster (PMp) had higher levels of *Itga5* expression, in comparison to all other clusters.

Can the authors provide feature plots for secreted factors *Ecm1* and *emilin2*? Can this be due to contaminating cells?

We now show the feature plots and violin plots for Ecm1 and Emilin2. The new data are shown in Supplemental Figure XXVIII and in this document as Reviewer Figure XI. The data clearly show that the in vivo findings do not reflect contamination, but rather expression of these angiogenic matricellular genes by specific myeloid cell clusters. The angiogenic macrophage cluster (Amp) expressed significantly higher levels of both Ecm1 and Emilin2 than other clusters, supporting the notion that ITGA5 may be involved in upregulation of these angiogenic genes. Ecm1 was also highly expressed in RMp, Mp5 and Mp12 clusters, whereas monocytes, granulocytes and the macrophage clusters Imp, CMp, Mp11 and Mp5 also expressed high levels of Emilin2 than other clusters. Thus, ECM1 and EMILIN2 may be additional ITGA5-mediated macrophage-derived angiogenic mediators.

Reviewer Figure XI: Cluster-specific patterns of Emilin2 and Ecm1 expression in CSF1R+ myeloid cells. ScRNA-seq showed that both Ecm1 and Emilin2 were broadly expressed by CSF1R+ macrophage clusters. A, C: Angiogenic macrophages (Amp), reparative macrophages (RMp), and the Mp5 and Mp12 clusters had higher expression of Ecm1 than other clusters. D-E: Angiogenic macrophages (Amp), monocytes (Mo), granulocytes (Gr), and the Imp, CMp, Mp5 and Mp11 clusters had higher expression of Emilin2. Panel B illustrates the color coding for cluster

identification in violin plots.

4. It is appreciated that the authors performed new flow cytometry experiments to demonstrate the upregulation of ITGA5 in macrophages (Figure 1). As this is not immediately visible with the histogram representation, authors are suggested to present this as a density plot.

We have revised Figure 1 to show the density plots instead of the histogram.

5. Figure 4: were there differences in the neutrophil population?

There were no differences in neutrophil numbers. We have revised Figure 4 to show the quantitative analysis of neutrophils.

Reviewer #4 (Remarks to the Author):

Thank you very much for your careful review of our manuscript and for your kind and insightful comments. We have revised our manuscript following your recommendations.

Minor:

1) The authors should use the same color code to identify specific cell populations in the t-SNE plot (Figure 7A) and the heatmaps (Figure 7B-D & 8A).

We have revised figures 7 and 8 to provide consistent color codes.

2) Figure design needs to be improved (e.g. harmonizing font sizes, and placement of the panel labels etc. etc.). Most panels look like the 'raw data', visually not appealing. I think this is important too.

We have made every effort to improve the quality of the figures and to harmonize fonts and presentation.

REFERENCES

1. Ho MK, Springer TA. Tissue distribution, structural characterization, and biosynthesis of Mac-3, a macrophage surface glycoprotein exhibiting molecular weight heterogeneity. *J Biol Chem* 1983;258:636-42.
2. Gu J, Isaji T, Sato Y, Kariya Y, Fukuda T. Importance of N-glycosylation on alpha5beta1 integrin for its biological functions. *Biol Pharm Bull* 2009;32:780-5.
3. Rainero E, Norman JC. Late endosomal and lysosomal trafficking during integrin-mediated cell migration and invasion: cell matrix receptors are trafficked through the late endosomal pathway in a way that dictates how cells migrate. *Bioessays* 2013;35:523-32.
4. Bridgewater RE, Norman JC, Caswell PT. Integrin trafficking at a glance. *J Cell Sci* 2012;125:3695-701.
5. Ranek MJ, Kokkonen-Simon KM, Chen A et al. PKG1-modified TSC2 regulates mTORC1 activity to counter adverse cardiac stress. *Nature* 2019;566:264-269.
6. Lee J, Robinson ME, Ma N et al. IFITM3 functions as a PIP3 scaffold to amplify PI3K signalling in B cells. *Nature* 2020;588:491-497.
7. Juste YR, Kaushik S, Bourdenx M et al. Reciprocal regulation of chaperone-mediated autophagy and the circadian clock. *Nat Cell Biol* 2021;23:1255-1270.

REVIEWERS' COMMENTS

Reviewer #2 (Remarks to the Author):

The authors addressed all my concerns.

Reviewer #3 (Remarks to the Author):

Review of revised NCOMMS-22-08126B by Li et al.

The authors have done extensive work and prepared a careful revision. I appreciate that the title now refrains from the previous overstated claim. The authors have adequately addressed my previous concerns.